# Bayes-optimal learning of an extensive-width neural network from quadratically many samples

**Antoine Maillard**
Department of Mathematics
ETH Zürich, Switzerland

**Emanuele Troiani**
Statistical Physics Of Computation Laboratory
EPFL, Switzerland

**Simon Martin**
INRIA & Laboratoire de Physique
ENS, Université PSL, France

**Lenka Zdeborová**
Statistical Physics Of Computation Laboratory
EPFL, Switzerland

**Florent Krzakala**
Information Learning and Physics Laboratory
EPFL, Switzerland

## Abstract

We consider the problem of learning a target function corresponding to a single hidden layer neural network, with a quadratic activation function after the first layer, and random weights. We consider the asymptotic limit where the input dimension and the network width are proportionally large. Recent work [Cui et al., 2023] established that linear regression provides Bayes-optimal test error to learn such a function when the number of available samples is only linear in the dimension. That work stressed the open challenge of theoretically analyzing the optimal test error in the more interesting regime where the number of samples is quadratic in the dimension. In this paper, we solve this challenge for quadratic activations and derive a closed-form expression for the Bayes-optimal test error. We also provide an algorithm, that we call GAMP-RIE, which combines approximate message passing with rotationally invariant matrix denoising, and that asymptotically achieves the optimal performance. Technically, our result is enabled by establishing a link with recent works on optimal denoising of extensive-rank matrices and on the ellipsoid fitting problem. We further show empirically that, in the absence of noise, randomly-initialized gradient descent seems to sample the space of weights, leading to zero training loss, and averaging over initialization leads to a test error equal to the Bayes-optimal one.

## 1 Introduction

Learning with multi-layer neural networks brought impressive progress and applications in many areas. It is well established that a large enough non-linear neural network can represent a large class of functions [Cybenko, 1989]. Yet the conditions under which the values of the weights can be found efficiently, and from how many samples of the data, remain theoretically elusive. While one may hope that a detailed understanding of these fundamental limitations will eventually allow for a more efficient training, answering such questions for general data and target function remains, however, beyond the reach of current theoretical methods.

In an early attempt to overcome the difficulty of the above generic question, a long line of work originating in Gardner and Derrida [1989], Sompolinsky et al. [1990] proposed to study the optimal

38th Conference on Neural Information Processing Systems (NeurIPS 2024).

sample-complexity in the so-called teacher-student setting, where the target function corresponds to a "teacher" neural network. The architecture of this teacher neural network is chosen to be fully connected feed-forward with a given number of layers, their widths, and activations. The values of each of the weights are generated independently, from a Gaussian distribution. This teacher neural network is then used to generate an output label $y_i \in \mathbb{R}$ for each input data sample $\mathbf{x}_i \in \mathbb{R}^d$. Given the architecture of the teacher networks (but not the values of the teacher-weights $\mathbf{W}^*$) and the training set of input-output pairs $\{y_i, \mathbf{x}_i\}_{i=1}^n$, the smallest achievable test error can then be obtained by averaging the output of a student-neural network (with the same architecture as the teacher) over the values of weights drawn from the posterior distribution. We will refer to the accuracy reached this way as the Bayes-optimal one. It yields the fundamental limitations in learning such tasks, by any possible means, and can therefore serve as a benchmark.

In the so-called *high-dimensional limit* [Donoho, 2000], when the input training data are $d$-dimensional Gaussian vectors, in the limit $d \to \infty$, the above research program has been carried out in detail over the last decades for small neural networks having only $m = O_d(1)$ hidden units, and learning from $n = \alpha d$ data samples, where $\alpha = O_d(1)$ (see, e.g. Györgyi [1990], Opper and Haussler [1991], Seung et al. [1992], Watkin et al. [1993], Schwarze [1993], Barbier et al. [2019], Aubin et al. [2019b]). In the more recent literature, this setting is sometimes referred to as learning single-index and multi-index functions [Damian et al., 2022, Bietti et al., 2023, Collins-Woodfin et al., 2023, Dandi et al., 2023, 2024b, Damian et al., 2024]. While early works in this line originated in statistical physics and used the heuristic replica method [Mézard et al., 1987] to derive the closed-form expressions for quantities of interest in the high-dimensional limit (with $d \to \infty$, $m = O_d(1)$ and $n = O_d(d)$), a mathematical establishment followed using rigorous probabilistic methods [Barbier et al., 2019, Aubin et al., 2019b].

Reaching a closed-form expression for the Bayes-optimal sample complexity for target functions corresponding to multi-layer teacher neural networks is the next open and very challenging task. Among the recent work is Cui et al. [2023], that established (non-rigorously, using the replica method) the Bayes-optimal error for a target function corresponding to a multi-layer neural network of *extensive width* (i.e. linearly proportional to the dimension) from a number of samples also *linear* in the dimension. Interestingly, in this limit, the Bayes-optimal error resulted in a quite poor approximation of the function, which can be achieved as well by a simple linear regression on the input-output pairs. No method, be it a multi-layer neural network (or even refinements like a transformer), will be able to achieve better performance. [Cui et al., 2023] further argue, based on numerical evidence, that *quadratically* many samples in the dimension are necessary in order to be able to learn the target function with non-linear activations[1] to an infinitesimally small test error. This is perhaps intuitive as, with an extensive width, the number of parameters/weights in the teacher network is quadratic in dimension. However, such a regime is challenging for current theoretical tools. Reaching an analytical explicit expression for the Bayes-optimal performance in this regime, for the target function in the form of a neural network of extensive width, is an open, challenging, theoretical problem that has not yet been solved even for a single hidden layer architecture.

**Our contributions –** In this paper, we step up to this challenge and derive a closed-form expression for the Bayes-optimal test error for a target/teacher function corresponding to a one-hidden layer neural network of extensive width, from quadratically many samples, for a particular case where the activation function (after the hidden layer) is quadratic. In particular, our main contributions are:

- We provide a closed-form expression for the Bayes-optimal error of learning an extensive-width neural network from quadratically many samples, which is the first type of such result to the best of our knowledge. Such a form is enabled by the high-dimensional limit and corresponding concentration of quantities of interest. It notably follows from our formula that, in the absence of noise in the target function, zero test error is achievable for a sample complexity $\alpha = n/d^2$ larger than a perfect-recovery threshold $\alpha > \alpha_{\mathrm{PR}}$ where

$$\alpha_{\mathrm{PR}} = \kappa - \frac{\kappa^2}{2} \quad \text{if} \quad \kappa \leq 1; \qquad \alpha_{\mathrm{PR}} = \frac{1}{2} \quad \text{if} \quad \kappa \geq 1, \qquad (1)$$

  with $\kappa = m/d$ the ratio between the width $m$ and the dimension $d$. We further notice that this matches a naive counting of the number of degrees of freedom in the target function.

---

[1] Note that for linear activations, the target functions reduces to linear regression and can be learned from linearly many samples.

- We introduce the GAMP-RIE algorithm that combines generalized approximate message passing (GAMP) [Donoho et al., 2009, Rangan, 2011, Zdeborová and Krzakala, 2016] with a matrix denoiser that is based on so-called rotationally-invariant estimators (RIE) [Bun et al., 2016], and show that in the large size limit, this algorithm reaches the Bayes-optimal error for all $\alpha, \kappa = \Theta(1)$.
- On the technical level, our result is enabled by combining results from the analysis of single-layer neural networks [Barbier et al., 2019] and extensive-rank matrix denoising [Maillard et al., 2022b]. The derived formula involves the asymptotics of the Harish-Chandra-Itzykson-Zuber integral of random matrix theory [Harish-Chandra, 1957, Itzykson and Zuber, 1980]. Our approach is notably inspired by recent results on the ellipsoid fitting problem [Maillard and Kunisky, 2024, Maillard and Bandeira, 2023]. These tools are of independent interest to the machine learning community, and we anticipate they will have other applications in the theory of learning.
- We empirically compare the Bayes-optimal performance to the one obtained by gradient descent. In the noiseless case we observe a rather unusual and surprising scenario, as randomly-initialized gradient descent seems to be sampling the space of interpolants, and leads to twice the Bayes-optimal error. When averaged over initialization the gradient descent reaches an error that is very close to the Bayes-optimal. The rigorous establishment of these properties of gradient descent is left open.

Our experiments are reproducible, and accessible freely in a public repository [Maillard et al., 2024].

**Further related works –** The problem studied in this work is known as *phase retrieval* in the case of a single hidden unit ($m = 1$). Many works considered this problem in the high-dimensional limit $d \to \infty$, in the regime of $n = O(d \log d)$ samples; see e.g. Candes et al. [2013], Chen et al. [2019], Demanet and Hand [2014]. A subsequent line of work established that the problem can be solved with only $O(d)$ samples [Candès and Li, 2014, Chen and Candes, 2015, Cai et al., 2022].

Eventually, for Gaussian i.i.d. input data and i.i.d. teacher weights, the optimal sample complexity for learning phase retrieval in the high-dimensional limit has been established down to the constant in $\alpha = n/d$. Authors of Mondelli and Montanari [2019] derived the weak recovery threshold for the noiseless case to be $\alpha_{\mathrm{WR}} = 1/2$ for phase retrieval, and optimal spectral methods were shown to match this threshold in Luo et al. [2019], Maillard et al. [2022a]. The information-theoretically optimal accuracy and the one achieved by an approximate message passing algorithm were then derived in Barbier et al. [2019] for a general i.i.d. prior for the teacher weights. In the absence of noise, these results imply sample complexities $\alpha_{\mathrm{IT}} = 1$ and $\alpha_{\mathrm{AMP}} \approx 1.13$ needed to achieve perfect learning for a Gaussian prior. Authors of Song et al. [2021] proposed a non-robust polynomial algorithm capable of solving noiseless phase retrieval for $\alpha \geq \alpha_{\mathrm{IT}}$. Algorithms based on gradient descent were argued not to achieve the optimal sample complexity in Sarao Mannelli et al. [2020a], Mignacco et al. [2021]. Maillard et al. [2020] derived the MMSE for more general input data distributions, including the complex-valued case. Phase retrieval with generative priors was studied in Hand et al. [2018], Aubin et al. [2020]. We refer to a recent review [Dong et al., 2023] for an overview of the relations between these theoretical studies and practical applications of phase retrieval in imaging.

The case with different numbers of hidden units $m^\star$ in the teacher and $m$ in the student model, was also discussed in the literature. For $m^* = O_d(1)$, the problem is a special case of a multi-index model that has been recently actively considered, e.g. in Aubin et al. [2019b], Bietti et al. [2023], Damian et al. [2022, 2024], Collins-Woodfin et al. [2023], Dandi et al. [2023, 2024b]. This line of work has not focused on the quadratic activations, as it does not bring particular simplification in this case.

The geometry of loss landscapes of one hidden-layer networks with quadratic activations was studied, and the absence of spurious local minima was established for $m \geq d$ (when the read-out layer is fixed as in our setting) in Du and Lee [2018]. Similar results were established in Soltanolkotabi et al. [2018], Venturi et al. [2019] for a slightly more general setting where the readout layer is learned.

Establishing results about sample complexity required for generalization in cases where $m$ (or both $m$ and $m^*$) are $\Theta(d)$ is technically challenging, and so far, only a handful of works made progress in that direction. In particular, Gamarnik et al. [2019] considered $m^* \geq d$ and $m \geq d$, and have shown that a sample complexity $n \geq d(d + 1)/2$ is sufficient for perfect recovery of the target function. Sarao Mannelli et al. [2020b] considered the overparametrized case with $m^* = O_d(1)$ and $m > d$, and showed that gradient descent reaches exact recovery for a sample complexity $n > d(m^* + 1) - (m^* + 1)m^*/2$, again considering the high-dimensional limit. Gradient descent of the population risk has been studied for general values of $(m^*, m)$ in Martin et al. [2024], along with a discussion of the role of overparametrization.

## 2 Setting

As discussed above, we are studying the Bayes-optimal accuracy in the *teacher-student* setting. More concretely, we consider a dataset of $n$ samples $\mathcal{D} = \{y_i, \mathbf{x}_i\}_{i=1}^n$ where the input data is normal Gaussian of dimension $d$: $(\mathbf{x}_i)_{i=1}^n \overset{\text{i.i.d.}}{\sim} \mathcal{N}(0, \mathrm{I}_d)$. We then draw i.i.d. $d$-dimensional teacher-weight vectors $(\mathbf{w}_k^*)_{k=1}^m \overset{\text{i.i.d.}}{\sim} \mathcal{N}(0, \mathrm{I}_d)$, and noise $(\mathbf{z}_i)_{i=1}^n \overset{\text{i.i.d.}}{\sim} \mathcal{N}(0, \mathrm{I}_m)$. Finally, the output labels $(y_i)_{i=1}^n$ are obtained by a one-hidden layer teacher network with $m$ hidden units and quadratic activation:

$$ y_i = f_{\mathbf{W}^*}(\mathbf{x}_i) := \frac{1}{m} \sum_{k=1}^m \left[ \frac{1}{\sqrt{d}} (\mathbf{w}_k^*)^\intercal \mathbf{x}_i + \sqrt{\Delta} z_{i,k} \right]^2 . \tag{2} $$

Crucially, we assume we know the form of the (stochastic) target function $f_{\mathbf{W}^*}(\cdot)$ (i.e. the value of $m$, $\Delta$, and the form of eq. (2), including the fact that the activation function is quadratic) but we do not know the realization of neither the teacher weights $\mathbf{W}^* = (\mathbf{w}_1^\star, \cdots, \mathbf{w}_m^\star)$ nor the noise $\mathbf{z}_i$.

**Universality over the noise and weights distribution** – While we consider Gaussian distributions for the sake of our theoretical analysis, we expect our results to hold under more general i.i.d. models on both the noise and the teacher weights, under mild conditions of existence of moments. This is related to a recent conjecture of Semerjian [2024], see Sections 3 and 4.

**Bayes-optimal test error** – Since we know the law of the dataset $\mathcal{D}$, we can study the *Bayes-optimal (BO) estimator*, which minimizes the test error over all possible estimators. To do this, we use Bayes' theorem to obtain the posterior distribution $\mathbb{P}(\mathbf{W}|\mathcal{D})$ of the weights $\mathbf{W}$ given the dataset:

$$ \mathbb{P}(\mathbf{W}|\mathcal{D}) = \frac{1}{\mathcal{Z}(\mathcal{D})} P_{\mathrm{prior}}(\mathbf{W}) \mathbb{P}(\mathbf{y}|\mathbf{W}, \{\mathbf{x}_i\}_{i=1}^n) $$

where $P_{\mathrm{prior}}(\mathbf{W})$ is a prior distribution on the teacher weights $\mathbf{W}^*$, and the likelihood $\mathbb{P}(\mathbf{y}|\mathbf{W}, \mathbf{X})$ can be seen as a probabilistic channel that generates the labels given the input data $(\mathbf{x}_i)_{i=1}^n$ and the teacher weights $\mathbf{W}^*$, and $\mathcal{Z}(\mathcal{D})$ is a normalization constant. The Bayes-optimal (BO) estimator of the labels for a test sample $\mathbf{x}_{\mathrm{test}}$ not seen in the training set $\mathcal{D}$ then involves the average over the posterior distribution as follows (where $\mathbb{E}_{\mathbf{z}}$ denotes the expectation over $z_1, \cdots, z_k$)

$$ \hat{y}_{\mathcal{D}}^{\mathrm{BO}}(\mathbf{x}_{\mathrm{test}}) := \mathbb{E}\left[ y_{\mathrm{test}} | \mathbf{x}_{\mathrm{test}}, \mathcal{D} \right] = \int \mathbb{E}_{\mathbf{z}}[f_{\mathbf{W}}(\mathbf{x}_{\mathrm{test}})] \, \mathbb{P}(\mathbf{W}|\mathcal{D}) \, \mathrm{d}\mathbf{W} . \tag{3} $$

We will evaluate the BO estimator in terms of its average generalization error, i.e. the mean squared error (MSE) achieved on a new sample. We define it in the following way:

$$ \mathrm{MMSE}_d := \frac{m}{2} \mathbb{E}_{\mathbf{W}^*, \mathcal{D}} \mathbb{E}_{y_{\mathrm{test}}, \mathbf{x}_{\mathrm{test}}} \left[ \left( y_{\mathrm{test}} - \hat{y}_{\mathcal{D}}^{\mathrm{BO}}(\mathbf{x}_{\mathrm{test}}) \right)^2 \right] - \Delta(2 + \Delta) . \tag{4} $$

We denote it $\mathrm{MMSE}_d$, standing for minimum-MSE, as it is the minimum MSE achievable given the setting of the model, and we call $\mathrm{MMSE} := \lim_{d \to \infty} \mathrm{MMSE}_d$.

**Conventions for the MMSE** – Notice the peculiar multiplicative factor $(m/2)$ and the additive term $-\Delta(2 + \Delta)$ in eq. (4). As we detail in Appendix F.1, these factors ensure that $\mathrm{MMSE} \to 1$ for $\alpha \to 0$ (i.e. in the absence of data), and $\mathrm{MMSE} \to 0$ if the posterior concentrates around the true $\mathbf{W}^\star$ (i.e. if $\hat{y}_{\mathcal{D}}^{\mathrm{BO}}(\mathbf{x}) = \mathbb{E}_{\mathbf{z}}[f_{\mathbf{W}^\star}(\mathbf{x})]$). Moreover, as we also detail in Appendix F.1, eq. (4) matches the MMSE of a matrix estimation task to which we will reduce the original problem, see Section 3.

As motivated above, our goal is to analyze the MMSE in the high-dimensional limit, with an extensive-width architecture and quadratically many data samples:

$$ d \to \infty, \quad \alpha := {}^n/d^2 = \Theta(1), \quad \kappa := {}^m/d = \Theta(1), \tag{5} $$

In all that follows, we consider the limit of eq. (5), so that $n, d, m$ all go to infinity together when we write e.g. $\lim_{d \to \infty}$. As we will see, in this limit, the value of the MMSE for a given realization of the randomness concentrates on the averaged value defined in eq. (4).

**Empirical risk minimization estimator** – A more standard way of learning the target function (2) is to minimize the empirical loss $\mathcal{L}$ corresponding to a "student" neural network

$$ \mathcal{L}(\mathbf{W}) = \frac{1}{n} \sum_{i=1}^n \left( y_i - \tilde{f}_{\mathbf{W}}(\mathbf{x}_i) \right)^2, \quad \text{where} \quad \tilde{f}_{\mathbf{W}}(\mathbf{x}) := \frac{1}{m} \sum_{k=1}^m \left[ \frac{1}{\sqrt{d}} (\mathbf{w}_k)^\intercal \mathbf{x} \right]^2 . \tag{6} $$

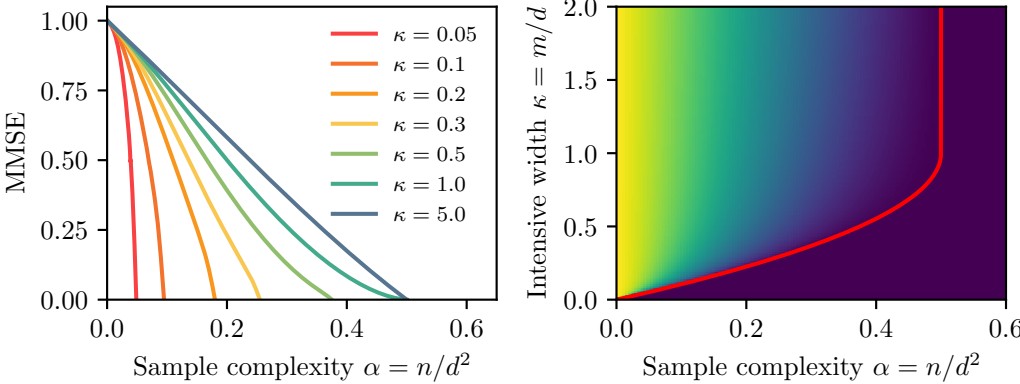

Figure 1: Left: The asymptotic MMSE of eq. (7) for the noiseless ($\Delta = 0$) case, as a function of the sample complexity $\alpha$, for various width ratios $\kappa$. Right: Phase diagram representing the MMSE, brighter color indicates a higher value. The red curve is the perfect recovery transition line $\alpha_{\mathrm{PR}}$, see eq. (1), and its origin is discussed in Section 5.

Note that this does not account for the noise, but activations in neural networks are commonly considered deterministic, so we consider this the most natural choice.

Minimization of the loss over the weights $\mathbf{W} = (\mathbf{w}_k)_{k=1}^m$ is commonly done using gradient descent (GD): one initializes the weights as $\mathbf{W}^{(0)} \sim P_{\mathrm{prior}}$ and then updates them to minimize the empirical loss, for an appropriately choice of learning rate, until convergence. Denoting the weights at convergence as $\hat{\mathbf{W}}(\mathbf{W}^{(0)}, \mathcal{D})$ the estimator for test labels reads $\hat{y}_{\mathbf{W}^{(0)}, \mathcal{D}}^{\mathrm{GD}}(\mathbf{x}_{\mathrm{test}}) := \tilde{f}_{\hat{\mathbf{W}}(\mathbf{W}^{(0)}, \mathcal{D})}(\mathbf{x}_{\mathrm{test}})$. As we will see, it will be interesting to consider also an estimator $\hat{y}^{\mathrm{AGD}}$ obtained by averaging the GD estimator on the labels over the initializations $\mathbf{W}^{(0)}$ of the weights.

## 3   Main results

**Notations –** We use $\mathrm{tr}(\cdot) := (1/d)\mathrm{Tr}[\cdot]$ for the normalized trace. We denote $\mathrm{GOE}(d)$ the distribution of symmetric matrices $\boldsymbol{\xi} \in \mathbb{R}^{d \times d}$ such that $\boldsymbol{\xi}_{ij} \overset{\mathrm{i.i.d.}}{\sim} \mathcal{N}(0, (1 + \delta_{ij})/d)$, for $i \leq j$. For $m = \kappa d$ with $\kappa > 0$, we denote $\mathcal{W}_{m,d}$ the Wishart distribution, and $\mu_{\mathrm{MP}, \kappa}$ the Marchenko-Pastur distribution with ratio $\kappa$. More details on classical definitions and notational conventions are given in Appendix A.

We start by stating the main result of our analysis, applied to the problem of eq. (2).

**Result 1.** *The MMSE of eq. (4) is given in the high-dimensional limit of eq. (5) by:*

$$\mathrm{MMSE} = \frac{2\alpha\kappa}{\hat{q}} - \frac{\kappa\tilde{\Delta}}{2}, \tag{7}$$

*where $\tilde{\Delta} := 2\Delta(2 + \Delta)/\kappa$, and where $\hat{q}$ is a solution of the following equation:*

$$(1 - 2\alpha) + \frac{\tilde{\Delta}\hat{q}}{2} = \frac{4\pi^2}{3\hat{q}} \int \mu_{1/\hat{q}}(y)^3 \mathrm{d}y. \tag{8}$$

*Here, $\mu_t := \mu_{\mathrm{MP}, \kappa} \boxplus \sigma_{\mathrm{s.c.}, \sqrt{t}}$ (for $t \geq 0$) is the free convolution of the Marchenko-Pastur law and a scaled semicircular density, see Appendix A for its precise definition.*

Eq. (8) can be efficiently solved using a numerical scheme, which is detailed in Appendix H.1. We present the results in Fig. 1. In what follows, we detail our approach towards deriving Result 1, which is a consequence of our main theoretical result stated in Claim 2.

**Reduction to a matrix estimation problem –** We first notice that by expanding the square in eq. (2), we can effectively reduce our learning task to an estimation problem in terms of $\mathbf{S}^\star := (1/m)\sum_{k=1}^m \mathbf{w}_k^\star (\mathbf{w}_k^\star)^\intercal$. We give an analytical argument backing this observation in Appendix F.5.

Its conclusion is that, at leading order, the distribution of $y = f_{\mathbf{W}^\star}(\mathbf{x})$ can be reduced to the following form, with $\widetilde{y} := \sqrt{d}(y - 1 - \Delta)$:

$$\widetilde{y} = \operatorname{Tr}[\mathbf{Z}\mathbf{S}^\star] + \sqrt{\widetilde{\Delta}}\xi, \tag{9}$$

with $\xi \sim \mathcal{N}(0, 1)$, $\widetilde{\Delta} := 2\Delta(2 + \Delta)/\kappa$, and where we defined $\mathbf{Z} := (\mathbf{x}\mathbf{x}^{\mathsf{T}} - \mathrm{I}_d)/\sqrt{d}$.

**Generalization error and MMSE on S –** This equivalent problem gives us a way to interpret the convention we chose for eq. (4). Indeed, if we denote $\hat{\mathbf{S}}^{\mathrm{opt}} = \mathbb{E}[\mathbf{S}|\widetilde{\mathbf{y}}, \mathbf{Z}]$ the Bayes-optimal estimator related to the problem of eq. (9), then $\mathrm{MMSE} = \kappa\mathbb{E}\mathrm{tr}[(\mathbf{S}^\star - \hat{\mathbf{S}}^{\mathrm{opt}})^2]$, as proven in Lemma F.1.

**The limit of the MMSE –** We now describe the general form of estimation problems covered by our theoretical analysis, which encompasses the one described in eq. (9) (and thus the original eq. (2)). The goal is to recover the symmetric matrix $\mathbf{S}^\star \in \mathbb{R}^{d \times d}$, which was generated from the Wishart distribution $\mathcal{W}_{m,d}$, from observations $(y_i)_{i=1}^n$, generated as

$$y_i \sim P_{\mathrm{out}}\left(\cdot|\operatorname{Tr}[\mathbf{Z}_i\mathbf{S}^\star]\right), \tag{10}$$

with $\mathbf{Z}_i := (\mathbf{x}_i\mathbf{x}_i^{\mathsf{T}} - \mathrm{I}_d)/\sqrt{d}$. The "channel" $P_{\mathrm{out}}$ accounts for possible non-linearities and noise, encompassing the case of additive Gaussian noise in eq. (9). We define the partition function as:

$$\mathcal{Z}(\{y_i, \mathbf{x}_i\}_{i=1}^n) := \mathbb{E}_{\mathbf{S}\sim\mathcal{W}_{m,d}} \prod_{i=1}^n P_{\mathrm{out}}\left(y_i|\operatorname{Tr}[\mathbf{S}\mathbf{Z}_i]\right). \tag{11}$$

Notice that the averaged logarithm of $\mathcal{Z}$ is (up to an additive constant) equal to the *mutual information* between the observations and the hidden variables: $I(\mathbf{W}^\star; \{y_i\}|\{\mathbf{x}_i\}) = \mathbb{E}\log\mathcal{Z} + n\mathbb{E}\log P_{\mathrm{out}}(y_1|\operatorname{Tr}[\mathbf{Z}_1\mathbf{S}^\star])$. This links $\mathcal{Z}$ to the optimal estimation of $\mathbf{W}$, an important idea behind our study. We are now ready to state our main theoretical result. It gives a sharp characterization of the Bayes-optimal error in any estimation problem of the type of eq. (10). By the reduction described above, it can be directly applied to the original model of eq. (2), and will imply Result 1.

**Claim 2.** *Assume that $m = \kappa d$ with $\kappa > 0$, and $n = \alpha d^2$ with $\alpha > 0$. Let $Q_0 := 1 + \kappa^{-1}$. Then:*

- *The limit of the averaged log-partition function (sometimes called the* free entropy*) is given by*

$$\lim_{d\to\infty} \frac{1}{d^2}\mathbb{E}_{\{y_i,\boldsymbol{x}_i\}}\log\mathcal{Z} = \sup_{q\in[1,Q_0]}\left[I(q) + \alpha\int_{\mathbb{R}\times\mathbb{R}} \mathrm{d}y\mathcal{D}\xi\, J_q(y,\xi)\log J_q(y,\xi)\right], \tag{12}$$

*where*

$$\begin{cases} I(q) & := \inf_{\hat{q}\geq 0}\left[\dfrac{(Q_0 - q)\hat{q}}{4} - \dfrac{1}{2}\Sigma(\mu_{1/\hat{q}}) - \dfrac{1}{4}\log\hat{q} - \dfrac{1}{8}\right], \\[2mm] J_q(y,\xi) & := \displaystyle\int \dfrac{\mathrm{d}z}{\sqrt{4\pi(Q_0 - q)}}\exp\left\{-\dfrac{(z - \sqrt{2q}\xi)^2}{4(Q_0 - q)}\right\} P_{\mathrm{out}}(y|z). \end{cases} \tag{13}$$

*Here, $\Sigma(\mu) := \mathbb{E}_{X,Y\sim\mu}\log|X - Y|$, and, for $t \geq 0$, $\mu_t := \mu_{\mathrm{MP},\kappa} \boxplus \sigma_{\mathrm{s.c.},\sqrt{t}}$ is the free convolution of the Marchenko-Pastur distribution and a (scaled) semicircle law, see Appendix A for its definition.*

- *For any $\alpha > 0$, except possibly in a countable set, the supremum in eq. (12) is reached in a unique $q^\star \in [1, Q_0]$. Moreover, the asymptotic minimum mean-squared error on the estimation of $\mathbf{S}^\star$, achieved by the Bayes-optimal estimator $\hat{\mathbf{S}}^{\mathrm{BO}} := \mathbb{E}[\mathbf{S}|\{y_i, \boldsymbol{x}_i\}]$, is equal to $Q_0 - q^\star$:*

$$\lim_{d\to\infty} \mathbb{E}\mathrm{tr}[(\mathbf{S}^\star - \hat{\mathbf{S}}^{\mathrm{BO}})^2] = Q_0 - q^\star. \tag{14}$$

*It is related to the* MMSE *of eq. (4) by* $\mathrm{MMSE} = \kappa(Q_0 - q^\star)$.

Specifying Claim 2 to the problem of eq. (9), we derive (details are given in Appendix F.7) Result 1, more precisely eqs. (7) and (8).

**Polynomial-time optimal estimation with the GAMP-RIE algorithm –** In Appendix B, we motivate the definition of an algorithm (that we call GAMP-RIE ) to solve the problem of eq. (10). We further argue (based on a combination of theoretical results and numerical observations) that this algorithm is able to reach, in all regions of parameters we investigated, the optimal error described by Claim 2.

**The condition $q \geq 1$ –** Notice that $q^\star = \lim_{d\to\infty}\mathbb{E}[\mathrm{tr}(\mathbf{S}^\star\hat{\mathbf{S}}^{\mathrm{BO}})]$ according to Claim 2. As the MMSE decreases with $\alpha$, it is clear that $q^\star \geq q^\star(\alpha = 0)$. When $\alpha = 0$, we have $\hat{\mathbf{S}}^{\mathrm{BO}} = \mathbb{E}[\mathbf{S}^\star] = \mathrm{I}_d$, and thus $q^\star(\alpha = 0) = 1$. We check in Appendix F.8 that the value $q^\star(\alpha = 0) = 1$ is recovered by eq. (12).

# 4 Derivation of the main results

We derive our main result (Claim 2) in two ways. First, we show how one can show Claim 2 using the *replica method*, a heuristic but exact method (hence the word "claim") which originated in statistical physics [Mézard et al., 1987], and has been used extensively in theoretical physics, as well as in a growing body of work in high-dimensional statistics, theoretical computer science, and theoretical machine learning [Mezard and Montanari, 2009, Zdeborová and Krzakala, 2016, Gabrié, 2020, Charbonneau et al., 2023]. The derivation, that has an interest on its own, is performed in detail in Appendix D and leverages recent progress on the problems of ellipsoid fitting [Maillard and Kunisky, 2024, Maillard and Bandeira, 2023] and extensive-rank matrix denoising [Maillard et al., 2022b, Pourkamali et al., 2024, Semerjian, 2024].

Despite the replica method being conjectured to yield exact results in a large class of high-dimensional models, a rigorous treatment of it remains elusive. It is important, we feel, to present as well a more mathematically sound derivation of our claims, and we thus give an alternative derivation of the Claim 2 using probabilistic techniques amenable to rigorous treatment. In what follows, we present a three-step sketch of a mathematical proof of Claim 2 that combines recent progress performed on the study of a problem known as the ellipsoid fitting conjecture [Maillard and Kunisky, 2024, Maillard and Bandeira, 2023] with the analysis of the fundamental limits of so-called generalized linear models [Barbier et al., 2019], as well as matrix denoising problems [Bun et al., 2016, Maillard et al., 2022b, Pourkamali et al., 2024, Semerjian, 2024]. While a complete mathematical treatment requires more work, we detail the main challenges arising in each of these steps, outlining a fully rigorous establishment of Claim 2.

We denote the *free entropy* $\Phi_d := (1/d^2)\mathbb{E}\log \mathcal{Z}(\{y_i, \mathbf{x}_i\})$, cf. eq. (11). We detail three precise results (two conjectures and a theorem), motivated by recent mathematical works, whose combination would rigorously establish the results of Claim 2. Recall that we consider the high-dimensional limit of eq. (5).

**Step 1: Universality with a "Gaussian equivalent" problem –** The first step of our approach is inspired by recent literature on the *ellipsoid fitting problem* [Maillard and Kunisky, 2024, Maillard and Bandeira, 2023]. It amounts to notice that, if $\mathbf{Z}_i := (\mathbf{x}_i\mathbf{x}_i^T - \mathrm{I}_d)/\sqrt{d}$, by the central limit theorem, for any symmetric matrix $\mathbf{S}$, $\mathrm{Tr}[\mathbf{Z}_i\mathbf{S}]$ is (under mild boundedness conditions on the spectrum of $\mathbf{S}$) approximately distributed as $\mathcal{N}(0, 2\mathrm{tr}[\mathbf{S}^2])$ as $d \to \infty$. A large body of recent literature has established that the free entropy is universal for all data distributions sharing the same asymptotic distribution of their "one-dimensional projections", see e.g. Hu and Lu [2022], Montanari and Saeed [2022], Dandi et al. [2024a], Maillard and Bandeira [2023]. This motivates the conjecture that the free entropy should remain identical (to leading order) if one replaces the matrices $\mathbf{Z}_i$ with $\mathbf{G}_i \sim \mathrm{GOE}(d)$.

**Conjecture 4.1** (Universality). *We define*

$$\Phi_d^{(G)} := \frac{1}{d^2}\mathbb{E}_{(\{y_i', \mathbf{G}_i\})} \log \mathbb{E}_{\mathbf{S}\sim\mathcal{W}_{m,d}} \prod_{i=1}^n P_{\mathrm{out}}\left(y_i' | \mathrm{Tr}[\mathbf{G}_i\mathbf{S}]\right), \tag{15}$$

*where $y_i' \sim P_{\mathrm{out}}(\cdot|\mathrm{Tr}[\mathbf{G}_i\mathbf{S}^\star])$, with $\mathbf{S}^\star \sim \mathcal{W}_{m,d}$ and $\mathbf{G}_i \overset{\mathrm{i.i.d.}}{\sim} \mathrm{GOE}(d)$. Then*

$$\lim_{d\to\infty} |\Phi_d - \Phi_d^{(G)}| = 0.$$

Conjecture 4.1 can be seen as an extension of Corollary 4.10 of Maillard and Bandeira [2023], in the context of a teacher-student model. In particular, we expect it to hold under mild regularity conditions on the channel density $P_{\mathrm{out}}$ (which are satisfied by the Gaussian additive noise we consider).

**Step 2: A matrix generalized linear model with a Wishart prior –** By the first step above, we can focus on $\Phi_d^{(G)}$, and the corresponding estimation problem. A key observation is that one can view this problem as an instance of a *generalized linear model* on $\mathbf{S}^\star$, with a Gaussian data matrix whose $i$-th row is the flattening of the matrix $\mathbf{G}_i$. The limiting free entropy of such models has been worked out in Barbier et al. [2019], when the "ground-truth vector" (here $\mathbf{S}^\star$) has i.i.d. elements. However, here the prior is far from being i.i.d. since $\mathbf{S}^\star \sim \mathcal{W}_{m,d}$. The results of Barbier et al. [2019] generalize naturally to other priors, but such extensions have only been rigorously analyzed in specific settings, e.g. for generative priors rather than i.i.d. [Aubin et al., 2019a, 2020]. In our setting, the structure

of the Wishart prior raises several technical difficulties preventing to directly transpose the proof approaches of Barbier et al. [2019], so we state the following result as a conjecture.

**Conjecture 4.2** (The free entropy of a matrix generalized linear model). *We have*

$$\lim_{d\to\infty} \Phi_d^{(G)} = \sup_{q\in[1,Q_0]} \inf_{\hat{q}\geq 0} \left[ \frac{(Q_0-q)\hat{q}}{4} + \Psi(\hat{q}) + \alpha \int_{\mathbb{R}\times\mathbb{R}} dy \mathcal{D}\xi J_q(y,\xi) \log J_q(y,\xi) \right],$$

*where*

$$\Psi(\hat{q}) := \frac{1}{4} + \lim_{d\to\infty} \frac{1}{d^2} \mathbb{E}_{\boldsymbol{Y}} \log \mathbb{E}_{\boldsymbol{S}\sim\mathcal{W}_{m,d}} \exp\left( -\frac{d}{4} \mathrm{Tr}[(\boldsymbol{Y}-\sqrt{\hat{q}}\boldsymbol{S})^2] \right) \tag{16}$$

*is the asymptotic free entropy of the* matrix denoising *problem* $\boldsymbol{Y} = \sqrt{\hat{q}}\boldsymbol{S}^\star + \boldsymbol{\xi}$, *with* $\boldsymbol{\xi} \sim \mathrm{GOE}(d)$, *and* $\boldsymbol{S}^\star \sim \mathcal{W}_{m,d}$, *and we assume that the* $d \to \infty$ *limit in eq.* (16) *is well-defined.*

**Step 3: Extensive-rank matrix denoising –** As a last step, we study the function $\Psi(\hat{q})$ defined in eq. (16). The optimal estimators and limiting free entropy in matrix denoising have been worked out in Bun et al. [2016], Maillard et al. [2022b], and formally proven (under some assumptions) in Pourkamali et al. [2024], Semerjian [2024].

**Theorem 4.1** (Free entropy of matrix denoising). *For any* $\hat{q} \geq 0$, *the limit in eq.* (16) *is well-defined, and moreover (recall the definition of* $\Sigma(\mu)$ *and* $\mu_t$ *in Claim* 2*)*

$$\Psi(\hat{q}) = -\frac{1}{2}\Sigma(\mu_{1/\hat{q}}) - \frac{1}{4}\log\hat{q} - \frac{1}{8}. \tag{17}$$

We provide a very short and assumption-free proof of Theorem 4.1 in Appendix F.2, which combines a relation between $\Psi(\hat{q})$ and HCIZ integrals of random matrix theory, proven in Pourkamali et al. [2024] (without any assumptions), and fundamental results on the large deviations of the Dyson Brownian motion [Guionnet and Zeitouni, 2002]. As a final remark, we notice that a recent conjecture[2] of Semerjian [2024] states that the free entropy of matrix denoising of $\boldsymbol{S}^\star = (1/m)\sum_{k=1}^m \mathbf{w}_k^\star(\mathbf{w}_k^\star)^\intercal$ remains the same if one considers any i.i.d. prior for $\mathbf{w}_k^\star$, under the matching of its first two moments with the Gaussian and the existence of all other moments. While the validity of this conjecture is subject to debate (see Section VII of Semerjian [2024], and the findings of Camilli and Mézard [2023, 2024]), in the present model it would imply universality of the generalization error given by Claim 2 for any such teacher weight distribution.

**The second part of Claim 2 –** We briefly discuss the second part of Claim 2, concerning the large $d$ limit of $\mathbb{E}\,\mathrm{tr}[(\boldsymbol{S}^\star - \hat{\boldsymbol{S}}^{\mathrm{BO}})^2]$. The fact that the maximizer of eq. (12) is unique for almost all values of $\alpha$ can be seen by simple convexity arguments, see Appendix F.6. The relationship of $q^\star$ with the asymptotic MMSE on the estimation of $\boldsymbol{S}^\star$ is a classical consequence of the I-MMSE theorem in generalized linear models of which eq. (9) is an instance, see e.g. Barbier et al. [2019] and Section D.5 of Maillard et al. [2020].

## 5 Discussion of the main results

**Analysis of the Bayes-optimal estimator –** We start by discussing the noiseless case ($\Delta = 0$), which is described by the phase diagram in Fig. 1. Since there is no noise in the target function, we expect a sharp transition to zero MMSE at a critical sample complexity $\alpha_{\mathrm{PR}}$. We analytically show in Appendix F.3 from eq. (8) that $\alpha_{\mathrm{PR}}$ is given by the expression of eq. (1), and discuss how it is related to a naive counting argument of the "degrees of freedom" of the target function. This transition was known for $\kappa \geq 1$ where the problem is convex, where Gamarnik et al. [2019] shows that there is perfect recovery as soon as $\alpha > 1/2$. For all values of $\kappa$ we see the MMSE is a smooth curve going continuously from 1 at $\alpha = 0$ to 0 at $\alpha_{\mathrm{PR}}$. We derived the slope of the curve at $\alpha_{\mathrm{PR}}$ to be (see Appendix F.4)

$$\frac{\partial \mathrm{MMSE}}{\partial \alpha}\bigg|_{\alpha_{\mathrm{PR}}} = \begin{cases} -2 - \dfrac{4}{\kappa} + \dfrac{12}{1+\kappa} & \text{if } \kappa \leq 1, \\ -2 + \dfrac{2}{\kappa} & \text{if } \kappa \geq 1. \end{cases}$$

---

[2]We mention here the "strong" conjecture of Semerjian [2024]. A weaker form of this conjecture is the universality of the best low-degree polynomial estimator for any i.i.d. prior.

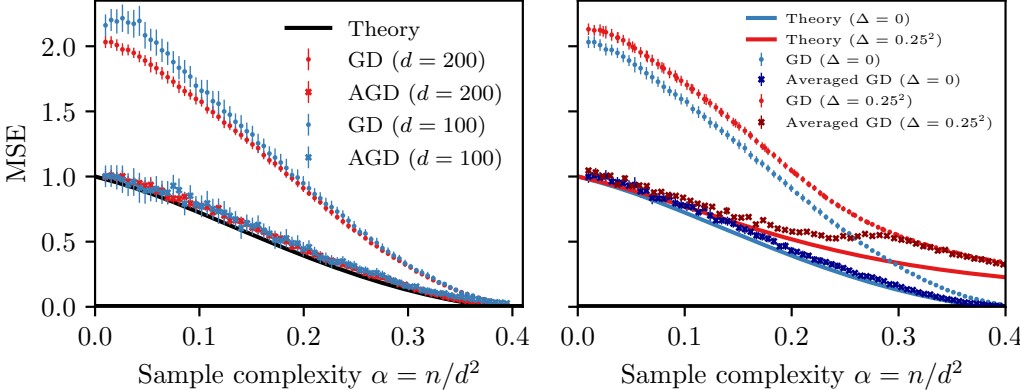

Figure 2: Mean squared error (MSE) as a function of the sample complexity $\alpha$ for $\kappa = 1/2$. Dots are simulations using GD with a single initialization averaged over 32 realizations of the dataset, crosses are averages over 64 initializations with 2 realizations of the dataset. The continuous lines are the asymptotic MMSE given by (7). Left: noiseless $\Delta = 0$ case. The colors indicate the size $d$. We can see how AGD appears to be well described by the theoretical MMSE. We used the learning rates 0.2 for $d = 200$ and 0.07 for $d = 100$. Right: Comparison of GD between the noisy $\sqrt{\Delta} = 0.25$ case (red) and noiseless $\Delta = 0$ case (blue). Adding noise makes AGD worse than the MMSE, and for sample complexity $\alpha \gtrsim 0.3$, all the initializations of GD converge to the same point, making the GD and AGD curves collapse.

It is interesting to observe that the convexity of the curve changes. While we are observing concave dependence on $\alpha$ for small $\kappa$ it becomes convex when $\kappa$ increases and $\alpha$ is close to $\alpha_{\mathrm{PR}}$. We also note that the smooth limit MMSE $\to 1$ as $\alpha \to 0$ supports the result of Cui et al. [2023] about a quadratic number of samples being needed to learn better than linear regression.

We also evaluated the MMSE in the presence of noise, where we observed it to decrease smoothly as $\alpha$ increases with no particular phase transition. We show an example of the theoretical prediction for the MMSE in this case in Fig. 2 right. As expected, in the presence of noise, it decreases monotonically and smoothly, and goes to zero as $\alpha \to \infty$.

We considered analytically the limits $\kappa \to 0$ and $\kappa \to \infty$, i.e. the limits of small and large (but still extensive in $d$) hidden layer. The analysis of these limits are detailed in Appendix E.

Further, in Appendix B.3 we compare the asymptotic theoretical result for the Bayes-optimal error with the performance of the GAMP-RIE algorithm on finite-size instances. In all the cases we evaluated, we observed that GAMP-RIE reached the Bayes-optimal error characterized by Claim 2.

Finally, while we assumed in eq. (2) that the second layer weights are fixed and equal to 1, in Appendix G we generalize all our main results, theoretical and algorithmic, to learnable second layer weights.

**Comparison to the ERM estimator obtained by gradient descent –** The results discussed so far concern the Bayes-optimal MMSE, which requires evaluating the marginals of the posterior distribution. We now investigate numerically the performance of empirical risk minimization via gradient descent, which is the standard method of machine learning. It would be typical to expect a gradient based approach to be suboptimal, as the problem is non-convex for $\kappa < 1$. In Fig. 2, we compare (a) the MSE $\kappa\mathrm{tr}[(\mathbf{S}^\star - \hat{\mathbf{S}}_{\mathrm{GD}})^2]$ reached by gradient descent (GD) minimizing the loss (6) from random initialization, (b) the MSE reached by GD averaged over initializations, and (c) the MMSE derived from the theory.

In the noiseless case, $\Delta = 0$, we very remarkably observe that the MSE reached by gradient descent is very close to exactly twice larger than the asymptotic MMSE. Such a relation is known in high-dimensional generalized linear regression to hold between the Gibbs estimator, where test error is evaluated for weights that are sampled uniformly from the posterior, and the Bayes-optimal estimator that averages over the weights sampled from the posterior [Engel, 2001, Barbier et al., 2019]. In

general, there is no reason why the randomly initialized gradient descent should be able to sample the posterior measure. We nevertheless evaluate the average over the initialization of gradient descent and observe that, indeed, the MSE reached this way is consistent with the MMSE. This leads us to conjecture that in the noiseless one-hidden layer neural network with quadratic activation and a target function matching this architecture, randomly-initialized gradient descent samples the posterior despite the problem being non-convex, and hence its average achieves the MMSE.

Let us offer a heuristic argument for this perhaps intriguing phenomenon. It starts with the equivalent of the representer theorem: one can write $\mathbf{S}$ in the span of $\{\mathbf{x}_i\mathbf{x}_i^T\}_{i=1}^n$, plus a matrix in the orthogonal space, that is $\mathbf{S} = \sum_{i=1}^n \beta_i\mathbf{x}_i\mathbf{x}_i^T + \mathbf{Z}$. This means that gradient descent reaches one solution of the minimization with one additional spurious component. The Bayes optimal procedure would be to set this spurious reminder to zero since the data are not informative in this direction. It is reasonable (although non-trivial) to assume that this is what is achieved by averaging over initialization.

When comparing the MMSE to the performance of GD in the noisy setting, we observe a gap between the MMSE and the performance of gradient descent, even averaged over initialization or regularized (as shown in Appendix H.3, Figure 5 left). In particular, for the noisy case, we see that for small sample complexity, the averaged GD is close to matching the MMSE, but as the number of available samples increases, the error of the averaged and non-averaged versions of GD coincide. This is a sign of the trivialization of the landscape, in the sense that GD converges to the same function independently of the initialization: it can be quantified using the variances of the function reached by GD. This is investigated further in Appendix H.3, together with the effect of $\ell_2$ regularization. We can characterize empirically another phase transition: for a sample complexity larger than $\alpha_T(\Delta)$, GD converges to the same function independently of the initialization. In the noiseless $\Delta = 0$ case, this is simply the perfect recovery transition, and $\alpha_{\mathrm{PR}} = \alpha_T(\Delta = 0)$, while increasing the noise intensity makes the threshold lower until it reaches a plateau, which for $\kappa = 0.5$ is at $\alpha_T(\Delta \to \infty) \approx 0.2$. We display this numerical finding in Figure 5 (right) in Appendix H.3. A tight analytical study of the landscape-trivialization threshold $\alpha_T(\Delta)$ as a function of the noise variance $\Delta$ is left for future work.

## 6  Conclusion and limitations

In this work, we provide an explicit formula for the generalization MMSE when learning a target function in the form of a one-hidden layer neural network with quadratic activation in the limit of large dimensions, extensive width and a quadratic number of samples. The techniques deployed to obtain this result are novel and, we believe, of independent interest. There are many natural extensions of the present works. While we presented, additionally to the replica derivation, a mathematically sound derivation, a fully rigorous treatment, a technical and lengthy task, is left for an extended version of this work. We analyzed the Bayes-optimal MMSE, presented the GAMP-RIE algorithm that is able to reach it in polynomial time, and compared it to the performance of gradient descent numerically. We leave for future work the theoretical analysis of the properties of gradient descent that we discovered numerically. Of particular interest is the role played by the implicit nuclear norm regularization when starting from small initialization, as discussed for the matrix sensing problem e.g. in Gunasekar et al. [2017], Li et al. [2020], Stöger and Soltanolkotabi [2021]. Finally, we also presented the natural extension of our results and techniques to the case of a learnable second layer.

The main limitations of our setting are its restriction to Gaussian input data, random i.i.d. weights of the target/teacher neural network, quadratic activation, and a single hidden layer. Going beyond any of these limitations would be a compelling direction of research, in particular for more generic activation such as the ReLU or sigmoid function (we sketch this extension in Appendix C) and multiple layers, and we hope our work will spark interest in these directions.

### Acknowledgements

We want to thank Giulio Biroli, Francis Bach, Guilhem Semerjian, Pierfrancesco Urbani, Vittorio Erba, Jason Lee and Afonso Bandeira for insightful discussions about this work. This work was supported by the Swiss National Science Foundation under grants SNSF SMArtNet (grant number 212049) and SNSF OperaGOST (grant number 200390).

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

# A   Additional definitions and conventions

**Convention** – Throughout this manuscript, we use $\mathbb{E}_X$ to denote the expectation solely over the random variable $X$. We denote $\mathcal{M}_1^+(\mathbb{R})$ the set of real probability distributions.

**Random matrix ensembles** – For any $d \geq 1$, we define two standard random matrix distributions over the space of symmetric $d \times d$ real matrices:

- A matrix $\boldsymbol{\xi}$ is distributed according to the $\mathrm{GOE}(d)$ distribution (standing for *Gaussian Orthogonal Ensemble*) if $\xi_{ij} \overset{\text{i.i.d.}}{\sim} \mathcal{N}(0, [1 + \delta_{ij}]/d)$ for any $1 \leq i \leq j \leq d$.
- For any $m \geq 1$, a matrix $\mathbf{S}$ is distributed according to the Wishart distribution $\mathcal{W}_{m,d}$ if $\mathbf{S} = \mathbf{W}^\intercal \mathbf{W}/m$, where $\mathbf{W} \in \mathbb{R}^{m \times d}$ with $W_{ki} \overset{\text{i.i.d.}}{\sim} \mathcal{N}(0,1)$ for $k \in [m], i \in [d]$.

For a symmetric matrix $\mathbf{M}$ with eigenvalues $(\lambda_i)_{i=1}^d$, we denote $\mu_{\mathbf{M}} := (1/d) \sum_{i=1}^d \delta_{\lambda_i}$ its empirical eigenvalue distribution (ESD). It is well known that for $d \to \infty$ the ESD of $\mathrm{GOE}(d)$ and $\mathcal{W}_{m,d}$ matrices converge to (respectively) the Wigner semicircle and the Marchenko-Pastur density.

**Theorem A.1.** *[Wigner [1955], Marchenko and Pastur [1967]] Let $m = \kappa d$ for $\kappa > 0$, and let $\boldsymbol{\xi} \sim \mathrm{GOE}(d)$ and $\boldsymbol{S} \sim \mathcal{W}_{m,d}$. Then, as $d \to \infty$, the ESDs of $\boldsymbol{\xi}$ and $\boldsymbol{S}$ almost surely converge (in the sense of weak convergence) to the following probability distributions (respectively).*

- *The semicircle law, with density*

$$\sigma_{\text{s.c.}}(x) = \frac{\sqrt{4 - x^2}}{\sqrt{2\pi}} \mathbb{1}\{|x| \leq 2\}. \tag{18}$$

   *We denote $\sigma_{\text{s.c.}, \sqrt{t}}(x) := t^{-1/2} \sigma_{\text{s.c.}}(x/\sqrt{t})$ the scaled semicircle law with variance $t$.*

- *The Marchenko-Pastur law, with density*

$$\mu_{\text{MP}, \kappa}(x) = \begin{cases} (1 - \kappa)\delta(x) + \dfrac{\kappa \sqrt{(\lambda_+ - x)(x - \lambda_-)}}{2\pi x} & \text{if } \kappa \leq 1, \\[3mm] \dfrac{\kappa \sqrt{(\lambda_+ - x)(x - \lambda_-)}}{2\pi x} & \text{if } \kappa \geq 1. \end{cases} \tag{19}$$

   *Here $\lambda_\pm := (1 \pm \kappa^{-1/2})^2$.*

**Transforms of probability distributions** – For any real probability measure $\mu$, we define its *Stieltjes transform* $g_\mu(z) := \mathbb{E}_\mu[1/(X - z)]$ for $z \in \mathbb{C}$. If $\mathbb{C}_+ := \{z \in \mathbb{C} : \mathrm{Im}(z) > 0\}$, then $g_\mu(z) \in \mathbb{C}_+$ for all $z \in \mathbb{C}_+$. Moreover, we have the Stieltjes-Perron inversion formula:

**Theorem A.2** (Stieltjes-Perron inversion formula). *For all $a < b$, we have*

$$\mu((a,b)) = \lim_{\delta \downarrow 0} \lim_{\epsilon \downarrow 0} \frac{1}{2i\pi} \int_{a+\delta}^{b-\delta} [g_\mu(x + i\epsilon) - g_\mu(x - i\epsilon)]\mathrm{d}x.$$

*In particular, if $\mu$ has a continuous density with respect to the Lebesgue measure then:*

$$\forall x \in \mathbb{R}, \quad \frac{\mathrm{d}\mu}{\mathrm{d}x} = \lim_{\epsilon \downarrow 0} \frac{1}{\pi} \mathrm{Im}\, g_\mu(x + i\epsilon).$$

We often use the logarithmic potential function $\Sigma(\mu) := \int \mu(\mathrm{d}x)\mu(\mathrm{d}y) \log|x - y|$. We further define the $\mathcal{R}$-transform of $\mu$ as:

$$\mathcal{R}_\mu(s) := g_\mu^{-1}(-s) - \frac{1}{s}. \tag{20}$$

We refer to Tulino and Verdú [2004] for more details on the definitions of this transform, e.g. concerning its complete domain of definition. Informally, the $\mathcal{R}$ transform is well-defined in a neighborhood of $0$ for all measures which have bounded support. In particular, we have for the semicircle and the Marchenko-Pastur distributions:

$$\begin{cases} \mathcal{R}_{\sigma_{\text{s.c.}, \sqrt{t}}}(s) & = ts, \\[2mm] \mathcal{R}_{\mu_{\text{MP}, \kappa}}(s) & = \dfrac{\kappa}{\kappa - s}. \end{cases} \tag{21}$$

---
**Algorithm 1:** GAMP-RIE
---

**Result:** The estimator $\hat{\mathbf{S}}$

**Input:** Observations $\mathbf{y} \in \mathbb{R}^n$ and "sensing vectors" $\mathbf{Z}_i := (\mathbf{x}_i \mathbf{x}_i^\mathsf{T} - \mathrm{I}_d)/\sqrt{d} \in \mathbb{R}^{d \times d}$;

*Initialize* $\hat{\mathbf{S}}^0 \sim \mathcal{W}_{m,d}$ and $\hat{\mathbf{c}}, \boldsymbol{\omega}, \mathbf{V}$ randomly;

**while** *not converging* **do**

  • *Estimation of the variance and mean of* $\mathrm{Tr}[\mathbf{Z}_i \hat{\mathbf{S}}]$;

  $V^t = \hat{c}^t$ and $\omega_i^t = \mathrm{Tr}[\mathbf{Z}_i \hat{\mathbf{S}}^t] - g_{\mathrm{out}}(y_i, \omega_i^{t-1}, V^{t-1})V^t$ ;

  • *Variance and mean of* $\mathbf{S}$ *estimated from the "channel" observations*;

  $A^t = \dfrac{2\alpha}{n} \sum\limits_{i=1}^{n} g_{\mathrm{out}}(y_i, \omega_i^t, V^t)^2$ and $\mathbf{R}^t = \hat{\mathbf{S}}^t + \dfrac{1}{dA^t} \sum\limits_{i=1}^{n} g_{\mathrm{out}}(y_i, \omega_i^t, V^t)\mathbf{Z}_i$ ;

  • *Update of the estimation of* $\mathbf{S}^\star$ *with the "prior" information*;

  $\hat{\mathbf{S}}^{t+1} = f_{\mathrm{RIE}}\left(\mathbf{R}^t, \dfrac{1}{2A^t}\right)$ and $\hat{c}^{t+1} = 2F_{\mathrm{RIE}}\left(\dfrac{1}{2A^t}\right)$;

  $t = t + 1$;

**end**
---

**Free additive convolution –** The main interest of the $\mathcal{R}$-transform lies in its connection to the (additive) *free convolution* of measures. Informally, we can interpret the free convolution $\mu \boxplus \nu$ of two measures $\mu$ and $\nu$ as the limiting spectral measure of $\mathbf{A} + \mathbf{B}$, where $\mathbf{A}$ and $\mathbf{B}$ are symmetric $d \times d$ random matrices, with limiting spectral distributions $\mu$ and $\nu$, and which are *asymptotically free*. While we refer to Anderson et al. [2010], Tulino and Verdú [2004] for mathematical discussions of asymptotic freeness, we recall that in particular if $\mathbf{B}$ is a $\mathrm{GOE}(d)$ matrix independent of $\mathbf{A}$, then $\mathbf{A}$ and $\mathbf{B}$ are asymptotically free. Crucially, the $\mathcal{R}$ transform is additive under free convolution (see Theorem 2.64 in Tulino and Verdú [2004] e.g.):

$$\mathcal{R}_{\mu \boxplus \nu}(s) = \mathcal{R}_\mu(s) + \mathcal{R}_\nu(s). \tag{22}$$

Eq. (22) allows to efficiently compute the density of $\mu \boxplus \nu$ given the ones of $\mu$ and $\nu$, by relating the $\mathcal{R}$ transform to the Stieltjes transform, and then using the Stieltjes-Perron inversion theorem (Theorem A.2).

## B   The GAMP-RIE algorithm

### B.1   Polynomial-time optimal estimation with the GAMP-RIE algorithm

Let us recall a crucial observation of Section 3: the learning problem of eq. (2) can be effectively reduced to a *generalized linear model* (GLM) on the matrix $\mathbf{S}^\star$ (cf. eq. (10)):

$$y_i \sim P_{\mathrm{out}}(\cdot | \mathrm{Tr}[\mathbf{Z}_i \mathbf{S}^\star]), \tag{23}$$

with $\mathbf{Z}_i := (\mathbf{x}_i \mathbf{x}_i^\mathsf{T} - \mathrm{I}_d)/\sqrt{d}$, $\mathbf{S}^\star \sim \mathcal{W}_{m,d}$, and $P_{\mathrm{out}}$ a noise channel (which would be Gaussian in eq. (9)). An important difficulty in analyzing eq. (23) is the rather complex structure of the matrices $\mathbf{Z}_i$ (which can be viewed as "sensing vectors" applied to $\mathbf{S}^\star$). Determining the optimal algorithm in GLMs when the sensing vectors have arbitrary structure is in general open. Anticipating on a universality argument for the MMSE (cf. Section 4), we "forget" momentarily about the structure of $\{\mathbf{Z}_i\}$, and assume that the optimal algorithm takes the form it would have if the $\{\mathbf{Z}_i\}$ were instead Gaussian matrices (i.e. $\mathrm{GOE}(d)$). For generalized linear models with Gaussian sensing vectors, a class of *generalized approximate message-passing* (GAMP) algorithms have been extensively studied, and argued to reach optimal performance in the absence of a computational-to-statistical gap [Donoho et al., 2009, Rangan, 2011, Zdeborová and Krzakala, 2016]. The GAMP algorithm includes a denoiser that is adjusted to the prior information about the signal $\mathbf{S}^\star$, that is in our case a Wishart distribution. Combining these two facts, we propose the GAMP-RIE algorithm in Algorithm 1. An implementation of GAMP-RIE is accessible in the GitHub repository associated to this work [Maillard et al., 2024].

The functions $g_{\text{out}}$, $f_{\text{RIE}}$ and $F_{\text{RIE}}$ appearing in Algorithm 1 are defined as follows. First, we let

$$g_{\text{out}}(y, \omega, V) := \frac{1}{V} \frac{\int \mathrm{d}z \, (z - \omega) \, e^{-\frac{(z-\omega)^2}{2V}} \, P_{\text{out}}(y|z)}{\int \mathrm{d}z \, e^{-\frac{(z-\omega)^2}{2V}} \, P_{\text{out}}(y|z)}. \tag{24}$$

In particular, for the problem of eq. (9), we have

$$g_{\text{out}}(y, \omega, V) = \frac{y - \omega}{\widetilde{\Delta} + V}.$$

The two functions $(f_{\text{RIE}}, F_{\text{RIE}})$ are related to the problem of *matrix denoising*, in which one aims at recovering a matrix $\mathbf{S}_0 \sim \mathcal{W}_{m,d}$ from the observation of $\mathbf{R} = \mathbf{S}_0 + \sqrt{\Delta}\boldsymbol{\xi}$, with $\boldsymbol{\xi} \sim \text{GOE}(d)$. We recall some important results on this problem, and how they relate to the definition of the functions $(f_{\text{RIE}}, F_{\text{RIE}})$.

(i) The optimal estimator (in the sense of mean squared error) of $\mathbf{S}_0$ has been worked out in Bun et al. [2016], and belongs to the class of *rotationally-invariant estimators* (RIE). $f_{\text{RIE}}(\mathbf{R}, \Delta)$ is this optimal estimator, and it admits the following explicit form. If $\mathbf{R} = \mathbf{U}\boldsymbol{\Lambda}\mathbf{U}^\mathsf{T}$ is the spectral decomposition of $\mathbf{R}$, and letting $\rho_\Delta := \mu_{\text{MP},\kappa} \boxplus \sigma_{\text{s.c.},\sqrt{\Delta}}$ be its asymptotic eigenvalue distribution (see Appendix A for the definition of the free convolution $\mu \boxplus \nu$ and its relation to the sum of asymptotically free matrices), then $f_{\text{RIE}}(\mathbf{R}, \Delta) = \mathbf{U}f_\Delta(\boldsymbol{\Lambda})\mathbf{U}^\mathsf{T}$, where $f_\Delta(\lambda) = \lambda - 2\Delta h_\Delta(\lambda)$, with $h_\Delta$ the Hilbert transform of $\rho_\Delta$. More precisely:

$$h_\Delta(\lambda) := \text{P.V.} \int \frac{1}{\lambda - t} \rho_\Delta(t) \mathrm{d}t.$$

$\rho_\Delta$ and $h_\Delta$ can be evaluated numerically very efficiently, see Appendix A for details.

(ii) $F_{\text{RIE}}(\Delta)$ is defined as the asymptotic MMSE of the same matrix denoising problem. It can be written in the two equivalent forms (see Maillard et al. [2022b], Pourkamali et al. [2024], Semerjian [2024]):

$$F_{\text{RIE}}(\Delta) = \Delta - \frac{4\pi^2}{3}\Delta^2 \int \mathrm{d}\lambda \, \rho_\Delta(\lambda)^3 = \Delta - 4\Delta^2 \int \mathrm{d}\lambda \, \rho_\Delta(\lambda) h_\Delta(\lambda)^2. \tag{25}$$

In Appendix B.2 we sketch the derivation of the *state evolution* of Algorithm 1, assuming a universality result discussed in Section 4 holds as well for GAMP-RIE. Concretely, we show that one can analytically track the performance of its iterates in the high-dimensional limit, and we draw a formal connection with the information-theoretic predictions of Claim 2. Notably, we obtain a so-called state-evolution of the GAMP-RIE algorithm (which turns out to follow from rigorous work on non-separable estimation with GAMP [Berthier et al., 2020, Gerbelot and Berthier, 2023]), and show that its fixed points agree with the fixed point equations that provide the Bayes-optimal error. In all regions of parameters that we investigated below we observed a unique fixed point, meaning that the GAMP-RIE algorithm asymptotically reaches the Bayes-optimal performance, see Appendix B.3

## B.2 State evolution: a connection between Algorithm 1 and Result 1

We briefly sketch here the statistical-physics style derivation of the so-called *state evolution* of Algorithm 1: this will draw a connection between the performance of the Bayes-optimal estimator, characterized by Result 1, and the estimator of Algorithm 1. We define $q^t := \text{tr}[(\hat{\mathbf{S}}^t)^2]$, and $m^t := \text{tr}[\hat{\mathbf{S}}^t \mathbf{S}^\star]$. Thanks to Bayes-optimality, one can show that, along the GAMP-RIE trajectory, the so-called Nishimori identities are preserved (see Zdeborová and Krzakala [2016] for more details), so that we have, at leading order as $d \to \infty$, that $q^t = m^t$.

Up to some critical differences, we can transpose the derivation of Zdeborová and Krzakala [2016] of the state evolution of GAMP for generalized linear models with Gaussian sensing vectors, and i.i.d. priors, to our GAMP-RIE algorithm. The differences with our setting are twofold:

(i) The "sensing vectors" $\mathbf{Z}_i$ are not Gaussian. We conjecture that the universality arguments discussed in Section 4 extend to the analysis of the GAMP-RIE algorithm. This allows us to replace $\mathbf{Z}_i$ by $\mathbf{G}_i \overset{\text{i.i.d.}}{\sim} \text{GOE}(d)$ when evaluating $(q^t, m^t)$ (i.e. when studying the high-dimensional performance of Algorithm 1). We are then able to make a direct use of some results of Zdeborová and Krzakala [2016].

($ii$) The prior over $\mathbf{S}^\star$ is not i.i.d.: as we saw, this led to a non-trivial "denoising" part in Algorithm 1. The performance of this denoising procedure in the high-dimensional limit can however be characterized precisely, as the function $F_{\mathrm{RIE}}$ admits a closed-form expression.

We now briefly expose the derivation, transposed to our setting under the universality assumption above. By definition of $q^t$, we have $\hat{c}^t = 2(Q_0 - q^t)$. If $\omega, z$ are centered and jointly Gaussian variables with $\mathbb{E}[\omega^2] = 2q^t$, $\mathbb{E}[z^2] = 2Q_0$, and $\mathbb{E}[\omega z] = 2m^t = 2q^t$, and $y \sim P_{\mathrm{out}}(\cdot|z)$, we define

$$\hat{q}^t := 4\alpha \, \mathbb{E}_{y,w}[g_{\mathrm{out}}(y, \omega, V^t)^2], \tag{26}$$

so that $A^t = \hat{q}^t/2$ in the $n, d \to \infty$ limit. For the "channel" part of the GAMP-RIE algorithm, the standard analysis for generalized linear model, alongside the universality phenomenon discussed above (which allows replacing $\mathrm{Tr}[\mathbf{Z}_i \hat{\mathbf{S}}^t]$ by $\mathrm{Tr}[\mathbf{G}_i \hat{\mathbf{S}}^t]$ in the update of $\omega_i^t$) shows that $\hat{q}^t$ satisfies the equation:

$$\hat{q}^t = 4\alpha \frac{\partial}{\partial q} \left[ \int \mathrm{d}y \mathcal{D}\xi \, J_q(y, \xi) \log J_q(y, \xi) \right]_{q=q^t}, \tag{27}$$

where $J_q(y, \xi)$ is defined in eq. (13). Eq. (27) is the very same as for the standard GAMP for generalized linear models [Zdeborová and Krzakala, 2016].

We have, however, a more structured prior. After replacing $\mathbf{Z}_i$ by Gaussian matrices $\mathbf{G}_i$ in Algorithm 1, the argument is that at leading order as $d \to \infty$ one has:

$$\mathbf{R}^t \overset{\mathrm{d}}{=} \mathbf{S}^\star + \frac{1}{\sqrt{\hat{q}^t}} \boldsymbol{\xi}, \tag{28}$$

with $\boldsymbol{\xi} \sim \mathrm{GOE}(d)$. Heuristic details on how to derive eq. (28) can be found again in Zdeborová and Krzakala [2016], see Section 6.4.1[3] there. By definition of $F_{\mathrm{RIE}}$, this implies $q^{t+1} := \mathrm{tr}[(\hat{\mathbf{S}}^{t+1})^2] = Q_0 - F_{\mathrm{RIE}}((\hat{q}^t)^{-1})$, so that by eq. (25):

$$Q_0 - q^{t+1} = \frac{1}{\hat{q}^t} - \frac{4\pi^2}{3(\hat{q}^t)^2} \int \mathrm{d}\lambda \mu_{1/\hat{q}^t}(\lambda)^3, \tag{29}$$

with $\mu_t := \mu_{\mathrm{MP},\kappa} \boxplus \sigma_{\mathrm{s.c.},\sqrt{t}}$. Notice that remarkably, eqs. (27) and (29) precisely match the extremization equations of the asymptotic free entropy, as given in Claim 2 and Result 1, exactly as for "usual" generalized linear models with i.i.d. priors [Rangan, 2011, Javanmard and Montanari, 2013, Zdeborová and Krzakala, 2016].

**Mathematical consequences** – The fact that, assuming the universality property above, our GAMP-RIE algorithm can be seen as the usual GAMP algorithm in a generalized linear model with a *non-separable prior* has a very interesting consequence. Indeed, the latter model admits a rigorous state evolution thanks to the analysis of Berthier et al. [2020], Gerbelot and Berthier [2023]. To make this point clearer, we notice that (after replacing $\mathbf{Z}_i$ by $\mathrm{GOE}(d)$ matrices $\mathbf{G}_i$) Algorithm 1 can be written in the following form:

$$\begin{cases} \boldsymbol{\omega}^t & = \mathbf{G}\hat{\mathbf{v}}(\mathbf{u}^t, \Sigma_t) - V^t g_{\mathrm{out}}(\mathbf{y}, \boldsymbol{\omega}^{t-1}, V^{t-1}), \\ \mathbf{u}^t & = \frac{1}{d}\mathbf{G}^\intercal g_{\mathrm{out}}(\mathbf{y}, \boldsymbol{\omega}^t, V^t) + \Sigma_t^{-1}\hat{\mathbf{v}}(\mathbf{u}^t, \Sigma_t). \end{cases} \tag{30}$$

Let us clarify some notations used in eq. (30):

- $\mathbf{u}^t \in \mathbb{R}^p$, with $p := \binom{d+1}{2}$, can be seen as the flattening of the symmetric matrix $A^t \mathbf{R}^t$ of Algorithm 1 via the following canonical mapping. For $\mathbf{S} \in \mathcal{S}_d$, we define $\mathrm{vec}(\mathbf{S}) \in \mathbb{R}^p$ by $\mathrm{vec}(\mathbf{S})_{ii} = S_{ii}$ and $\mathrm{vec}(\mathbf{S})_{ij} = \sqrt{2}S_{ij}$ for $i < j$. This flattening is an isometry: $\langle \mathrm{vec}(\mathbf{S}), \mathrm{vec}(\mathbf{R}) \rangle = \mathrm{Tr}[\mathbf{SR}]$. We have $\mathbf{u}^t = \mathrm{vec}(A^t \mathbf{R}^t)$.
- $\mathbf{G} \in \mathbb{R}^{n \times p}$ is a Gaussian i.i.d. matrix, whose elements have variance $2/d$.
- $\Sigma_t^{-1} := -2\alpha\mathbb{E}\mathrm{div}[g_{\mathrm{out}}(\mathbf{t}, \boldsymbol{\omega}^t, V^t)]$ is related to $A^t$ by $A^t = \Sigma_t^{-1}$.
- $V^t := 2F_{\mathrm{RIE}}(\Sigma_t/2)$.

---

[3]Section VI.D.1 in the arXiv version of Zdeborová and Krzakala [2016].

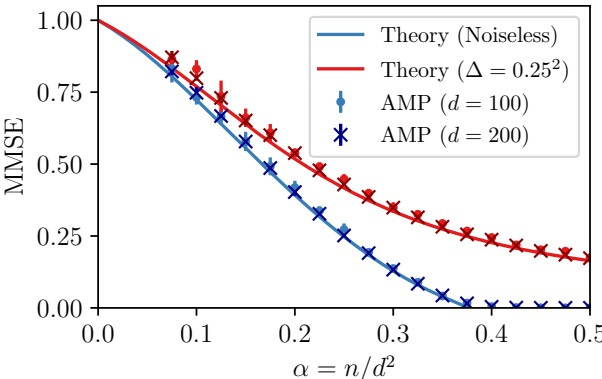

Figure 3: Comparison of the performance of GAMP-RIE with the asymptotic MMSE (7) both in the noiseless ($\Delta = 0$) and in a noisy ($\sqrt{\Delta} = 0.25$) case, with $\kappa = 0.5$. Each dot is the average over 8 runs of GAMP-RIE at a moderate size of either $d = 100$ (circle dots) or $d = 200$ (crosses). The error bars are the standard deviations of the MSE.

- $\hat{\mathbf{v}}^t(\mathbf{u}^t, \Sigma_t) \in \mathbb{R}^p$ is the flattening of the RIE denoiser of Algorithm 1, i.e. if we denote $\mathbf{R}^t/\Sigma_t$ the matrix such that $\mathbf{u}^t = \text{vec}(\mathbf{R}^t/\Sigma_t)$:

$$\hat{\mathbf{v}}^t(\mathbf{u}^t, \Sigma^t) \coloneqq \text{vec}[f_{\text{RIE}}(\mathbf{R}^t, \Sigma_t/2)].$$

Eq. (30) is the canonical form of the GAMP algorithm, as written e.g. in Berthier et al. [2020], Gerbelot and Berthier [2023]. In particular, we can leverage their results to write:

**Theorem B.1** (State Evolution (informal) Berthier et al. [2020], Gerbelot and Berthier [2023]). *Denote $q_{\text{AMP}}^t \coloneqq \text{tr}[\hat{\mathbf{S}}_t \mathbf{S}^\star]$ and $\hat{q}_{\text{AMP}}^t \coloneqq \frac{4\alpha}{n} \sum_{i=1}^n g_{\text{out}}(y_i, \omega_i^t, V^t)^2$ (recall the definition of these quantities in Algorithm 1). Assume that the "sensing matrices" $(\mathbf{Z}_i)_{i=1}^n$ in Algorithm 1 are replaced by $(\mathbf{G}_i)_{i=1}^n$, which are i.i.d. GOE($d$) matrices. Then for any $t \geq 0$, $q_{\text{AMP}}^t$ and $\hat{q}_{\text{AMP}}^t$ follow the state evolution equations (27) and (29) asymptotically as $d, n \to \infty$.*

Beyond the rigorous control of the GAMP-RIE algorithm, Theorem B.1 has an additional mathematical consequence: it allows to leverage a set of mathematical techniques that use AMP algorithms to prove results on the asymptotic MMSE and on the mutual information, as Theorem B.1 implies that they can be used verbatim in our setting. More precisely, the fact that the GAMP-RIE algorithm achieves an MSE with value given by Claim 2 immediately yields that the latter is, at least, an upper bound on the asymptotic MMSE (when assuming Gaussian GOE($d$) "sensing vectors" $\mathbf{G}_i$). Additionally, the application of the I-MMSE theorem [Guo et al., 2005] shows that our claimed free entropy (i.e. the limit of $(1/d^2)\mathbb{E} \log \mathcal{Z}$ in Claim 2) is a lower bound on the real one (see e.g. section 2.C in Barbier et al. [2016]).

### B.3  GAMP-RIE algorithm reaching the optimal error

In Fig. 3 we compare the asymptotic theoretical result for the Bayes-optimal error with the performance of the GAMP-RIE algorithm for $d = 100$ and $d = 200$, in both the noiseless (blue) and noisy (red) cases. We observe that even for such moderate sizes the agreement between the algorithmic performance and the theory is excellent.

We also stress that in all the cases we evaluated, the state evolution of the GAMP-RIE converges to the fixed point that corresponds to the Bayes-optimal performance. This means that the Bayes-optimal error discussed above is reachable efficiently with the GAMP-RIE algorithm. In particular, unlike in the canonical phase retrieval problems (i.e. when $m = 1$) [Barbier et al., 2019], we did not identify a computational-to-statistical gap when learning this extensive-width quadratic-activation neural network.

## C   Generic activations

We give here a brief discussion on the extension of our work to a more generic activation function. While our derivation (cf. Section 4) heavily relies on the non-linearity being quadratic, a first natural extension would be to consider *polynomial* activations, with an output generated as (assuming a noiseless setting):

$$y_i = \frac{1}{m} \sum_{k=1}^{m} \left( \frac{(\mathbf{w}_k^\star)^\mathsf{T} \mathbf{x}_i}{\sqrt{d}} \right)^p,$$

for some integer $p \geq 3$. One could also "linearize" this model, by writing it as $y_i = \langle T^\star, X_i \rangle$, in which $T^\star, X_i$ are now $p$-tensors, defined as

$$\begin{cases} T^\star & := \dfrac{1}{m} \sum_{k=1}^{m} (\mathbf{w}_k^\star)^{\otimes p}, \\ X_i & := \dfrac{1}{d^{p/2}} \mathbf{x}_i^{\otimes p}. \end{cases}$$

However, two main challenges arise when carrying out the program of Section 4 in this "tensor" model:

($i$)   First, determining whether the universality Conjecture 4.1 holds for these models (and if yes, in which scaling of the number of samples $n$ with $d$) is a challenging open question that falls outside the scope of our results as well as of previous works on free entropy universality [Hu and Lu, 2022, Montanari and Saeed, 2022, Dandi et al., 2024a, Maillard and Bandeira, 2023].

($ii$)   Secondly, the generalized form of Conjecture 4.2 would involve the free entropy of a *tensor denoising* problem. While a rich literature has studied the fundamental limits of denoising low-rank tensors (see Lesieur et al. [2017], Ben Arous et al. [2019], Ros et al. [2019], Perry et al. [2020], Gamarnik et al. [2022] and references therein), here $T^\star$ has rank $m = \mathcal{O}(d)$, and the optimal denoising of a large-rank tensor is, as far as we know, a completely open question.

These two challenges form the basis of an exciting but very challenging research program, which we leave for future work. Provided such a program could be carried out for any polynomial activation, one might then hope to analyze generic activation functions, such as the ReLU or sigmoid, e.g. by decomposition over a basis of orthogonal polynomials (such as the Hermite basis), see Ben Arous et al. [2021], Abbe et al. [2023], Dandi et al. [2023, 2024b] for examples of such analyses in the case $m = \mathcal{O}(1)$.

## D   Derivation of Claim 2 from the replica method

In this section, we give a non-rigorous derivation of eq. (12) using classical methods of statistical physics. We start from the definition of the partition function in eq. (11). We denote $\mathcal{D}$ the standard Gaussian measure, and $\mathbf{S}(\mathbf{W}) = \mathbf{W}^\mathsf{T} \mathbf{W}/m$.

$$\mathcal{Z}(\mathbf{S}^\star, \{\mathbf{x}_i\}_{i=1}^n) = \int_{\mathbb{R}^{m \times d}} \mathcal{D}\mathbf{W} \prod_{i=1}^{n} P_{\text{out}} \left( y_i | \text{Tr}[\mathbf{Z}_i \mathbf{S}(\mathbf{W})] \right).$$

**The replica method –** We make use of the heuristic *replica trick* [Mézard et al., 1987]. Letting $\Phi_d := (1/d^2)\mathbb{E} \log \mathcal{Z}$, it consists in writing that $\lim_{d \to \infty} \Phi_d = \lim_{r \to 0}(\partial/\partial r) \lim_{d \to \infty} \Phi_d(r)$, with $\Phi_d(r) := (1/d^2) \log \mathbb{E}[\mathcal{Z}^r]$. One then computes the $d \to \infty$ limit of $\Phi_d(r)$ for *integer* $r \in \mathbb{N}$, before extending analytically the result to any $r \geq 0$. While being non-rigorous, the replica method has achieved a great success in the study of both spin glasses and statistical learning models, and is widely conjectured to yield exact predictions. We refer the reader to Mézard et al. [1987] for an introduction to the replica method in the context of the statistical physics of disordered systems, Maillard et al. [2023], Montanari and Sen [2024] for mathematically-friendly descriptions of the method, and to Mezard and Montanari [2009], Zdeborová and Krzakala [2016], Gabrié [2020] for

some of its applications in the context of theoretical computer science, high-dimensional statistics, and machine learning.

**The replicated free entropy –** We now compute the "replicated free entropy" $\Phi_d(r)$, for $r \in \mathbb{N}$. Thanks to Bayes-optimality, we can write it as an average over $r + 1$ *replicas* of the system, writing $\mathbf{S}^\star$ as the replica of index 0. We write $\mathbf{S}^a := \mathbf{S}(\mathbf{W}^a)$ to simplify notations. We reach:

$$\Phi_d(r) = \frac{1}{d^2} \log \int \prod_{a=0}^{r} \mathcal{D}\mathbf{W}^a \left[ \int \mathrm{d}y \, \mathbb{E}_{\mathbf{Z}} \prod_{a=0}^{r} P_{\mathrm{out}}(y|\mathrm{Tr}[\mathbf{S}^a \mathbf{Z}]) \right]^n . \tag{31}$$

For a fixed set of matrices $\{\mathbf{S}^a\}_{a=0}^r$, by the central limit theorem the law of the variables $z^a := \mathrm{Tr}[\mathbf{S}^a \mathbf{Z}]$ approach, as $d \to \infty$, a correlated Gaussian distribution, with mean $\mathbb{E}[z^a] = 0$, and covariance $\mathbb{E}[z^a z^b] = \mathbb{E}_{\mathbf{Z}}[\mathrm{Tr}[\mathbf{S}^a \mathbf{Z}]\mathrm{Tr}[\mathbf{S}^b \mathbf{Z}]] = 2\mathrm{tr}(\mathbf{S}^a \mathbf{S}^b)$, as is easily checked from the fact that $\mathbf{Z} \overset{d}{=} (\mathbf{x}\mathbf{x}^\intercal - \mathrm{I}_d)/\sqrt{d}$, with $\mathbf{x} \sim \mathcal{N}(0, \mathrm{I}_d)$. Since $n = \Theta(d^2)$, the leading order of the term $\int \mathrm{d}y \, \mathbb{E}_{\mathbf{Z}} \prod_{a=0}^{r} P_{\mathrm{out}}(y|\mathrm{Tr}[\mathbf{S}^a \mathbf{Z}])$ will be the only one entering the leading order of $\Phi_d(r)$. This means that we have, denoting the *overlap matrix*

$$Q_{ab} := \mathrm{tr}(\mathbf{S}^a \mathbf{S}^b), \tag{32}$$

that

$$\Phi_d(r) = \frac{1}{d^2} \log \int \prod_{a=0}^{r} \mathcal{D}\mathbf{W}^a \left[ \int_{\mathbb{R} \times \mathbb{R}^{r+1}} \frac{\mathrm{d}y \, \mathrm{d}\mathbf{z} \, e^{-\frac{1}{4}\mathbf{z}^\intercal \mathbf{Q}^{-1}\mathbf{z}}}{(4\pi)^{r+1/2}\sqrt{\det \mathbf{Q}}} \prod_{a=0}^{r} P_{\mathrm{out}}(y|z^a) \right]^n + o_d(1),$$

$$= \frac{1}{d^2} \log \int \mathrm{d}\mathbf{Q} \int \prod_{a=0}^{r} \mathcal{D}\mathbf{W}^a \left[ \int_{\mathbb{R} \times \mathbb{R}^{r+1}} \mathrm{d}y \, \mathrm{d}\mathbf{z} \frac{e^{-\frac{1}{4}\mathbf{z}^\intercal \mathbf{Q}^{-1}\mathbf{z}}}{(4\pi)^{r+1/2}\sqrt{\det \mathbf{Q}}} \prod_{a=0}^{r} P_{\mathrm{out}}(y|z^a) \right]^n$$

$$\times \prod_{a \leq b} \delta(d^2 Q_{ab} - d\mathrm{tr}(\mathbf{S}^a \mathbf{S}^b)) + o_d(1). \tag{33}$$

Notice that the CLT-based argument above is made formal in Conjecture 4.1, and implies the universality of $\Phi_d$ under the replacement of $\mathbf{Z}_i$ by Gaussian GOE matrices $\mathbf{G}_i$. Since $\mathbf{Q} \in \mathbb{R}^{(r+1)\times(r+1)}$ is of finite size as $d \to \infty$, we can perform the Laplace method over $\mathbf{Q}$ in eq. (33), and we reach (omitting $o_d(1)$ terms as $d \to \infty$, and recall $n/d^2 \to \alpha$):

$$\Phi_d(r) = \sup_{\mathbf{Q} \in \mathcal{S}_{r+1}^+} \left[ J(\mathbf{Q}) + \alpha J_{\mathrm{out}}(\mathbf{Q}) \right], \tag{34}$$

where $\mathcal{S}_{r+1}^+$ is the set of positive semi-definite symmetric matrices of size $r + 1$, and:

$$\begin{cases} J(\mathbf{Q}) & := \frac{1}{d^2} \log \int \prod_{a=0}^{r} \mathcal{D}\mathbf{W}^a \prod_{a \leq b} \delta(d^2 Q_{ab} - d\mathrm{Tr}[\mathbf{S}^a \mathbf{S}^b]), \\ J_{\mathrm{out}}(\mathbf{Q}) & := \log \int_{\mathbb{R} \times \mathbb{R}^{r+1}} \mathrm{d}y \, \mathrm{d}\mathbf{z} \frac{e^{-\frac{1}{4}\mathbf{z}^\intercal \mathbf{Q}^{-1}\mathbf{z}}}{(4\pi)^{r+1/2}\sqrt{\det \mathbf{Q}}} \prod_{a=0}^{r} P_{\mathrm{out}}(y|z^a). \end{cases} \tag{35}$$

Notice that we can rewrite $J(\mathbf{Q})$ using Lagrange multipliers $\hat{\mathbf{Q}} \in \mathcal{S}_{r+1}$ (or equivalently using the Fourier transform of the delta distribution, and the saddle point method on the Fourier parameters) as:

$$J(\mathbf{Q}) = \inf_{\hat{\mathbf{Q}} \in \mathcal{S}_{r+1}} \left[ \frac{1}{4}\mathrm{Tr}[\mathbf{Q}\hat{\mathbf{Q}}] + \frac{1}{d^2} \log \int \prod_{a=0}^{r} \mathcal{D}\mathbf{W}^a \, e^{-\frac{d}{4}\sum_{a,b} \hat{Q}_{ab}\mathrm{Tr}[\mathbf{S}^a \mathbf{S}^b]} \right]. \tag{36}$$

**The replica-symmetric ansatz –** An important assumption we make now is that there is a permutation symmetry between the different replicas in eq. (34), and we assume that this symmetry is not broken by the maximizer $\mathbf{Q}$. This assumption is usually called *replica symmetry*, and is known to hold in generic statistical learning problems when they are in the Bayes-optimal setting [Zdeborová and Krzakala, 2016, Barbier et al., 2019]. Formally, we assume that the supremum over $\mathbf{Q}$ in eq. (34) (and the infimum over $\hat{\mathbf{Q}}$ in eq. (36)) are reached in matrices such that, for all $a, b \in \{0, \cdots, r\}$ with $a \neq b$:

$$\begin{cases} Q_{aa} = Q, & \hat{Q}_{ab} = \hat{Q}, \\ Q_{ab} = q, & \hat{Q}_{ab} = -\hat{q}, \end{cases} \tag{37}$$

with $0 \leq q \leq Q$, and $\hat{Q}, \hat{q} \geq 0$.

**The term $J_{\text{out}}(\mathbf{Q})$** – Under the ansatz of eq. (37), it is a classical computation [Zdeborová and Krzakala, 2016] to reach:

$$J_{\text{out}}(\mathbf{Q}) = \log \int_{\mathbb{R}^2} \mathrm{d}y \, \mathcal{D}\xi \left\{ \int \frac{\mathrm{d}z}{\sqrt{4\pi(Q-q)}} \exp\left[-\frac{(z-\sqrt{2q}\xi)^2}{4(Q-q)}\right] P_{\text{out}}(y|z) \right\}^{r+1}. \tag{38}$$

**The term $J(\mathbf{Q})$** – Using the replica-symmetric ansatz of eq. (37) in eq. (36), we get:

$$J(\mathbf{Q}) = \inf_{\hat{Q},\hat{q}} \left[ \frac{(r+1)(Q\hat{Q} - rq\hat{q})}{4} + \frac{1}{d^2} \log \int \prod_{a=0}^{r} \mathcal{D}\mathbf{W}^a \, e^{-\frac{d(\hat{Q}+\hat{q})}{4} \sum_a \mathrm{Tr}[(\mathbf{S}^a)^2] + \frac{d\hat{q}}{4} \mathrm{Tr}\left[\left(\sum_a \mathbf{S}^a\right)^2\right]} \right].$$

We now use the following Gaussian integration identity, for any symmetric matrix $\mathbf{M}$:

$$\mathbb{E}_{\boldsymbol{\xi}\sim\text{GOE}(d)}\left[e^{\frac{d}{2}\mathrm{Tr}[\mathbf{M}\boldsymbol{\xi}]}\right] = e^{\frac{d}{4}\mathrm{Tr}[\mathbf{M}^2]}.$$

This allows to reach the following expression, which is analytic in $r$:

$$J(\mathbf{Q}) = \inf_{\hat{Q},\hat{q}} \left[ \frac{(r+1)}{4}Q\hat{Q} - \frac{r(r+1)}{4}q\hat{q} \right.$$

$$\left. + \frac{1}{d^2} \log \mathbb{E}_{\boldsymbol{\xi}} \left\{ \left(\int \mathcal{D}\mathbf{W} \, e^{-\frac{d(\hat{Q}+\hat{q})}{4}\mathrm{Tr}[\mathbf{S}^2] + \frac{d\sqrt{\hat{q}}}{2}\mathrm{Tr}[\mathbf{S}\boldsymbol{\xi}]}\right)^{r+1} \right\} \right]. \tag{39}$$

Recall that here $\mathbf{S} = \mathbf{S}(\mathbf{W}) = \mathbf{W}^{\mathsf{T}}\mathbf{W}/m$.

**The limit $r \to 0$** – From eqs. (34), (38) and (39), we have:

$$\Phi_d(r=0) = \sup_{Q\geq 0} \inf_{\hat{Q}\in\mathbb{R}} \left[ \frac{1}{4}Q\hat{Q} + \frac{1}{d^2} \log \int \mathcal{D}\mathbf{W} e^{-\frac{d\hat{Q}}{4}\mathrm{Tr}[\mathbf{S}^2]} \right]. \tag{40}$$

This implies that $\hat{Q} = 0$ and $Q = Q_0 = \lim_{d\to\infty} \mathbb{E}_{\mathbf{S}\sim\mathcal{W}_{m,d}} \mathrm{tr}[\mathbf{S}^2] = 1 + \kappa^{-1}$ (recall $m/d \to \kappa$), and we correctly recover that $\Phi_d(r=0) = 0$. Taking now the derivative with respect to $r$, followed by the $r \to 0$ limit, yields:

$$\lim_{d\to\infty} \Phi_d = \sup_{0\leq q\leq Q_0} \inf_{\hat{q}\geq 0} \left[ -\frac{q\hat{q}}{4} + \alpha \int_{\mathbb{R}^2} \mathrm{d}y \, \mathcal{D}\xi J_q(y,\xi) \log J_q(y,\xi) \right. \tag{41}$$

$$\left. + \lim_{d\to\infty} \frac{1}{d^2} \mathbb{E}_{\boldsymbol{\xi}\sim\text{GOE}(d)} \left[H_{\hat{q}}(\boldsymbol{\xi}) \log H_{\hat{q}}(\boldsymbol{\xi})\right] \right],$$

$$H_{\hat{q}}(\boldsymbol{\xi}) := \int_{\mathbb{R}^{m\times d}} \mathcal{D}\mathbf{W} \, e^{-\frac{d\hat{q}}{4}\mathrm{Tr}[\mathbf{S}^2] + \frac{d\sqrt{\hat{q}}}{2}\mathrm{Tr}[\mathbf{S}\boldsymbol{\xi}]}, \tag{42}$$

$$J_q(y,\xi) := \int \frac{\mathrm{d}z}{\sqrt{4\pi(Q_0-q)}} \exp\left[-\frac{(z-\sqrt{2q}\xi)^2}{4(Q_0-q)}\right] P_{\text{out}}(y|z). \tag{43}$$

In order to obtain from eq. (41) the prediction of eq. (12), it therefore suffices to show that, for any $\hat{q} \geq 0$:

$$\lim_{d\to\infty} \frac{1}{d^2} \mathbb{E}_{\boldsymbol{\xi}\sim\text{GOE}(d)} \left[H_{\hat{q}}(\boldsymbol{\xi}) \log H_{\hat{q}}(\boldsymbol{\xi})\right] = \frac{Q_0\hat{q}}{4} - \frac{1}{2}\Sigma(\mu_{1/\hat{q}}) - \frac{1}{4}\log\hat{q} - \frac{1}{8}. \tag{44}$$

We focus on deriving eq. (44) in the remaining of this section. We note that we can rewrite the left-hand side as the free entropy of the following denoising problem:

$$\mathbf{Y} = \mathbf{S}^{\star} + \boldsymbol{\xi}/\sqrt{\hat{q}}, \tag{45}$$

with $\boldsymbol{\xi} \sim \text{GOE}(d)$, $\mathbf{S}^{\star} \sim \mathcal{W}_{m,d}$, and which we consider in the Bayes-optimal setting. Indeed, we can define the free entropy of this problem as

$$\frac{1}{d^2} \mathbb{E}_{\mathbf{Y},\mathbf{S}^{\star}} \log \int \mathcal{D}\mathbf{W} \exp\left(-\frac{d\hat{q}}{4}\mathrm{Tr}[(\mathbf{Y}-\mathbf{S})^2]\right)$$

$$= -\frac{\hat{q}\mathbb{E}\mathrm{tr}[\mathbf{Y}^2]}{4} + \frac{1}{d^2} \mathbb{E}_{\mathbf{Y}} \log \int \mathcal{D}\mathbf{W} \exp\left(-\frac{d\hat{q}}{4}\mathrm{Tr}[\mathbf{S}^2] + \frac{d\hat{q}}{2}\mathrm{Tr}[\mathbf{Y}\mathbf{S}]\right),$$

$$= -\frac{1+\hat{q}Q_0}{4} + \frac{1}{d^2} \mathbb{E}_{\boldsymbol{\xi}\sim\text{GOE}(d)}[H_{\hat{q}}(\boldsymbol{\xi}) \log H_{\hat{q}}(\boldsymbol{\xi})]. \tag{46}$$

Crucially, this auxiliary problem is again Bayes-optimal, which we will use in what follows.

**Remark –** Eq. (45) defines a problem known as *extensive-rank matrix denoising*. The limit free entropy of this problem, as well as the analytical form of the Bayes-optimal estimator, for a rotationally-invariant prior on $\mathbf{S}^\star$ and a rotationally invariant noise $\boldsymbol{\xi}$ (which is here Gaussian) have both been understood and worked out completely [Bun et al., 2016, Maillard et al., 2022b, Pourkamali et al., 2024, Semerjian, 2024]. We will leverage these results (and partially re-derive them) in what follows.

We now use a change of variable to the singular values of $\mathbf{W}$, see e.g. Proposition 4.1.3 of Anderson et al. [2010]. We reach:

$$
\int \mathcal{D}\mathbf{W} \exp\left(-\frac{d\hat{q}}{4}\mathrm{Tr}[\mathbf{S}^2] + \frac{d\hat{q}}{2}\mathrm{Tr}[\mathbf{YS}]\right) = C_{d,m}\int_{\mathbb{R}^m_+}\prod_{k=1}^m \mathrm{d}\lambda_k\, e^{-\frac{m}{2}\sum_{k=1}^m \lambda_k}\prod_{k=1}^m \lambda_k^{\frac{d-m}{2}}
$$
$$
\times \prod_{k<k'}|\lambda_k - \lambda_{k'}|\, e^{-\frac{d\hat{q}}{4}\sum_{k=1}^m \lambda_k^2}\int_{\mathcal{O}(d)}\mathcal{D}\mathbf{O}\exp\left\{\frac{d\hat{q}}{2}\mathrm{Tr}[\mathbf{O}\boldsymbol{\Lambda}\mathbf{O}^\intercal\mathbf{Y}]\right\}, \quad (47)
$$

in which $\mathbf{S} = \mathbf{W}^\intercal\mathbf{W}/m = \mathbf{O}\boldsymbol{\Lambda}\mathbf{O}^\intercal$, with $\boldsymbol{\Lambda} = \mathrm{Diag}((\lambda_1, \cdots, \lambda_m, 0, \cdots, 0))$, and $C_{d,m} > 0$ is a constant depending only on $m$ and $d$. Notice that we (slightly abusively) used the notation $\mathcal{D}\mathbf{O}$ to denote here the Haar measure over the orthogonal group $\mathcal{O}(d)$. The large-$d$ limit of the last term is given by the *HCIZ integral* [Harish-Chandra, 1957, Itzykson and Zuber, 1980]:

$$
I_{\mathrm{HCIZ}}(\theta, \mathbf{R}, \mathbf{Y}) = I_{\mathrm{HCIZ}}(\theta, \mu_\mathbf{S}, \mu_\mathbf{Y}) := \lim_{d\to\infty}\frac{2}{d^2}\log\int_{\mathcal{O}(d)}\mathcal{D}\mathbf{O}\exp\left\{\frac{\theta d}{2}\mathrm{Tr}[\mathbf{O}\mathbf{S}\mathbf{O}^\intercal\mathbf{Y}]\right\}, \quad (48)
$$

where $\mathbf{S}$ and $\mathbf{Y}$ are $d \times d$ matrices with asymptotic eigenvalue distributions $\mu_\mathbf{S}$ and $\mu_\mathbf{Y}$. We can now apply the Laplace method in eq. (47) on the eigenvalue distribution of $\mathbf{S}$. As the problem of eq. (45) is Bayes-optimal, it is known that the typical eigenvalue distribution of $\mathbf{S}$ under the distribution of eq. (47) is $\mu_\mathbf{S} = \mu_{\mathbf{S}^\star} = \mu_{\mathrm{MP},\kappa}$, as a consequence of the so-called Nishimori identity, so that $\mu_{\mathrm{MP},\kappa}$ is the maximizer of the variational problem obtained by the use of Laplace's method, see Maillard et al. [2022b] for details. Since the asymptotic distribution of $\mathbf{Y}$ is (by eq. (45)) $\mu_\mathbf{Y} = \mu_{\mathrm{MP},\kappa}\boxplus\sigma_{\mathrm{s.c.},1/\sqrt{\hat{q}}}$, we reach by eq. (46):

$$
\lim_{d\to\infty}\frac{1}{d^2}\mathbb{E}_{\boldsymbol{\xi}\sim\mathrm{GOE}(d)}[H_{\hat{q}}(\boldsymbol{\xi})\log H_{\hat{q}}(\boldsymbol{\xi})] = C(\kappa) - \frac{\hat{q}Q_0}{4} + \frac{1}{2}I_{\mathrm{HCIZ}}(\hat{q}, \mu_{\mathrm{MP},\kappa}, \mu_\mathbf{Y}), \quad (49)
$$

where $C(\kappa)$ is a function of $\kappa = m/d$. It can be easily seen that $C(\kappa) = 0$ by considering $\hat{q} = 0$.

Fortunately, extensive-rank matrix denoising with Gaussian noise is one of the very few cases for which an easily tractable analytical form is known for the HCIZ integral. More specifically, we know that for any $t > 0$ and any $\nu$, we have with $\mu_t := \nu \boxplus \sigma_{\mathrm{s.c.},\sqrt{t}}$ [Maillard et al., 2022b]:

$$
-\frac{1}{2}\Sigma(\mu_t) + \frac{1}{4t}\mathbb{E}_{\mu_t}[X^2] - \frac{1}{2}I_{\mathrm{HCIZ}}(t^{-1}, \mu_t, \nu) - \frac{3}{8} + \frac{1}{4}\log t + \frac{1}{4t}\mathbb{E}_\nu[X^2] = 0,
$$

with $\Sigma(\mu) := \int \mu(\mathrm{d}x)\mu(\mathrm{d}y)\log|x-y|$. Applying this formula with $t = 1/\hat{q}$ we reach:

$$
\frac{1}{2}I_{\mathrm{HCIZ}}(\hat{q}, \mu_{\mathrm{MP},\kappa}, \mu_\mathbf{Y}) = -\frac{1}{2}\Sigma(\mu_\mathbf{Y}) + \frac{\hat{q}}{4}\mathbb{E}[\mathrm{tr}(\mathbf{Y}^2)] - \frac{3}{8} - \frac{1}{4}\log\hat{q} + \frac{\hat{q}}{4}\mathbb{E}\mathrm{tr}[(\mathbf{S}^\star)^2],
$$
$$
= -\frac{1}{2}\Sigma(\mu_\mathbf{Y}) + \frac{1}{4}(2Q_0\hat{q} + 1) - \frac{3}{8} - \frac{1}{4}\log\hat{q}.
$$

Combining it with eq. (49), we reach eq. (44) (recall that $\mu_\mathbf{Y} = \mu_{1/\hat{q}}$ with the notations of eq. (44)).

## E  Large and small $\kappa$ limits

### E.1  The small-$\kappa$ limit

We consider here the limit $\kappa \to 0$, i.e. the limit of small (but still extensively large) hidden layer, and compute the limit of the MMSE curves shown in Fig. 1. Since in the noiseless setting we have $\alpha_{\mathrm{PR}} = \kappa + \mathcal{O}(\kappa^2)$ (cf. eq. (1)), we will work in the rescaled regime $\alpha = \widetilde{\alpha}\kappa$, with $\widetilde{\alpha}$ remaining finite

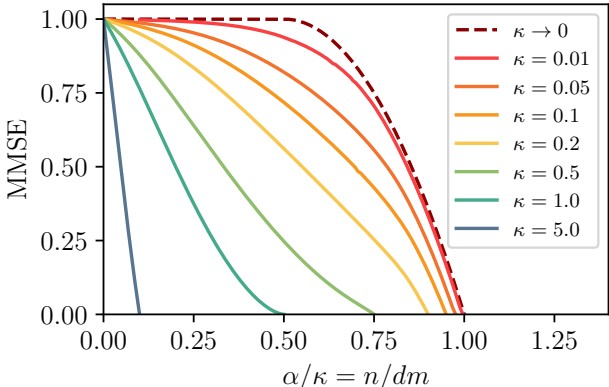

Figure 4: Behavior of the asymptotic MMSE in the noiseless ($\Delta = 0$) case as $\kappa$ gets increasingly small. The continuous lines are given by eq. (7), which we compare with the asymptotic $\kappa \to 0$ curve obtained by eq. (51). We emphasize that the horizontal axis is $\alpha/\kappa$, which remains of order $\Theta(1)$ as $\kappa \to 0$: it corresponds to a number of samples $n$ of the same order as the number of parameters $dm$.

as $\kappa \downarrow 0$. By analyzing eq. (8) in this regime (details are given in Appendix E.1.1), we reach that the MMSE satisfies, as $\kappa \to 0$:

$$
\text{MMSE} = \begin{cases} 1 & \text{if } \widetilde{\alpha} \leq \dfrac{1 + \Delta(2 + \Delta)}{2}, \\ -\Delta(2 + \Delta) + 2\widetilde{\alpha}\left[1 - \widetilde{\alpha} + \sqrt{(1 - \widetilde{\alpha})^2 + \Delta(2 + \Delta)}\right] & \text{if } \widetilde{\alpha} \geq \dfrac{1 + \Delta(2 + \Delta)}{2}. \end{cases} \tag{50}
$$

In particular, in the noiseless case ($\Delta = 0$), we have:

$$
\text{MMSE} = \mathbb{1}\left\{\widetilde{\alpha} \leq \frac{1}{2}\right\} + 4\widetilde{\alpha}(1 - \widetilde{\alpha})\mathbb{1}\left\{\widetilde{\alpha} > \frac{1}{2}\right\}. \tag{51}
$$

and we reach perfect recovery for $\widetilde{\alpha} = 1$. This limit is illustrated in Fig. 4.

Remarkably, eq. (51) can be computed as well by taking the limit $m \to \infty$ when assuming that $m = \mathcal{O}(1)$ as $d \to \infty$, a setting which was studied extensively in the literature (see Aubin et al. [2019b] and references therein). We detail this computation in Appendix E.1.2.

### E.1.1   Details of the small-$\kappa$ limit

Recall that by Claim 2, we have $\text{MMSE} = \kappa(Q_0 - q^\star)$, with $Q_0 = 1 + \kappa^{-1}$. Since the MMSE remains finite as $\kappa \to 0$, we consider the scaling $q = \widetilde{q}/\kappa$, with $0 \leq \widetilde{q} \leq 1$. We start again from eqs. (7) and (8). We denote $\Lambda := \Delta(2 + \Delta)$. Eq. (7), combined with the scaling of $\alpha$, implies that $\hat{q} \sim \kappa^2/t$ for some finite $t > 0$, and since $\text{MMSE} = 1 - \widetilde{q}$ as $\kappa \to 0$, we have

$$
t = \frac{\kappa^2}{\hat{q}} = \frac{1 - \widetilde{q} + \Lambda}{2\widetilde{\alpha}}.
$$

Moreover, eq. (8) at order $\mathcal{O}(\kappa)$ yields:

$$
-2\widetilde{\alpha} + \frac{\Lambda}{t} = \partial_\kappa[F(t, \kappa)]_{\kappa=0}, \tag{52}
$$

where

$$
F(t, \kappa) := \frac{4\pi^2 t}{3\kappa^2} \int \mu_{t/\kappa^2}(y)^3 \mathrm{d}y.
$$

Notice that $F(t, 0) = 1$ since $\mu_\xi \simeq \sigma_{\text{s.c.}, \sqrt{\xi}}$ for $\xi \to \infty$, and $\int \sigma_{\text{s.c.}, \sqrt{\xi}}(y)^3 \mathrm{d}y = 3/[4\pi^2\xi]$. Thus, the leading order of eq. (8) as $\kappa \to 0$ is consistent but not informative.

In what follows, we work out the small $\kappa$ limit of $F(t,\kappa)$, at first order in $\kappa$. We denote $\nu_\kappa(y) := (1/\kappa)\mu_{t/\kappa^2}(y/\kappa)$, so that the Stieltjes transform $g_\kappa(z) := \int \nu(y)/(y-z)\mathrm{d}y$ of $\nu$ satisfies the self-consistent equation (see Appendix A):

$$z = \frac{\kappa}{1+g} - \frac{1}{g} - tg. \tag{53}$$

Moreover, we notice that $\nu_\kappa = (\kappa \# \mu_{\mathrm{MP},\kappa}) \boxplus \sigma_{\mathrm{s.c.},\sqrt{t}}$, so that the support of $\nu$ remains bounded as $\kappa \to 0$. We then proceed to expand in $\kappa$ eq. (53). For any finite $z \in \mathbb{C}$, the leading order of the expansion is easily given by $z = -1/h - th + o_\kappa(1)$, which gives that $\nu_\kappa \to \sigma_{\mathrm{s.c.},\sqrt{t}}$. However, as mentioned above, we need to go to the next order in this expansion to compute eq. (52).

**A BBP-type transition –** We notice that $\kappa \# \mu_{\mathrm{MP},\kappa}(x) \simeq (1-\kappa)\delta(x) + \kappa\delta(x-1)$ when $\kappa \to 0$. More precisely, it is composed of a mass $(1-\kappa)$ in $0$, and the rest of the mass $\kappa$ is made up of a continuous part supported between $(1-\sqrt{\kappa})^2 \simeq 1 - 2\sqrt{\kappa}$ and $(1+\sqrt{\kappa})^2 \simeq 1 + 2\sqrt{\kappa}$. $\nu_\kappa$ can thus be seen as the spectral density of the sum of a GOE matrix (with variance $t$) and a small-rank perturbation matrix of rank $m = \kappa d$, with all non-zero eigenvalues located close to $1$. We therefore expect by the so-called BBP transition phenomenon [Benaych-Georges and Nadakuditi, 2011] that $\nu_\kappa$ will possess a set of $m$ eigenvalues outside the semicircle bulk whenever the condition

$$1 \geq -\frac{1}{g_{\mathrm{s.c.},\sqrt{t}}(2\sqrt{t})} \tag{54}$$

is satisfied, with $g_{\mathrm{s.c.},\sqrt{t}}(z) := \mathbb{E}_{X \sim \sigma_{\mathrm{s.c.},\sqrt{t}}}[1/(X-z)]$ the Stieltjes transform of the semicircle. Since one can easily show that $g_{\mathrm{s.c.},\sqrt{t}}(2\sqrt{t}) = -t^{-1/2}$, eq. (54) is equivalent to $t \leq 1$. In this case, these "spiked" eigenvalues are located around the value [Benaych-Georges and Nadakuditi, 2011]

$$g_{\mathrm{s.c.},\sqrt{t}}^{-1}(-1) = \mathcal{R}_{\mathrm{s.c.},\sqrt{t}}(1) + 1 = 1 + t.$$

Moreover, as the width of the continuous part of $\kappa \# \mu_{\mathrm{MP},\kappa}$ is of size $\mathcal{O}(\sqrt{\kappa})$, we also expect this "spiked" part of the spectrum to have a width $\mathcal{O}(\sqrt{\kappa})$.

**Expansion of $\nu$ –** Based on the remarks of the previous paragraph, we assume the following behavior for $\nu_\kappa$, as $\kappa \to 0$. For any $y \in \mathbb{R}$ with $y \neq 1 + t$, we have

$$\nu_\kappa(y) = \sigma_{\mathrm{s.c.},\sqrt{t}}(y) + \kappa\nu^{(1)}(y) + o(\kappa). \tag{55}$$

Furthermore, we also have, for all $y \in \mathbb{R}$, when $t \leq 1$:

$$\sqrt{\kappa}\nu_\kappa\left(\frac{y - (1+t)}{\sqrt{\kappa}}\right) \to_{\kappa \to 0} \rho^{(1)}(y), \tag{56}$$

for a finite density $\rho^{(1)}$, with $\int \rho^{(1)}(y)\mathrm{d}y = 1$. Eqs. (55) and (56) can be used to expand the Stieltjes transform of $\nu_\kappa$ as a function of $\nu^{(1)}, \rho^{(1)}$, and then eq. (53) used to find the values of these two functions. These computations are straightforward, and yield:

$$\begin{cases} \nu^{(1)}(y) &= \dfrac{(y-2)}{2\pi(1+t-y)\sqrt{4t-y^2}}\mathbb{1}\{|y| \leq 2\sqrt{t}\}, \\ \rho^{(1)} &= \rho_{\mathrm{s.c.},\sqrt{1-t}}. \end{cases} \tag{57}$$

Notice that the second equation of eq. (57) is only valid for $t \leq 1$, while the first one is valid for all $t \geq 0$. One checks for instance that $\int \nu^{(1)}(\mathrm{d}y) = -\mathbb{1}\{t \leq 1\}$, which implies that the normalization condition $\int \nu_\kappa(y)\mathrm{d}y = 1$ is well satisfied for all values of $t \geq 0$. Using the expansion of eq. (57), we obtain that

$$F(t,\kappa) = \frac{4\pi^2 t}{3}\int \nu_\kappa(y)^3\mathrm{d}y,$$

$$= 1 - \kappa\begin{cases} 2-t & \text{if } t \leq 1, \\ 1/t & \text{if } t \geq 1 \end{cases} + o(\kappa).$$

So finally eq. (52) becomes

$$2\widetilde{\alpha} - \frac{\Lambda}{t} = \begin{cases} 2-t & \text{if } t \leq 1, \\ 1/t & \text{if } t \geq 1, \end{cases}$$

And recall that MMSE $= 2\widetilde{\alpha}t - \Lambda$, so that

$$
\text{MMSE} = \begin{cases} t(2-t) & \text{if } t \leq 1, \\ 1 & \text{if } t \geq 1, \end{cases}
$$

Since $t = (\text{MMSE} + \Lambda)/(2\widetilde{\alpha})$, we reach that $t = (1 + \Lambda)/(2\widetilde{\alpha})$ if $\widetilde{\alpha} \leq (1 + \Lambda)/2$, and $t = 1 - \widetilde{\alpha} + \sqrt{(1 - t\alpha)^2 + \Lambda}$ otherwise. This yields eq. (50).

### E.1.2 The small-$\kappa$ limit from a large but finite hidden layer

We consider here the noiseless case:

$$
y_i = \frac{1}{m} \sum_{k=1}^{m} \left[ \frac{(\mathbf{w}_k^\star)^\intercal \mathbf{x}_i}{\sqrt{d}} \right]^2,
$$

with $m = \mathcal{O}(1)$ as $n, d \to \infty$. We denote $\alpha = n/d = \widetilde{\alpha}m$, and we assume that $\widetilde{\alpha} = \Theta(1)$ as $m \to \infty$ (*after* $n, d \to \infty$). We can write the partition function (cf. eq. (11)) as:

$$
\mathcal{Z} = \int_{\mathbb{R}^{d \times m}} \mathcal{D}\mathbf{W} \prod_{i=1}^{n} P_{\text{out}} \left( y_i \bigg| \frac{\mathbf{w}_k^\intercal \mathbf{x}_i}{\sqrt{d}} \right), \tag{58}
$$

with $P_{\text{out}}(y|\mathbf{z}) = \delta(y - \|\mathbf{z}\|^2/m)$. We can make a direct use of the results of Aubin et al. [2019b] to write:

$$
\lim_{d \to \infty} \frac{1}{d} \mathbb{E} \log \mathcal{Z} = \text{extr}_{\mathbf{q}, \hat{\mathbf{q}}} \left\{ -\frac{1}{2} \text{Tr}[\mathbf{q}\hat{\mathbf{q}}] + I_P + m\widetilde{\alpha}I_C \right\}, \tag{59}
$$

$$
\begin{cases}
I_P & := \int_{\mathbb{R}^m} \mathcal{D}\boldsymbol{\xi} \int_{\mathbb{R}^m} \mathcal{D}\mathbf{w}^0 \exp\left[ -\frac{1}{2}(\mathbf{w}^0)^\intercal \hat{\mathbf{q}}\mathbf{w}^0 + \boldsymbol{\xi}^\intercal \hat{\mathbf{q}}^{1/2}\mathbf{w}^0 \right] \\
& \qquad\qquad \times \log\left[ \int_{\mathbb{R}^m} \mathcal{D}\mathbf{w}^0 \exp\left[ -\frac{1}{2}\mathbf{w}^\intercal \hat{\mathbf{q}}\mathbf{w} + \boldsymbol{\xi}^\intercal \hat{\mathbf{q}}^{1/2}\mathbf{w} \right] \right], \\
I_C & := \int_0^\infty dy \int_{\mathbb{R}^m} \mathcal{D}\boldsymbol{\xi} \int_{\mathbb{R}^m} \mathcal{D}\mathbf{Z}^0 P_{\text{out}} \left\{ y|(\mathbf{I}_m - \mathbf{q})^{1/2}\mathbf{Z}^0 + \mathbf{q}^{1/2}\boldsymbol{\xi} \right\} \\
& \qquad\qquad \times \log\left[ \int_{\mathbb{R}^m} \mathcal{D}\mathbf{Z} P_{\text{out}} \left\{ y|(\mathbf{I}_m - \mathbf{q})^{1/2}\mathbf{Z} + \mathbf{q}^{1/2}\boldsymbol{\xi} \right\} \right].
\end{cases}
$$

Here, $\mathbf{q}, \hat{\mathbf{q}}$ are symmetric $m \times m$ matrices, which satisfy moreover $\mathbf{I}_m \succeq \mathbf{q} \succeq 0$ and $\hat{\mathbf{q}} \succeq 0$. The informal notation "extr $f$" in eq. (59) means that one should zero-out the gradient of the function $f$ to compute the values of $\mathbf{q}, \hat{\mathbf{q}}$.

**The matrix $\mathbf{q}$** – Importantly, the matrix $\mathbf{q}$ can be interpreted as the "overlap matrix" of the model: if we denote $\langle \cdot \rangle$ the average under the posterior measure in eq. (58), then we have

$$
q_{kl} = \mathbb{E} \left\langle \frac{\mathbf{w}_k^\intercal \mathbf{w}_l'}{d} \right\rangle, \tag{60}
$$

where $\mathbf{w}, \mathbf{w}'$ are two independent samples under $\langle \cdot \rangle$. Moreover, thanks to the Bayes-optimality of the problem, it is known that the overlap concentrates [Zdeborová and Krzakala, 2016], in the sense that the random variable $(\mathbf{w}_k^\intercal \mathbf{w}_l')/d$ concentrates on its average under $\mathbb{E}\langle \cdot \rangle$ as $d \to \infty$.

The "prior integral" $I_P$ can be very easily computed with Gaussian integrals, and yields:

$$
I_P = \frac{1}{2} \text{Tr}[\hat{\mathbf{q}}] - \frac{1}{2} \log \det(\mathbf{I}_m + \hat{\mathbf{q}}). \tag{61}
$$

We now focus on computing the leading order of $I_C$ in the large-$m$ limit. We can write

$$
\begin{cases}
I_C & = \int dy \, \mathcal{D}\boldsymbol{\xi} \, I_{\mathbf{q}}(y, \boldsymbol{\xi}) \log I_{\mathbf{q}}(y, \boldsymbol{\xi}), \\
I_{\mathbf{q}}(y, \boldsymbol{\xi}) & = \int_{\mathbb{R}^m} \mathcal{D}\mathbf{Z} \, \delta\left( y - \frac{1}{m} \left\| (\mathbf{I}_m - \mathbf{q})^{1/2}\mathbf{Z} + \mathbf{q}^{1/2}\boldsymbol{\xi} \right\|_2^2 \right).
\end{cases} \tag{62}
$$

Let $\widetilde{y} := \sqrt{m}[y - \mathrm{tr}(\mathrm{I}_m - \mathbf{q}) - (\boldsymbol{\xi}^\mathsf{T}\mathbf{q}\boldsymbol{\xi})/m]$. We can change variables in eq. (62), and obtain:

$$
\begin{cases}
I_C & = \int \mathrm{d}\widetilde{y}\, \mathcal{D}\boldsymbol{\xi}\, J_{\mathbf{q}}(\widetilde{y}, \boldsymbol{\xi}) \log J_{\mathbf{q}}(\widetilde{y}, \boldsymbol{\xi}) + \dfrac{1}{2}\log m, \\
J_{\mathbf{q}}(\widetilde{y}, \boldsymbol{\xi}) & = \displaystyle\int_{\mathbb{R}^m} \mathcal{D}\mathbf{Z}\,\delta\left(\widetilde{y} - \sqrt{m}\left[\dfrac{1}{m}\left\| (\mathrm{I}_m - \mathbf{q})^{1/2}\mathbf{Z} + \mathbf{q}^{1/2}\boldsymbol{\xi} \right\|_2^2 - \mathrm{tr}(\mathrm{I}_m - \mathbf{q}) - \dfrac{\boldsymbol{\xi}^\mathsf{T}\mathbf{q}\boldsymbol{\xi}}{m} \right]\right).
\end{cases}
\tag{63}
$$

Notice that the additive term $(1/2)\log m$ in $I_C$ just amounts to a renormalization of the partition function $\mathcal{Z}$, so we remove this additional constant in what follows. We proceed to simplify $J_{\mathbf{q}}(\widetilde{y}, \boldsymbol{\xi})$ in the large-$m$ limit. We have

$J_{\mathbf{q}}(\widetilde{y}, \boldsymbol{\xi})$

$$
= \int_{\mathbb{R}^m} \mathcal{D}\mathbf{Z}\,\delta\left(\widetilde{y} - \sqrt{m}\left[\dfrac{\mathbf{Z}^\mathsf{T}(\mathrm{I}_m - \mathbf{q})\mathbf{Z}}{m} - \mathrm{tr}(\mathrm{I}_m - \mathbf{q}) + 2\dfrac{\mathbf{Z}^\mathsf{T}(\mathrm{I}_m - \mathbf{q})^{1/2}\mathbf{q}^{1/2}\boldsymbol{\xi}}{m}\right]\right),
$$

$$
= \int \dfrac{\mathrm{d}u}{2\pi} e^{iu\widetilde{y} + iu\sqrt{m}\,\mathrm{tr}(\mathrm{I}_m - \mathbf{q})} \int \mathcal{D}\mathbf{Z}\, e^{-iu\sqrt{m}\left[\frac{\mathbf{Z}^\mathsf{T}(\mathrm{I}_m - \mathbf{q})\mathbf{Z}}{m} + 2\frac{\mathbf{Z}^\mathsf{T}(\mathrm{I}_m - \mathbf{q})^{1/2}\mathbf{q}^{1/2}\boldsymbol{\xi}}{m}\right]},
$$

$$
= \int \dfrac{\mathrm{d}u}{2\pi} e^{iu\widetilde{y} + iu\sqrt{m}\,\mathrm{tr}(\mathrm{I}_m - \mathbf{q}) - \frac{1}{2}\log\det\left[\mathrm{I}_m + 2\frac{iu(\mathrm{I}_m - \mathbf{q})}{\sqrt{m}}\right] - \frac{2u^2}{m}\boldsymbol{\xi}^\mathsf{T}\mathbf{q}^{1/2}(\mathrm{I}_m - \mathbf{q})^{1/2}\left[\mathrm{I}_m + \frac{2iu(\mathrm{I}_m - \mathbf{q})}{\sqrt{m}}\right]^{-1}(\mathrm{I}_m - \mathbf{q})^{1/2}\mathbf{q}^{1/2}\boldsymbol{\xi}},
$$

$$
= \int \dfrac{\mathrm{d}u}{2\pi} e^{iu\widetilde{y} - u^2\mathrm{tr}[(\mathrm{I}_m - \mathbf{q})^2] - \frac{2u^2}{m}\boldsymbol{\xi}^\mathsf{T}\mathbf{q}^{1/2}(\mathrm{I}_m - \mathbf{q})\mathbf{q}^{1/2}\boldsymbol{\xi} + \mathcal{O}(1/\sqrt{m})},
$$

$$
= \dfrac{1}{\sqrt{2\pi\sigma_{\boldsymbol{\xi}}^2}} e^{-\frac{(\widetilde{y})^2}{2\sigma_{\boldsymbol{\xi}}^2}} + \mathcal{O}(1/\sqrt{m}),
$$

where

$$
\sigma_{\boldsymbol{\xi}}^2 := 2\mathrm{tr}[(\mathrm{I}_m - \mathbf{q})^2] + \dfrac{4}{m}\boldsymbol{\xi}^\mathsf{T}\mathbf{q}^{1/2}(\mathrm{I}_m - \mathbf{q})\mathbf{q}^{1/2}\boldsymbol{\xi}.
$$

Plugging it back into eq. (63) yields:

$$
I_C = \int \mathrm{d}y\, \mathcal{D}\boldsymbol{\xi}\, \dfrac{1}{\sqrt{2\pi\sigma_{\boldsymbol{\xi}}^2}} e^{-\frac{(\widetilde{y})^2}{2\sigma_{\boldsymbol{\xi}}^2}}\left[-\dfrac{1}{2}\log 2\pi\sigma_{\boldsymbol{\xi}}^2 - \dfrac{y^2}{2\sigma_{\boldsymbol{\xi}}^2}\right] + \mathcal{O}(1/\sqrt{m}),
$$

$$
= -\dfrac{1}{2}\int \mathcal{D}\boldsymbol{\xi}\, \log[2\pi\sigma_{\boldsymbol{\xi}}^2] - \dfrac{1}{2} + \mathcal{O}(1/\sqrt{m}).
$$

Since $\boldsymbol{\xi} \sim \mathcal{N}(0, \mathrm{I}_m)$, it follows from elementary concentration of measure that $\sigma_{\boldsymbol{\xi}}^2$ concentrates on its average value $\sigma^2$ given by:

$$
\sigma^2 := 2\mathrm{tr}[(\mathrm{I}_m - \mathbf{q})^2] + 4\mathrm{tr}[(\mathrm{I}_m - \mathbf{q})\mathbf{q}] = 2\mathrm{tr}[(\mathrm{I}_m - \mathbf{q})(\mathrm{I}_m + \mathbf{q})] = 2\mathrm{tr}[\mathrm{I}_m - \mathbf{q}^2].
$$

All in all we reach that (up to additive constants):

$$
I_C = -\dfrac{1}{2}\log \mathrm{tr}[\mathrm{I}_m - \mathbf{q}^2] + \mathcal{O}(1/\sqrt{m}).
\tag{64}
$$

Combining eqs. (61) and (64) in eq. (59), we get at leading order in $m$, with $\Phi := \lim(1/d)\mathbb{E}\log \mathcal{Z}$:

$$
\dfrac{1}{m}\Phi = \operatorname*{extr}_{\mathbf{q},\hat{\mathbf{q}}}\left\{-\dfrac{1}{2}\mathrm{tr}[\mathbf{q}\hat{\mathbf{q}}] + \dfrac{1}{2}\mathrm{tr}[\hat{\mathbf{q}}] - \dfrac{1}{2}\mathrm{tr}\log(\mathrm{I}_m + \hat{\mathbf{q}}) - \dfrac{\widetilde{\alpha}}{2}\log \mathrm{tr}[\mathrm{I}_m - \mathbf{q}^2]\right\}.
\tag{65}
$$

Eq. (65) can be easily solved, and yields:

$$
\begin{cases}
\hat{\mathbf{q}} & = \mathbf{q}(\mathrm{I}_m - \mathbf{q})^{-1}, \\
\hat{\mathbf{q}} & = \dfrac{2\widetilde{\alpha}}{\mathrm{tr}[\mathrm{I}_m - \mathbf{q}^2]}\mathbf{q}.
\end{cases}
$$

This implies that (recall $0 \preceq \mathbf{q} \preceq \mathrm{I}_m$):

$$
\mathbf{q} = \begin{cases} 0 \text{ if } \widetilde{\alpha} \leq \dfrac{1}{2}, \\[2mm] (2\widetilde{\alpha} - 1)\mathrm{I}_m \text{ if } \dfrac{1}{2} \leq \widetilde{\alpha} \leq 1, \\[2mm] \mathrm{I}_m \text{ if } \widetilde{\alpha} \geq 1. \end{cases}
\tag{66}
$$

Now that we have obtained $\mathbf{q}$ in eq. (66), we can compute the MMSE, or generalization error. Defining it as in eq. (4):

$$
\mathrm{MMSE}_d := \frac{m}{2} \mathbb{E}_{\mathbf{W}^\star, \mathcal{D}} \mathbb{E}_{y_{\text{test}}, \mathbf{x}_{\text{test}}} [(y_{\text{test}} - \hat{y}^{\mathrm{BO}}(\mathbf{x}_{\text{test}}))^2],
$$

the same arguments used in the proof of Lemma F.1 show that in the large $m$ limit (but taken *after* $d \to \infty$), we have at leading order

$$
\mathrm{MMSE}_d = \frac{m}{d} \mathbb{E}\mathrm{tr}[(\mathbf{S}^\star - \mathbf{S}^{\mathrm{BO}})^2] = 1 - \frac{m}{d}\mathbb{E}\mathrm{tr}[(\mathbf{S}^{\mathrm{BO}})^2],
$$

with $\mathbf{S} := (1/m)\sum_{k=1}^m \mathbf{w}_k \mathbf{w}_k^\mathsf{T}$. Notice that $\mathbf{S}^{\mathrm{BO}} = \langle \mathbf{S} \rangle$, so that

$$
\mathrm{MMSE}_d = 1 - \frac{1}{m}\mathbb{E}\sum_{1 \leq k,l \leq m} \left\langle \left(\frac{\mathbf{w}_k^\mathsf{T}\mathbf{w}_l'}{d}\right)^2 \right\rangle,
$$

where $\mathbf{w}, \mathbf{w}'$ are two independent samples under the posterior measure $\langle \cdot \rangle$. We know that the overlap concentrates (cf. the discussion around eq. (60)), so that at leading order, with $\mathrm{MMSE} := \lim_{d \to \infty} \mathrm{MMSE}_d$:

$$
\mathrm{MMSE} = 1 - \frac{1}{m}\mathbb{E}\sum_{1 \leq k,l \leq m} q_{kk'}^2 = 1 - \mathrm{tr}[\mathbf{q}^2].
$$

Combining it with eq. (66), we reach:

$$
\mathrm{MMSE} = \begin{cases} 1 \text{ if } \widetilde{\alpha} \leq \dfrac{1}{2}, \\[2mm] 4\widetilde{\alpha}(1 - \widetilde{\alpha}) \text{ if } \dfrac{1}{2} \leq \widetilde{\alpha} \leq 1, \\[2mm] 0 \text{ if } \widetilde{\alpha} \geq 1. \end{cases}
$$

We have recovered eq. (51) from the limit $m \to \infty$ taken after $d \to \infty$!

### E.2 The large-$\kappa$ limit

We consider here $\kappa \to \infty$, with $\alpha$ remaining of order $\Theta(1)$ as $\kappa \to \infty$. Since the MMSE remains finite as well, we see from eq. (7) that we must have the scaling $\hat{q} = \kappa t$, with $t$ remaining finite as $\kappa \to \infty$. A very similar derivation to the one of Appendix E.1.1 yields that eq. (8) in this limit becomes (with $\Lambda := \Delta(2 + \Delta)$):

$$
\begin{aligned}
1 - 2\alpha + \Lambda t &= \lim_{\kappa \to \infty} \frac{4\pi^2}{3\kappa t} \int \mu_{1/[\kappa t]}(y)^3 \mathrm{d}y, \\
&= \frac{1}{1 + t}.
\end{aligned}
$$

Combining it with eq. (7) yields that

$$
\mathrm{MMSE} = \frac{1 - 2\alpha - \Lambda + \sqrt{(1 - 2\alpha + \Lambda)^2 + 8\alpha\Lambda}}{2},
\tag{67}
$$

where we recall $\Lambda = \Delta(2 + \Delta)$. In particular, for $\Delta = 0$, we reach $\mathrm{MMSE} = \max(1 - 2\alpha, 0)$, coherently with the behavior shown in Fig. 1.

# F  Other technicalities

## F.1  Properties of the MMSE of eq. (4)

Let $\mathbf{S}^\star := (1/m)\sum_{k=1}^m \mathbf{w}_k^\star(\mathbf{w}_k^\star)^\mathsf{T}$, and $\hat{\mathbf{S}}^{\mathrm{BO}} := \mathbb{E}[\mathbf{S}|\mathcal{D}]$ the Bayes-optimal estimator of $\mathbf{S}^\star$. We show here the following lemma on the MMSE of eq. (4), under the high-dimensional limit of eq. (5):

**Lemma F.1.** *For a constant $C = C(\kappa) > 0$:*

$$\left| \mathrm{MMSE}_d - \kappa \mathbb{E}_{\mathbf{S}^\star,\mathcal{D}} \mathrm{tr}\left[ \left(\mathbf{S}^\star - \hat{\mathbf{S}}^{\mathrm{BO}}\right)^2 \right] \right| \leq \frac{C(\kappa)}{n}.$$

Lemma F.1 shows that we can consider the MMSE on $\mathbf{S}$ equivalently to the generalization MMSE of eq. (4).

**Limits –** Notice that if the posterior concentrates around the true $\mathbf{W}^\star$, then $\hat{\mathbf{S}}^{\mathrm{BO}} = \mathbb{E}[\mathbf{S}|\mathcal{D}]$ concentrates on $\mathbf{S}^\star$, which implies that $\mathrm{MMSE}_d \to 0$. Conversely, for $\alpha = 0$ (i.e. in the absence of data), the Bayes-optimal estimator becomes $\hat{\mathbf{S}}^{\mathrm{BO}} = \mathbb{E}[\mathbf{S}^\star] = \mathrm{I}_d$, so that $\mathbb{E}\mathrm{tr}[(\mathbf{S}^\star - \hat{\mathbf{S}}^{\mathrm{BO}})^2] = \kappa^{-1}$. Thus, we have $\mathrm{MMSE}_d \to 1$ for $\alpha = 0$.

*Proof of Lemma F.1.* – Notice that (cf. eq. (2)):

$$\mathbb{E}_{\mathbf{z}}[f_{\mathbf{W}}(\mathbf{x})] = \Delta + \frac{\mathbf{x}^\mathsf{T}\mathbf{S}\mathbf{x}}{d},$$

with $\mathbf{S} := (1/m)\sum_{k=1}^m \mathbf{w}_k\mathbf{w}_k^\mathsf{T}$. Using this in eq. (3), and plugging it in eq. (4), we get (with $\mathbf{z} \sim \mathcal{N}(0, \mathrm{I}_m)$ and $\mathbf{x} \sim \mathcal{N}(0, \mathrm{I}_d)$):

$$\mathrm{MMSE}_d = \frac{m}{2}\mathbb{E}_{\mathbf{S}^\star,\mathcal{D},\mathbf{z},\mathbf{x}}\left[\left(\Delta\left(1 - \frac{\|\mathbf{z}\|^2}{m}\right) + \frac{\mathbf{x}^\mathsf{T}(\hat{\mathbf{S}}^{\mathrm{BO}} - \mathbf{S}^\star)\mathbf{x}}{d} - \frac{2\sqrt{\Delta}}{m}\sum_{k=1}^m z_k\left(\frac{\mathbf{x}^\mathsf{T}\mathbf{w}_k^\star}{\sqrt{d}}\right)\right)^2\right]$$

$$- \Delta(2 + \Delta),$$

$$= \frac{m}{2}\mathbb{E}_{\mathbf{S}^\star,\mathcal{D},\mathbf{z},\mathbf{x}}\left[\Delta^2\left(1 - \frac{\|\mathbf{z}\|^2}{m}\right)^2 + \frac{[\mathbf{x}^\mathsf{T}(\hat{\mathbf{S}}^{\mathrm{BO}} - \mathbf{S}^\star)\mathbf{x}]^2}{d^2} + \frac{4\Delta}{m}\mathrm{tr}(\mathbf{S}^\star)\right] - \Delta(2 + \Delta),$$

$$\overset{(a)}{=} \frac{m}{2}\mathbb{E}_{\mathbf{S}^\star,\mathcal{D},\mathbf{x}}\left[\frac{[\mathbf{x}^\mathsf{T}(\hat{\mathbf{S}}^{\mathrm{BO}} - \mathbf{S}^\star)\mathbf{x}]^2}{d^2}\right],$$

$$\overset{(b)}{=} \frac{m}{2}\mathbb{E}_{\mathbf{S}^\star,\mathcal{D}}\left[\left(\mathrm{tr}(\mathbf{S}^\star - \hat{\mathbf{S}}^{\mathrm{BO}})\right)^2\right] + \kappa\mathbb{E}_{\mathbf{S}^\star,\mathcal{D}}\,\mathrm{tr}\left[\left(\mathbf{S}^\star - \hat{\mathbf{S}}^{\mathrm{BO}}\right)^2\right], \tag{68}$$

where we used $\mathbb{E}[\|\mathbf{z}\|^4] = m^2 + 2m$ and $\mathbb{E}\mathrm{tr}(\mathbf{S}^\star) = 1$ in (a), and $\mathbb{E}_{\mathbf{x}\sim\mathcal{N}(0,\mathrm{I}_d)}[(\mathbf{x}^\mathsf{T}\mathbf{M}\mathbf{x})^2] = \mathrm{Tr}[\mathbf{M}]^2 + 2\mathrm{Tr}[\mathbf{M}^2]$ in (b). It remains to bound the first term of eq. (68) to conclude the proof of Lemma F.1. We notice that, by linearity of the trace, $\mathrm{tr}(\hat{\mathbf{S}}^{\mathrm{BO}})$ is the Bayes-optimal estimator for $\mathrm{tr}(\mathbf{S}^\star)$, i.e.

$$\mathbb{E}_{\mathbf{S}^\star,\mathcal{D}}\left[\left(\mathrm{tr}(\mathbf{S}^\star - \hat{\mathbf{S}}^{\mathrm{BO}})\right)^2\right] = \min_{r(\mathcal{D})}\mathbb{E}_{\mathbf{S}^\star,\mathcal{D}}\left[(\mathrm{tr}(\mathbf{S}^\star) - r(\mathcal{D}))^2\right]. \tag{69}$$

In particular, considering the estimator

$$r(\mathcal{D}) := \frac{1}{n}\sum_{i=1}^n (y_i - \Delta),$$

$$= \frac{1}{n}\sum_{i=1}^n \left\{\frac{\mathbf{x}_i\mathbf{S}^\star\mathbf{x}_i}{d} + \Delta\left(\frac{\|\mathbf{z}_i\|^2}{m} - 1\right) + \frac{2\sqrt{\Delta}}{m}\sum_{k=1}^m z_{i,k}\left(\frac{\mathbf{x}_i^\mathsf{T}\mathbf{w}_k^\star}{\sqrt{d}}\right)\right\},$$

we have using eq. (69):

$$\mathbb{E}_{\mathbf{S}^\star, \mathcal{D}}\left[\left(\mathrm{tr}(\mathbf{S}^\star - \hat{\mathbf{S}}^{\mathrm{BO}})\right)^2\right] \leq \mathbb{E}_{\mathbf{S}^\star, \{\mathbf{x}_i\}, \{\mathbf{z}_i\}}\left[\left\{\mathrm{tr}\left[\mathbf{S}^\star\left(\frac{1}{n}\sum_{i=1}^n \mathbf{x}_i\mathbf{x}_i^\mathsf{T} - \mathrm{I}_d\right)\right] + \Delta\left(\frac{\sum_{i=1}^n \|\mathbf{z}_i\|^2}{nm} - 1\right)\right.\right.$$
$$\left.\left. + \frac{2\sqrt{\Delta}}{nm}\sum_{i=1}^n\sum_{k=1}^m z_{i,k}\left(\frac{\mathbf{x}_i^\mathsf{T}\mathbf{w}_k^\star}{\sqrt{d}}\right)\right\}^2\right],$$

$$\overset{(a)}{\leq} 3[I_1 + I_2 + I_3], \tag{70}$$

using the Cauchy-Schwarz inequality in (a), with

$$\begin{cases} I_1 &:= \mathbb{E}\left[\left(\mathrm{tr}\left[\mathbf{S}^\star\left(\frac{1}{n}\sum_{i=1}^n \mathbf{x}_i\mathbf{x}_i^\mathsf{T} - \mathrm{I}_d\right)\right]\right)^2\right], \\[2mm] I_2 &:= \Delta^2\mathbb{E}\left[\left(\frac{\sum_{i=1}^n \|\mathbf{z}_i\|^2}{nm} - 1\right)^2\right], \\[2mm] I_3 &:= 4\Delta\mathbb{E}\left[\left(\frac{1}{nm}\sum_{i=1}^n\sum_{k=1}^m z_{i,k}\left(\frac{\mathbf{x}_i^\mathsf{T}\mathbf{w}_k^\star}{\sqrt{d}}\right)\right)^2\right]. \end{cases}$$

It is a tedious but straightforward computation to compute $\{I_a\}_{a=1}^3$, as it only involves the first moments of Gaussian random variables. We get (recall $m = \kappa d$):

$$\begin{cases} I_1 &= \frac{2}{nd}(1 + \kappa^{-1}), \\[2mm] I_2 &= \frac{2\Delta^2}{\kappa nd}, \\[2mm] I_3 &= \frac{4\Delta}{\kappa nd}. \end{cases} \tag{71}$$

Combining eqs. (70) and (71), and plugging it back in eq. (68), we get

$$\left|\mathrm{MMSE}_d - \kappa\mathbb{E}_{\mathbf{S}^\star, \mathcal{D}}\,\mathrm{tr}\left[\left(\mathbf{S}^\star - \hat{\mathbf{S}}^{\mathrm{BO}}\right)^2\right]\right| \leq \frac{C(\kappa)}{n},$$

which ends the proof of Lemma F.1. $\qquad\square$

## F.2 Proof of Theorem 4.1

First, we note that Theorem 1 of Pourkamali et al. [2024] implies that:

$$\Psi(\hat{q}) = \frac{1}{2}I_{\mathrm{HCIZ}}(\hat{q}, \mu_{\mathrm{MP},\kappa}, \mu_{1/\hat{q}}) - \frac{Q_0\hat{q}}{2}, \tag{72}$$

and we recall the definition of $I_{\mathrm{HCIZ}}$ in eq. (48). We recall then a fundamental result proven in Guionnet and Zeitouni [2002]:

**Theorem F.2** (Theorem 1.1 of Guionnet and Zeitouni [2002])**.** *For any compactly supported probability measures $\nu$ and $\mu$, and any $t > 0$:*

$$\frac{1}{2}I_{\mathrm{HCIZ}}(t^{-1}, \nu, \mu) = -J(\nu; \mu) - \frac{1}{2}\Sigma(\nu) + \frac{1}{4t}\mathbb{E}_\nu[X^2] - \frac{3}{8} + \frac{1}{4}\log t + \frac{1}{4t}\mathbb{E}_\mu[X^2]. \tag{73}$$

*Moreover, the function $J(\nu; \mu)$ satisfies the following property. Let $d$ be a distance on the space of probability measures on $\mathbb{R}$ that is compatible with the weak topology. Let $\mathbf{X} := \mathbf{R} + \sqrt{t}\mathbf{W}$, where $\mathbf{W} \sim \mathrm{GOE}(d)$, and $\mathbf{R}$ is a fixed (deterministic) matrix, with uniformly bounded spectral norm, and a compactly supported limiting eigenvalue distribution $\mu$. Let $\mu_{\mathbf{X}}$ denote the empirical eigenvalue distribution of $\mathbf{X}$. Then, for any $\nu \in \mathcal{M}_1^+(\mathbb{R})$:*

$$\lim_{\delta\downarrow 0}\limsup_{d\to\infty}\frac{1}{d^2}\log\mathbb{P}[d(\nu, \mu_{\mathbf{X}}) < \delta] = \lim_{\delta\downarrow 0}\liminf_{d\to\infty}\frac{1}{d^2}\log\mathbb{P}[d(\nu, \mu_{\mathbf{X}}) < \delta],$$
$$= -J(\nu; \mu). \tag{74}$$

In other words, the function $J(\nu; \mu)$ is the large deviations rate function (in the scale $d^2$) for the empirical spectral measure of $\mathbf{R} + \sqrt{t}\mathbf{W}$, where $\mathbf{R}$ is a fixed (deterministic) matrix with asymptotic spectral distribution $\mu$, and $\mathbf{W} \sim \mathrm{GOE}(d)$. It is a well-known property of the free convolution [Speicher, 1993] that $\mu_{\mathbf{X}} \to \mu \boxplus \sigma_{\mathrm{s.c.}, \sqrt{t}}$ as $d \to \infty$, where the convergence is meant in the weak sense (and almost surely). Combining this result with eq. (74), we have $J(\mu \boxplus \sigma_{\mathrm{s.c.}, \sqrt{t}}; \mu) = 0$. This yields by eq. (73):

$$I_{\mathrm{HCIZ}}(t^{-1}, \mu, \mu \boxplus \sigma_{\mathrm{s.c.}, \sqrt{t}}) = -\Sigma(\mu \boxplus \sigma_{\mathrm{s.c.}, \sqrt{t}}) + \frac{1}{2}\log t - \frac{1}{4} + \frac{1}{t}\mathbb{E}_\mu[X^2]. \qquad (75)$$

Combining eqs. (72) and (75) yields eq. (17). $\quad\square$

**Remark –** The proof above can be straightforwardly extended to the free entropy of denoising any matrix $\mathbf{S}$ with a rotationally-invariant distribution and a compactly-supported limiting eigenvalue distribution (beyond the Wishart ensemble), as the results of Guionnet and Zeitouni [2002], Pourkamali et al. [2024] hold under these more general assumptions.

### F.3 Perfect recovery threshold in the noiseless case

In this section, we give an analytic argument to derive the value of the perfect recovery threshold $\alpha_{\mathrm{PR}}$ (see eq. (1)) in the noiseless setting. In the limit of perfect recovery the MMSE goes to $0$, thus by eq. (7) (with $\Delta = 0$) this implies $\hat{q} \to \infty$. Using eq. (8), we can then write the equation satisfied by the perfect recovery threshold as

$$\frac{3(1 - 2\alpha_{\mathrm{PR}})}{4\pi^2} = \lim_{t \downarrow 0} t \int \mathrm{d}y\, \mu_t(y)^3, \qquad (76)$$

in which $\mu_t = \mu_{\mathrm{MP}, \kappa} \boxplus \sigma_{\mathrm{s.c.}, \sqrt{t}}$, see Appendix A.

#### F.3.1 The case $\kappa < 1$

**Informal argument –** Recall that in this case we can write $\mu_{\mathrm{MP}, \kappa}(x) = (1 - \kappa)\delta(x) + \kappa\nu_{\mathrm{MP}, \kappa}(x)$, in which $\nu_{\mathrm{MP}, \kappa}$ is compactly supported away from zero, see Appendix A. As $t \to 0$, we thus expect $\mu_t$ to have a discontinuous support, made of two parts:

(a) A small semicircular density centered around $0$, of radius $\mathcal{O}(\sqrt{t})$, with mass $(1 - \kappa)$.

(b) A smooth density, compactly supported away from zero, which has a well-defined limit as $t \to 0$, and a mass $\kappa$.

Because of the factor $t$ in the right-hand side of eq. (76), only the part $(a)$ will matter in the limit.

**Formal derivation –** We first rewrite by a change of variable

$$t \int \mathrm{d}y\, \mu_t(y)^3 = \int \mathrm{d}z\, [\sqrt{t}\mu_t(\sqrt{t}z)]^3.$$

It is clear that for all $x \neq 0$, we have $\mu_t(x) \to \kappa\nu_{\mathrm{MP}, \kappa}(x)$ as $t \to 0$, and $\int \nu_{\mathrm{MP}, \kappa}(y)^3 \mathrm{d}y < \infty$, so that we can truncate the integral above to all $|z| \leq \varepsilon/\sqrt{t}$, for any $\varepsilon > 0$ finite as $t \to 0$. We will now show the following, for any $x \in \mathbb{R}$:

$$\lim_{t \to 0} \sqrt{t}\mu_t(x\sqrt{t}) = (1 - \kappa)\sigma_{\mathrm{s.c.}, \sqrt{1-\kappa}}(x). \qquad (77)$$

We fix $z \in \mathbb{C}_+$ (where $\mathbb{C}_+ := \{z \in \mathbb{C} : \mathrm{Im}(z) > 0\}$). Letting $y = \sqrt{t}z$, we know from the Marchenko-Pastur theorem [Marchenko and Pastur, 1967] that $g_t(y) := \mathbb{E}_{\mu_t}[1/(X - y)]$ is the unique solution in $\mathbb{C}_+$ to the equation

$$y = \frac{1}{1 + g/\kappa} - \frac{1}{g} - tg.$$

Since $y = \sqrt{t}z$, it is clear that $g = \mathcal{O}(1/\sqrt{t})$, and letting $h := \sqrt{t}g$, we easily get the expansion

$$z = -\frac{1 - \kappa}{h} - h + \mathcal{O}(\sqrt{t}),$$

which can be inverted to

$$h = \frac{-z \pm \sqrt{z^2 - 4(1-\kappa)}}{2} + \mathcal{O}(\sqrt{t}). \tag{78}$$

Notice that if we denote $\mathcal{S}_\kappa(z)$ the Stieltjes transform of $\sigma_{\text{s.c.},\sqrt{1-\kappa}}$, eq. (78) can be written as (see e.g. Anderson et al. [2010]) $h = (1-\kappa)\mathcal{S}_\kappa(z) + \mathcal{O}(\sqrt{t})$. By considering $z = x + i\varepsilon$ with $x \in \mathbb{R}$ and the limit $\varepsilon \to 0$, we reach using the Stieltjes-Perron inversion theorem (Theorem A.2) that for any $x \in \mathbb{R}$:

$$\lim_{t\to 0} \sqrt{t}\mu_t(x\sqrt{t}) = (1-\kappa)\sigma_{\text{s.c.},\sqrt{1-\kappa}}(x). \tag{79}$$

Coming back to eq. (76) this implies:

$$\frac{3(1-2\alpha_{\text{PR}})}{4\pi^2} = \lim_{t\to 0} t \int \mathrm{d}y\,\mu_t(y)^3,$$

$$= (1-\kappa)^3 \int \mathrm{d}y\,\sigma_{\text{s.c.},\sqrt{1-\kappa}}(y)^3,$$

$$= (1-\kappa)^2 \int \mathrm{d}y\,\sigma_{\text{s.c.}}(y)^3,$$

$$= \frac{3}{4\pi^2}(1-\kappa)^2.$$

Equivalently:

$$\alpha_{\text{PR}} = \frac{(1-\kappa)^2 - 1}{2} = \kappa - \frac{\kappa^2}{2}. \tag{80}$$

We notice that this critical value of $n/d^2$ coincides with a naive counting argument of degrees of freedom of $\mathbf{S}^\star$. Indeed, as can be seen by the spectral decomposition, the set of $d \times d$ symmetric matrices of rank $m$ has, to leading order in $d$, $p(\kappa)d^2$ degrees of freedom, where $p(\kappa)d^2$ is the dimension of the Stiefel manifold of orthonormal $m$-frames in $\mathbb{R}^d$. It is well-known that $p(\kappa) = \kappa - \kappa^2/2$ for $d \to \infty$ [Helmke and Moore, 2012].

### F.3.2  The case $\kappa \geq 1$

The case $\kappa > 1$ is simpler to carry out. In this case, $\mu_{\text{MP},\kappa}$ does not have a singular part at $x = 0$, and $\mu_t$ has a smooth density as $t \to 0$, and

$$\int \mathrm{d}y\,\mu_{\text{MP},\kappa}(y)^3 = \frac{3}{4\pi^2}\frac{\kappa^2}{\kappa - 1},$$

so that

$$\frac{3(1-2\alpha_{\text{PR}})}{4\pi^2} = \lim_{t\downarrow 0} t \int \mathrm{d}y\,\mu_t(y)^3 = 0,$$

and we reach $\alpha_{\text{PR}} = 1/2$, so that $\alpha_{\text{PR}}d^2$ (asymptotically) coincides with the number $d^2/2$ of degrees of freedom of symmetric matrices. Since $\alpha_{\text{PR}}$ is increasing with $\kappa$, and has limit $1/2$ both for $\kappa \uparrow 1$ and $\kappa \downarrow 1$, we deduce that $\alpha_{\text{PR}} = 1/2$ for $\kappa = 1$ as well.

### F.4  The derivative of the error at the perfect recovery threshold

Here, we extend the derivation of Section F.3 to compute the derivative of the MMSE with respect to $\alpha$ at the perfect recovery threshold. We start again from eqs. (7) and (8). Letting $t := 1/\hat{q}$, we get, with $\alpha = \alpha_{\text{PR}}$:

$$\left(\frac{\partial \text{MMSE}}{\partial \alpha}\right)_{\text{PR}} = 2\alpha\kappa \left(\frac{\partial t}{\partial \alpha}\right)_{\text{PR}},$$

$$= -\frac{3\alpha\kappa}{\pi^2}\left[\lim_{t\to 0}\partial_t\left(t\int\mu_t(y)^3\mathrm{d}y\right)\right]^{-1}. \tag{81}$$

We thus compute the next order of the expansion of $t\int\mu_t(y)^3\mathrm{d}y$ as $t \to 0$.

### F.4.1 The case $\kappa < 1$

We extend the argument made in Section F.3.1. Notice that here the smooth part of the density, compactly supported away from zero, contributes at this order. Formally, for any small enough $\varepsilon > 0$:

$$t \int_{|y|\geq\varepsilon} \mathrm{d}y\, \mu_t(y)^3 = t\kappa^3 \int \mathrm{d}y\, \nu_{\mathrm{MP},\kappa}(y)^3 + o_t(t),$$

$$= \frac{3t\kappa^4}{4\pi^2(1-\kappa)} + o_t(t), \tag{82}$$

On the other hand, we have around the singularity at $y = 0$:

$$t \int_{|y|\leq\varepsilon} \mathrm{d}y\, \mu_t(y)^3 = \int_{|z|\leq\varepsilon/\sqrt{t}} \mathrm{d}z\, [\sqrt{t}\mu_t(\sqrt{t}z)]^3. \tag{83}$$

We evaluate the next order of the right-hand side of eq. (83) using the same approach as in Section F.3.1, going to next orders in the expansion as $t \to 0$ of eq. (78). Using then again the Stieltjes-Perron inversion theorem, we reach with tedious but straightforward computations the generalization of eq. (79):

$$\sqrt{t}\mu_t(\sqrt{t}z) = (1-\kappa)\sigma_{\mathrm{s.c.},\sqrt{1-\kappa}}(z) - \sqrt{t}\frac{z\kappa^2}{2\pi(1-\kappa)\sqrt{4(1-\kappa)-z^2}}$$

$$+ t\frac{\kappa^3\left[z^4 - 6z^2(1-\kappa) + 2(4-\kappa)(1-\kappa)^2\right]}{2\pi(1-\kappa)^3[4(1-\kappa)-z^2]^{3/2}} + \mathcal{O}(t^{3/2}), \tag{84}$$

for any $|z| \leq 2\sqrt{1-\kappa}$, while $\sqrt{t}\mu_t(\sqrt{t}z) = \mathcal{O}(t^{3/2})$ if $|z| > 2\sqrt{1-\kappa}$. This then yields:

$$t \int_{|y|\leq\varepsilon} \mathrm{d}y\, \mu_t(y)^3 = \frac{3(1-\kappa)^2}{4\pi^2} + \frac{3t\kappa^3}{4\pi^2(1-\kappa)} + \mathcal{O}(t^{3/2}). \tag{85}$$

Combining eqs. (82) and (85) in eq. (81), we obtain (recall $\alpha = \alpha_{\mathrm{PR}} = \kappa - \kappa^2/2$):

$$\left(\frac{\partial\mathrm{MMSE}}{\partial\alpha}\right)_{\mathrm{PR}} = -2 - \frac{4}{\kappa} + \frac{12}{1+\kappa}.$$

### F.4.2 The case $\kappa \geq 1$

Again, we consider $\kappa > 1$. The argument of Section F.4.1 generalizes immediately, removing the analysis of the singular part around $y = 0$. We get directly

$$t \int \mathrm{d}y\, \mu_t(y)^3 = t \int \mathrm{d}y\, \mu_{\mathrm{MP},\kappa}(y)^3 + o_t(t),$$

$$= \frac{3t\kappa^2}{4\pi^2(\kappa-1)} + o_t(t). \tag{86}$$

Plugging it in eq. (81), we get in this case:

$$\left(\frac{\partial\mathrm{MMSE}}{\partial\alpha}\right)_{\mathrm{PR}} = -2 + \frac{2}{\kappa}.$$

Again, the specific case $\kappa = 1$ can be tackled by continuity, as the derivative tends to 0 both as $\kappa \uparrow 1$ and $\kappa \downarrow 1$.

### F.5 Details on the reduction to matrix estimation

We describe here how to effectively reduce the problem of eq. (2) to an estimation problem in terms of $\mathbf{S}^\star := (1/m)\sum_{k=1}^m \mathbf{w}_k^\star(\mathbf{w}_k^\star)^\intercal$.

**Remark –** While our argument is backed by precise probabilistic concentration arguments, we notice that it is not a proof of the equivalence of the problems of eq. (2) and eq. (9) under all statistical tests, as would be implied e.g. by the contiguity of distributions [Le Cam, 1960, Kunisky et al., 2019]. Rather, we analyze the leading order of eq. (2) and argue that (with high probability over

the distribution of the data and the teacher weights), the first non-trivial order of the observations is characterized by the equivalent model of eq. (9). Notably, we do not claim the statistical equivalence of the problems of eq. (2) and eq. (9), but rather only that their asymptotic MMSEs coincide. While even this weaker statement is not formally implied by the arguments sketched below, we expect that they form the backbone of a formal proof of this claim, which we leave for future work and would be carried e.g. by Gaussian interpolation techniques.

Let us define $\mathbf{Z}_i := (\mathbf{x}_i \mathbf{x}_i^\mathsf{T} - \mathrm{I}_d)/\sqrt{d}$, and recall that $\mathbf{x}_i \sim \mathcal{N}(0, \mathrm{I}_d)$. Expanding the square, we can rewrite the law of the output $y_i = f_{\mathbf{W}^\star}(\mathbf{x}_i)$ as

$$y_i = \Delta + \mathrm{tr}[\mathbf{S}^\star] + \frac{1}{\sqrt{d}}\mathrm{Tr}[\mathbf{Z}_i\mathbf{S}^\star] + \Delta\left(\frac{\|\mathbf{z}_i\|^2}{m} - 1\right) + \frac{2\sqrt{\Delta}}{m\sqrt{d}}\sum_{k=1}^m z_{i,k}\mathbf{x}_i^\mathsf{T}\mathbf{w}_k^\star, \qquad (87)$$

where $(\mathbf{z}_i)_{i=1}^n \overset{\text{i.i.d.}}{\sim} \mathcal{N}(0, \mathrm{I}_m)$. In what follows, we analyze the leading order of eq. (87). More specifically, we denote $\widetilde{y}_i := \sqrt{d}(y_i - 1 - \Delta)$, and we decompose

$$\widetilde{y}_i = \mathrm{Tr}[\mathbf{Z}_i\mathbf{S}^\star] + \underbrace{\sqrt{d}(\mathrm{tr}[\mathbf{S}^\star] - 1)}_{=:I_1} + \underbrace{\Delta\sqrt{d}\left(\frac{\|\mathbf{z}_i\|^2}{m} - 1\right) + \frac{2\sqrt{\Delta}}{m}\sum_{k=1}^m z_{i,k}\mathbf{x}_i^\mathsf{T}\mathbf{w}_k^\star}_{=:I_2}. \qquad (88)$$

Let us consider the leading order of the different terms of eq. (88). Since $\mathbf{S}^\star \sim \mathcal{W}_{m,d}$, $\mathrm{tr}[\mathbf{S}^\star] = \sum_{k=1}^m \|\mathbf{w}_k^\star\|^2/m$ strongly concentrates on its average. More precisely, by Bernstein's inequality (see Corollary 2.8.3 of Vershynin [2018]) we have, for all $t \geq 0$:

$$\mathbb{P}[|\mathrm{tr}(\mathbf{S}^\star) - 1| \geq t] \leq 2\exp\left(-Cd^2\min(t, t^2)\right),$$

where $C > 0$ depends only on $\kappa > 0$. In particular,

$$\mathbb{P}[|I_1| \geq d^{-1/4}] \leq 2\exp(-C\sqrt{d}),$$

so that we can replace $I_1$ by 0 at leading order in eq. (88).

We now tackle $I_2$, first for fixed $(\mathbf{x}_i, \mathbf{W}^\star)$. Using that $\|\mathbf{z}_i\|^2$ strongly concentrates around its average, and the central limit theorem applied to the fluctuations of $\|\mathbf{z}_i\|^2$, one can see that for all $i \in [n]$, we have (with $\mathbf{g}_i \sim \mathcal{N}(0, \mathrm{I}_m)$ independently of $\mathbf{z}_i$, and $\overset{\mathrm{d}}{=}$ denoting equality in distribution):

$$\sqrt{d}\left[\Delta\left(\frac{\|\mathbf{z}_i\|^2}{m} - 1\right) + \frac{2\sqrt{\Delta}}{m\sqrt{d}}\sum_{k=1}^m z_{i,k}\mathbf{x}_i^\mathsf{T}\mathbf{w}_k^\star\right]$$

$$\overset{\mathrm{d}}{=} \sqrt{d}\left[\Delta\left(\frac{\|\mathbf{z}_i\|^2}{m} - 1\right) + \frac{2\sqrt{\Delta}}{m\sqrt{d}}\frac{\|\mathbf{z}_i\|}{\|\mathbf{g}_i\|}\sum_{k=1}^m g_{i,k}\mathbf{x}_i^\mathsf{T}\mathbf{w}_k^\star\right],$$

$$\sim_{d\to\infty} \xi_i\sqrt{\frac{2\Delta^2}{\kappa}\frac{\mathbf{x}_i^\mathsf{T}\mathbf{S}^\star\mathbf{x}_i}{d} + \frac{4\Delta}{\kappa}}, \qquad (89)$$

with $\xi_i \overset{\text{i.i.d.}}{\sim} \mathcal{N}(0, 1)$, independently of $(\mathbf{x}_i, \mathbf{w}_k^\star)$. The equivalence as $d \to \infty$ is given for a fixed $i \in [n]$: coherently with the remark above, we notice that a formal mathematical proof of equivalence of the two problems of eq. (2) and eq. (9) would rather need to tackle the joint law of all the observations, and to quantitatively control the deviation between the left and right-hand sides of eq. (89) as $d \to \infty$. We leave such a proof for future work.

We finally note that the variance term on the right-hand side of eq. (89) strongly concentrates, uniformly in $i \in [n]$, as by the Hanson-Wright inequality and the union bound, we have (see Theorem 6.2.1 of Vershynin [2018]) for all $t \geq 0$:

$$\mathbb{P}_{\{\mathbf{x}_i\}}\left[\left|\frac{1}{d}\max_{i\in[n]}|\mathbf{x}_i^\mathsf{T}\mathbf{S}^\star\mathbf{x}_i - \mathrm{tr}(\mathbf{S}^\star)|\right| \geq t\right] \leq 2n\exp\left[-C\min\left(\frac{dt^2}{\|\mathbf{S}^\star\|_{\mathrm{op}}^2}, \frac{dt}{\|\mathbf{S}^\star\|_{\mathrm{op}}}\right)\right], \qquad (90)$$

for some constant $C > 0$. Since the spectral norm of a Wishart matrix $\|\mathbf{S}^\star\|_{\mathrm{op}}$ strongly concentrates on its average under the Wishart distribution (see Theorem 4.4.5 of Vershynin [2018]), we see that, uniformly over $i \in [n]$, the leading order of the variance in the right-hand side of eq. (89) is equal to $\widetilde{\Delta} := 2\Delta(2 + \Delta)/\kappa$. This ends our justification of eq. (9).

## F.6 Unique maximizer $q^\star$ in eq. (12)

Notice that if $J_{\text{out}}(q) := \int_{\mathbb{R} \times \mathbb{R}} \mathrm{d}y \mathcal{D}\xi \, J_q(y, \xi) \log J_q(y, \xi)$, then one can check that $J_{\text{out}}$ is a strictly increasing function of $q$ under mild regularity conditions on $P_{\text{out}}$ (namely assuming the presence of an additive Gaussian noise with arbitrarily small variance), see Proposition 21 of Barbier et al. [2019]. The fact that $q^\star$ is uniquely defined for all values of $\alpha > 0$ except possibly in a countable set follows then from Proposition 1 of Barbier et al. [2019], see also Appendix A.2 there.

## F.7 Derivation of Result 1 from Claim 2

In this section, we derive eqs. (7) and eq. (8) from Claim 2, in the case of Gaussian noise. More precisely, we assume $P_{\text{out}}(y|z) = \exp[-(y - z)^2/(2\widetilde{\Delta})]/\sqrt{2\pi\widetilde{\Delta}}$, in accordance with eq. (9). It is then an easy computation to check (recall the definition of $J_q$ in eq. (13)):

$$\int_{\mathbb{R} \times \mathbb{R}} \mathrm{d}y \mathcal{D}\xi \, J_q(y, \xi) \log J_q(y, \xi) = -\frac{1}{2} \log[\widetilde{\Delta} + 2(Q_0 - q)].$$

We then reach that $q = q^\star$ is characterized as the maximum of the following function:

$$\begin{cases} F(q) &= I(q) - \dfrac{\alpha}{2} \log[\widetilde{\Delta} + 2(Q_0 - q)], \\ I(q) &:= \inf\limits_{\hat{q} \geq 0} \left[ \dfrac{(Q_0 - q)\hat{q}}{4} - \dfrac{1}{2}\Sigma(\mu_{1/\hat{q}}) - \dfrac{1}{4}\log\hat{q} - \dfrac{1}{8} \right]. \end{cases} \tag{91}$$

Recall that here $\mu_t := \mu_{\text{MP},\kappa} \boxplus \sigma_{\text{s.c.},\sqrt{t}}$. It is known (see eqs. (77-78) of Semerjian [2024] e.g.) that

$$\frac{\partial \Sigma(\mu_t)}{\partial t} = \frac{2\pi^2}{3} \int \mu_t(y)^3 \mathrm{d}y.$$

Thus, $\hat{q} = \hat{q}(q)$ can be characterized as the solution[4] to

$$\frac{(Q_0 - q)}{4} + \frac{\pi^2}{3\hat{q}^2} \int \mu_{1/\hat{q}}(y)^3 \mathrm{d}y - \frac{1}{4\hat{q}} = 0. \tag{92}$$

By eq. (91), $q$ is a solution in $[1, Q_0]$ to:

$$\hat{q}(q) = \frac{4\alpha}{\widetilde{\Delta} + 2(Q_0 - q)}. \tag{93}$$

Recalling that $\text{MMSE} = \kappa(Q_0 - q)$ by Claim 2, eq. (93) implies eq. (7). Combining eq. (93) with eq. (92), we reach eq. (8).

## F.8 The limit $\alpha \to 0$

In this section, we check that the state evolution equations derived in Section F.7 yield indeed that $q \to 1$ as $\alpha \to 0$. Indeed, in this limit, $\mathbf{S}^{\text{BO}} = \mathbb{E}[\mathbf{S}^\star] = \mathbf{I}_d$, so that we must have $q = \mathbb{E}\text{tr}[\mathbf{S}^{\text{BO}}\mathbf{S}^\star] = 1$.

Recall that $\hat{q} = 4\alpha/[\widetilde{\Delta} + 2(Q_0 - q)]$, and that $\hat{q}$ is given by eq. (8). In particular, $\hat{q} \to 0$ as $\alpha \to 0$. Assuming the scaling $\hat{q} \sim \hat{q}_0 \alpha$ as $\alpha \to 0$, we get

$$\begin{cases} q &= Q_0 - \dfrac{2}{\hat{q}_0} + \dfrac{\widetilde{\Delta}}{2}, \\ -2 + \dfrac{\widetilde{\Delta}\hat{q}_0}{2} &= \hat{q}_0 F'(0), \end{cases} \tag{94}$$

where $F(p) := (4\pi^2/3) \int [p^{-1/2}\mu_{1/p}(z \cdot p^{-1/2})]^3 \mathrm{d}z$. Letting $\nu_p(z) := p^{-1/2}\mu_{1/p}(z \cdot p^{-1/2})$, we know by a similar reasoning as the one of Section F.3 that the Stieltjes transform $h = h_p(z)$ of $\nu_p$ satisfies the equation:

$$z = \frac{\kappa\sqrt{p}}{\kappa + h\sqrt{p}} - \frac{1}{h} - h.$$

---

[4]Notice that one can show that $\hat{q}$ is the minimizer of a convex function in eq. (91). This can be shown e.g. by recalling the relationship of this function to the free entropy of a matrix denoising problem (Theorem 4.1) and using the I-MMSE theorem. We refer to Barbier et al. [2019], Maillard et al. [2020] for more details.

As $p \to 0$, we can thus compute the expansion of $h_p(z)$ in powers of $p$. Applying then the Stieltjes-Perron inversion theorem (Theorem A.2), we get the expansion of $\nu_p(z)$ in powers of $p$ as:

$$\nu_p(z) = \frac{\sqrt{4-z^2}}{2\pi} + \sqrt{p}\,\frac{3z\sqrt{4-z^2}}{8\pi^3} - \frac{3p(2-z^2)(4+\kappa-z^2)}{8\pi^3\kappa\sqrt{4-z^2}} + \mathcal{O}(p^{3/2}),$$

for $|z| \leq 2$, and $\nu_p(z) = \mathcal{O}(p^{3/2})$ for $|z| \geq 2$. Plugging this expansion into $F(p)$, we get:

$$F(p) = 1 - \frac{p}{\kappa} + o(p).$$

Coming back to eq. (94), this gives $\hat{q}_0 = 4\kappa/[2 + \widetilde{\Delta}\kappa]$, and (recall $Q_0 = 1 + \kappa^{-1}$) then $q = 1$, so that our equations are indeed consistent in the limit $\alpha \to 0$.

# G  Learning the second layer weights

We sketch here in a mathematically informal way the generalization of our results to the setting where the second layer weights are also learned. The second layer weights $(a_k^\star)_{k=1}^m$ are drawn i.i.d. from a probability distribution $P_a$, and the student must learn $(\mathbf{w}_k^\star, a_k^\star)_{k=1}^m$ from the observation of $\{\mathbf{x}_i\}_{i=1}^n$ and of

$$y_i = \frac{1}{m}\sum_{k=1}^m a_k^\star \left[\frac{1}{\sqrt{d}}(\mathbf{w}_k^*)^\mathsf{T}\mathbf{x}_i + \sqrt{\Delta}z_{i,k}\right]^2. \tag{95}$$

In the rest of this paper we focused on the case $P_a = \delta_1$. However, all our techniques and results can be generalized to more generic choices of $P_a$, as we know show: in particular, Claim 3 is the generalization of Claim 2 to this more general setting.

Throughout this section, we will assume for simplicity that $P_a$ has bounded support, although we expect our results to hold also for more general choices of $P_a$. We show how to extend Claim 2 to this case, by detailing the differences in the steps outlined in Section 4. We eventually show that Algorithm 1 can also be straightforwardly extended to this setting as well.

## G.1  Generalizing the derivation

### G.1.1  Reduction to matrix estimation

We first discuss the reduction to a matrix estimation problem, generalizing Section F.5 to this setting. We define

$$\mathbf{S}^\star := \frac{1}{m}\sum_{k=1}^m a_k^\star \mathbf{w}_k^\star (\mathbf{w}_k^\star)^\mathsf{T}, \tag{96}$$

and we denote $m_a := \mathbb{E}_{P_a}[a]$ and $c_a := \mathbb{E}_{P_a}[a^2]$. We define the MMSE as (notice the additional factor $c_a$ with respect to eq. (4)):

$$\text{MMSE}_d := \frac{m}{2}\mathbb{E}_{\mathbf{W}^*,\mathcal{D}}\mathbb{E}_{y_{\text{test}},\mathbf{x}_{\text{test}}}\left[\left(y_{\text{test}} - \hat{y}_{\mathcal{D}}^{\text{BO}}(\mathbf{x}_{\text{test}})\right)^2\right] - \Delta(2 + c_a\Delta). \tag{97}$$

By repeating the (mathematically informal) arguments of Section F.5 to this setting, we find that, at leading order as $m, d \to \infty$:

$$\sqrt{d}(y_i - \Delta - \text{tr}[\mathbf{S}^\star]) = \text{Tr}[\mathbf{Z}_i\mathbf{S}^\star] + \sqrt{\widetilde{\Delta}}\xi_i, \tag{98}$$

with $\xi_i \overset{\text{i.i.d.}}{\sim} \mathcal{N}(0,1)$, and $\widetilde{\Delta} := 2\Delta(2 + \Delta c_a)/\kappa$. We let

$$\begin{cases} \widetilde{y}_i & := \sqrt{d}\left[y_i - \frac{1}{n}\sum_{j=1}^n y_j\right], \\ Y & := \frac{1}{n}\sum_{i=1}^n y_i. \end{cases}$$

The observation of $(y_i)_{i=1}^n$ is equivalent to the one of $(\widetilde{y}_i)_{i=1}^n$ and $Y$. Notice that by eq. (98), we have

$$|Y - \Delta - \text{tr}(\mathbf{S}^\star)| = \frac{1}{n\sqrt{d}}\left|\sum_{i=1}^n \{\text{Tr}[\mathbf{Z}_i\mathbf{S}^\star] + \sqrt{\widetilde{\Delta}}\xi_i\}\right|. \tag{99}$$

Conditionally on $\mathbf{S}^\star$, the right-hand-side of eq. (99) is a sum of $n$ independent zero-mean random variables, which thus typically fluctuates in the scale[5] $\mathcal{O}[(nd)^{-1/2}] = \mathcal{O}(d^{-3/2})$. Since $\widetilde{y}_i = \sqrt{d}[y_i - Y]$, this implies that at leading order we have

$$\widetilde{y}_i = \text{Tr}[\mathbf{Z}_i\mathbf{S}^\star] + \sqrt{\widetilde{\Delta}}\xi_i. \tag{100}$$

The observer also has access to $Y$, alongside $\{\widetilde{y}_i\}_{i=1}^n$. Notice that by the argument above, $Y$ is (up to order $d^{-3/2}$) a *deterministic* observation of $\text{tr}[\mathbf{S}^\star]$. By eq. (97), and repeating the arguments of the proof of Lemma F.1, we reach that again we have $\text{MMSE} = \kappa\mathbb{E}\text{tr}[(\mathbf{S}^\star - \hat{\mathbf{S}}^{\text{BO}})^2]$ as $d \to \infty$. Moreover:

$$\text{MMSE} = \kappa\mathbb{E}_{\mathbf{S}^\star,Y,\{\widetilde{y}_i\}}\text{tr}[(\mathbf{S}^\star - \hat{\mathbf{S}}^{\text{BO}})^2],$$
$$= \kappa\mathbb{E}_Y[\mathbb{E}_{\mathbf{S}^\star,\{\widetilde{y}_i\}}(\text{tr}[(\mathbf{S}^\star - \hat{\mathbf{S}}^{\text{BO}})^2]|Y)].$$

Conditioning on $Y$ amounts to condition on the value of $\text{tr}(\mathbf{S}^\star)$, as detailed above. Let us make two important remarks:

(i) As $d \to \infty$, $Y$ concentrates around its typical value $\mathbb{E}[Y] = m_a$. Since the MMSE is bounded, we therefore have as $d \to \infty$ that $\text{MMSE} = \kappa\mathbb{E}_{\mathbf{S}^\star,\{\widetilde{y}_i\}}(\text{tr}[(\mathbf{S}^\star - \hat{\mathbf{S}}^{\text{BO}})^2]|Y = m_a)$.

(ii) As we will see in what follows (and exactly like in the case of fixed second layer), the leading order of the MMSE of the inference problem of eq. (100) only depends on the *asymptotic spectral distribution* of $\mathbf{S}^\star$. In particular, at leading order:

$$\text{MMSE} = \kappa\mathbb{E}_{\mathbf{S}^\star,\{\widetilde{y}_i\}}(\text{tr}[(\mathbf{S}^\star - \hat{\mathbf{S}}^{\text{BO}})^2]|Y = m_a),$$
$$= \kappa\mathbb{E}_{\mathbf{S}^\star,\{\widetilde{y}_i\}}(\text{tr}[(\mathbf{S}^\star - \hat{\mathbf{S}}^{\text{BO}})^2]|\text{tr}(\mathbf{S}^\star) = m_a),$$
$$\overset{(a)}{=} \kappa\mathbb{E}_{\mathbf{S}^\star,\{\widetilde{y}_i\}}(\text{tr}[(\mathbf{S}^\star - \hat{\mathbf{S}}^{\text{BO}})^2]), \tag{101}$$

where in (a) we used that conditioning on $\text{tr}(\mathbf{S}^\star) = m_a$ does not change the asymptotic spectral distribution of $\mathbf{S}^\star$.

All in all, we focus on characterizing the MMSE given in eq. (101), for the inference problem of recovering $\mathbf{S}^\star$ from the knowledge of $\{\mathbf{Z}_i, y_i\}$ generated by eq. (100).

### G.1.2 Further steps of the derivation

Here, we notice that the arguments detailed in Section 4 on how to obtain an asymptotic expression of eq. (101) do not depend on the specific asymptotic spectral distribution of $\mathbf{S}^\star$. More precisely:

**A.** Conjecture 4.1 can be directly extended to more general distributions of $\mathbf{S}^\star$ than the Wishart distribution. Indeed, the heuristic argument explaining this universality phenomenon does not depend on the distribution of $\mathbf{S}^\star$, and on a technical level, as mentioned in the main text, Conjecture 4.1 is an extension of Corollary 4.10 of Maillard and Bandeira [2023], which holds for generic choices of distributions of matrices.

**B.** Conjecture 4.2 is also straightforwardly extended here, simply replacing the Wishart prior by the more generic prior of eq. (96). More generally, we expect it to hold for any prior such that the function $\Psi(\hat{q})$ of eq. (16) is well-defined [Aubin et al., 2019a, 2020].

**C.** Finally, the proof of Theorem 4.1 (see Appendix F.2) relies solely on the *rotation invariance* of the distribution of $\mathbf{S}^\star$, as well as the fact that $\mathbf{S}^\star$ admits a *compactly supported* asymptotic eigenvalue distribution. These two facts hold for the distribution of eq. (96) for compactly supported $P_a$, see e.g. Silverstein and Choi [1995], Lee and Schnelli [2016].

---

[5]Recall that $\text{tr}[(\mathbf{S}^\star)^2] = \mathcal{O}(1)$ with high probability.

## G.2 Conclusion: Claim 2 when learning the second layer

We are now ready to state the generalization of Claim 2 to a learnable second layer. The effective problem we consider is the recovery of a symmetric matrix $\mathbf{S}^\star \in \mathbb{R}^{d \times d}$, which was generated as $\mathbf{S}^\star = (1/m) \sum_{k=1}^m a_k^\star \mathbf{w}_k^\star (\mathbf{w}_k^\star)^\intercal$, from observations $(y_i)_{i=1}^n$, generated as

$$y_i \sim P_{\text{out}}\left(\cdot | \text{Tr}[\mathbf{Z}_i \mathbf{S}^\star]\right), \tag{102}$$

with $\mathbf{Z}_i := (\mathbf{x}_i \mathbf{x}_i^\intercal - \text{I}_d)/\sqrt{d}$ and $\mathbf{x}_i \overset{\text{i.i.d.}}{\sim} \mathcal{N}(0, \text{I}_d)$.

The asymptotic spectral distribution $\mu^\star$ of $\mathbf{S}^\star$ is called a *generalized Marchenko-Pastur distribution* (or a free compound Poisson distribution: it is also the free multiplicative convolution of the Marchenko-Pastur law and $P_a$, see Anderson et al. [2010]). $\mu^\star$ is compactly supported, and can be characterized by its $\mathcal{R}$ transform [Marchenko and Pastur, 1967, Silverstein and Choi, 1995, Tulino and Verdú, 2004]:

$$\mathcal{R}_{\mu^\star}(s) = \int \frac{\kappa a}{\kappa - sa} P_a(a) \mathrm{d}a. \tag{103}$$

Eq. (103) allows for an efficient numerical evaluation of $\mu^\star$ given $P_a$. Notice that $\mathbb{E}_{\mu^\star}[X] = m_a$, and $\mathbb{E}_{\mu^\star}[X^2] = m_a^2 + c_a/\kappa$.

The partition function for the learning problem of eq. (102) is again defined as:

$$\mathcal{Z}(\{y_i, \mathbf{x}_i\}_{i=1}^n) := \mathbb{E}_{\mathbf{S}} \prod_{i=1}^n P_{\text{out}}\left(y_i | \text{Tr}[\mathbf{S} \mathbf{Z}_i]\right). \tag{104}$$

We then obtain the following generalization of Claim 2.

**Claim 3.** *Assume that $m = \kappa d$ with $\kappa > 0$, and $n = \alpha d^2$ with $\alpha > 0$. Recall that $m_a := \mathbb{E}_{P_a}[a]$ and $c_a := \mathbb{E}_{P_a}[a^2]$. Let $Q_0 := \mathbb{E}_{\mu^\star}[X^2] = m_a^2 + c_a/\kappa$. Then:*

- *The limit of the averaged log-partition function of eq. (104) is given by*

$$\lim_{d \to \infty} \frac{1}{d^2} \mathbb{E}_{\{y_i, \mathbf{x}_i\}} \log \mathcal{Z} = \sup_{q \in [m_a^2, Q_0]} \left[ I(q) + \alpha \int_{\mathbb{R} \times \mathbb{R}} \mathrm{d}y \mathcal{D}\xi \, J_q(y, \xi) \log J_q(y, \xi) \right], \tag{105}$$

*where*

$$\begin{cases} I(q) & := \inf_{\hat{q} \geq 0} \left[ \frac{(Q_0 - q)\hat{q}}{4} - \frac{1}{2}\Sigma(\mu_{1/\hat{q}}) - \frac{1}{4}\log \hat{q} - \frac{1}{8} \right], \\ J_q(y, \xi) & := \int \frac{\mathrm{d}z}{\sqrt{4\pi(Q_0 - q)}} \exp\left\{ -\frac{(z - \sqrt{2q}\xi)^2}{4(Q_0 - q)} \right\} P_{\text{out}}(y|z). \end{cases} \tag{106}$$

*Here, $\Sigma(\mu) := \mathbb{E}_{X, Y \sim \mu} \log |X - Y|$, and, for $t \geq 0$, $\mu_t := \mu^\star \boxplus \sigma_{\text{s.c.}, \sqrt{t}}$ is the free convolution of $\mu^\star$ and a (scaled) semicircle law (see Appendix A).*

- *For any $\alpha > 0$, except possibly in a countable set, the supremum in eq. (105) is reached in a unique $q^\star \in [m_a^2, Q_0]$. Moreover, the asymptotic minimum mean-squared error on the estimation of $\mathbf{S}^\star$, achieved by the Bayes-optimal estimator $\hat{\mathbf{S}}^{\text{BO}} := \mathbb{E}[\mathbf{S}|\{y_i, \mathbf{x}_i\}]$, is equal to $Q_0 - q^\star$:*

$$\lim_{d \to \infty} \mathbb{E}\text{tr}[(\mathbf{S}^\star - \hat{\mathbf{S}}^{\text{BO}})^2] = Q_0 - q^\star. \tag{107}$$

*It is related to the MMSE of eq. (97) by MMSE $= \kappa(Q_0 - q^\star)$.*

Therefore, generalizing Section F.7, Result 1 holds as well in this case, with $\widetilde{\Delta} = 2\Delta(2 + c_a\Delta)/\kappa$, and $\mu_t := \mu^\star \boxplus \sigma_{\text{s.c.}, \sqrt{t}}$, where $\mu^\star$ is characterized by eq. (103).

## G.3 The GAMP-RIE algorithm

Finally, one can also generalize Algorithm 1 to this setting: the only change to perform is to adapt the functions $F_{\text{RIE}}$ and $f_{\text{RIE}}$. Indeed, instead of denoising a Wishart matrix (with an asymptotic spectrum given by the Marchenko-Pastur distribution), here one must denoise a matrix $\mathbf{S}_0$ with asymptotic spectral distribution given by $\mu^\star$ defined in Appendix G.2. As mentioned, eq. (103) allows for an efficient numerical evaluation of $\mu^\star$ given $P_a$. From there, one can adapt Algorithm 1 to this case simply by replacing in the definitions of $F_{\text{RIE}}$ and $f_{\text{RIE}}$ the distribution $\rho_\Delta$ by $\rho_\Delta = \mu^\star \boxplus \sigma_{\text{s.c.}, \sqrt{\Delta}}$.

# H  Details on the numerics

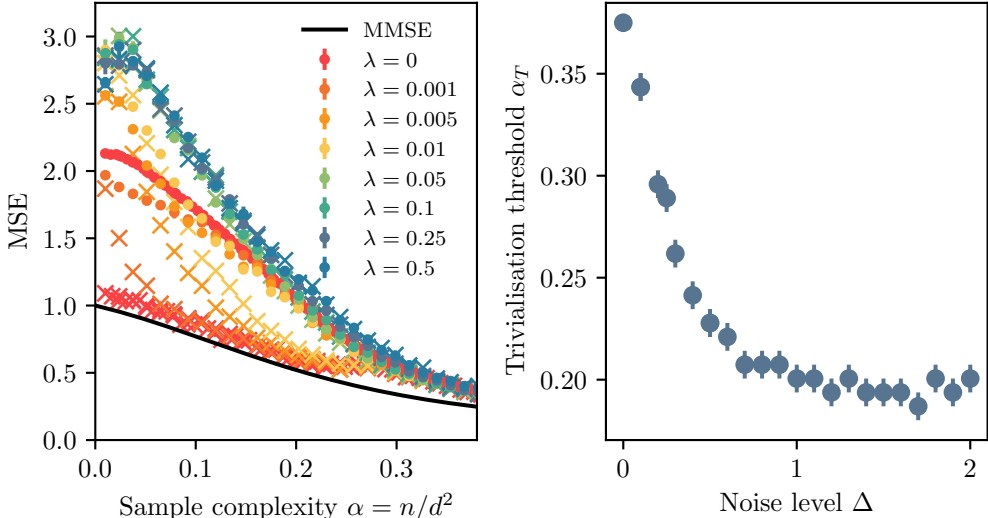

Figure 5: Left: Mean squared error as a function of the sample complexity $\alpha$, for $\kappa = 1/2$ and $\Delta = 0.25^2$. Dots are simulations using GD with a single initialization averaged over 32 realizations of the dataset, crosses are averages over 64 initializations. The continuous line is the asymptotic MMSE given by (7). The colors indicate the strength of the regularization. Right: Trivialization threshold in the sample complexity $\alpha_T$ as a function of the noise level $\Delta$ in the teacher without regularization, $\lambda = 0$. The measurement has a resolution of $0.1$ on the noise level and of $0.007$ on the sample complexity

## H.1  Solutions to the "state evolution" equations

We describe here how to solve eqs. (7),(8). The first step to solve is to obtain an analytical expression for $\mu_t$. We refer to Appendix A for the definition of quantities used in this section. We recall that $\mu_t := \mu_{\mathrm{MP},\kappa} \boxplus \sigma_{\mathrm{s.c.},\sqrt{t}}$ is the free convolution of the Marchenko-Pastur law and a scaled semicircular density. The $\mathcal{R}$-transform of the scaled semicircle distribution is [Tulino and Verdú, 2004]:

$$\mathcal{R}_{\sigma_{\mathrm{s.c.},\sqrt{t}}}(z) = zt,$$

while for the Marchenko-Pastur law we have

$$\mathcal{R}_{\mu_{\mathrm{MP},\kappa}}(z) = \frac{\kappa}{\kappa - z}.$$

We can now use (cf. Appendix A):

$$\mathcal{R}_{\mu_t}(z) = \mathcal{R}_{:=\mu_{\mathrm{MP},\kappa} \boxplus \sigma_{\mathrm{s.c.},\sqrt{t}}} = \mathcal{R}_{\sigma_{\mathrm{s.c.},\sqrt{t}}}(z) + \mathcal{R}_{\mu_{\mathrm{MP},\kappa}}(z) = zt + \frac{\kappa}{\kappa - z}.$$

The Stieltjes transform $g(z) = \mathbb{E}_{\mu_t}[1/(X - z)]$ of $\mu_t$ is the solution of the equation

$$z + \frac{1}{g(z)} = \mathcal{R}_{\mu_t}(-g(z)),$$

or equivalently

$$z = -tg(z) + \frac{\kappa}{\kappa + g(z)} - \frac{1}{g(z)}. \tag{108}$$

Among all the solutions to this equation, $g(z)$ must be such that $\mathrm{Im}[g(z)] > 0$ if $\mathrm{Im}(z) > 0$, and also satisfies $g(z) \sim 1/z$ for $z \to \infty$. Eq. (108) is a third degree polynomial in $g(z)$, and can easily be solved by algebraic solvers, and has a single solution satisfying the constraints we described. Finally, $\mu_t(x)$ is given by the Stieltjes-Perron inversion theorem (see Appendix A):

$$\mu_t(x) = \lim_{\varepsilon \to 0} \frac{\mathrm{Im}[g(x + i\varepsilon)]}{\pi}, \tag{109}$$

and we numerically choose $\varepsilon = 10^{-8}$. We now discuss the computation of the integral of $\mu_t(x)^3$ in (8). Notice that the integrand is only non-zero over at most two finite intervals. Exact values of the edges are given by setting the discriminant of equation (108) to zero. The last step is finding a solution in $\hat{q}$ to equation (8). We find the function "root" in Scipy, which uses a variant of the Powell hybrid method, to be performing quite well when initialized in the value $2\alpha/Q_0$. This whole procedure is quite efficient and can be reproduced easily on any machine.

## H.2   Gradient descent

In our experiments with gradient descent we are minimizing the objective $\mathcal{R}(\mathbf{W})$:

$$\mathcal{R}(\mathbf{W}) \coloneqq \frac{1}{4} \sum_{i=1}^{n} \left( y_i - f_{\mathbf{W}}(\mathbf{x}_i) \right)^2 + \frac{\lambda}{2} \sum_{k=1}^{m} \sum_{l=1}^{d} w_{kl}^2. \tag{110}$$

All the simulations are done in PyTorch with the student weights initialized in the prior. For "vanilla" gradient descent we iterate until convergence, and average over several repetitions. For averaged gradient descent (AGD) we first generate the dataset, then train the student several times with starting weights independently sampled in the prior, and "average the weights" at the end of training. By this we mean that for each run we train until convergence, then obtain the matrix $\mathbf{S}$ and average it. Finally, we average this procedure over several repetitions. The learning rate is chosen to be suitably large, as it's typically better to train a networks with giant steps [Dandi et al., 2023].

In Figure 2 the gradient descent is run for zero regularization, $\lambda = 0$. In Figure 5 (left) we then study the effect of regularization to check whether regularization helps to achieve the Bayes-optimal error, but conclude that it does not and in fact it hurts the performance. In Figure 5 (right) we study the effect of the noise on the landscape of GD. We will expand on this in Appendix H.3. All the error bars reported in Figure 2 and Figure 5 (left) are standard deviations of the MSE measured on the samples. Figure 5 (right) has a finite resolution indicated in the caption. A single run of vanilla GD for the models we display can be completed in at most 30 minutes on an average machine without using GPUs. For producing our figures we used around $30\,000$ hours of computing time.

## H.3   Additional experiments with GD

Here we study in more detail the phenomenology observed in Figure 2 (right) where in the presence of noise and at a large sample complexity all the runs of GD seem to converge to the same prediction. In the figure we noticed that above certain sample complexity the averaged and non-averaged GD errors are identical. This suggests that GD will eventually lead the weights of the network to the same configuration up to the symmetries of the problem independently of the initial state. We call this a *trivialization* of the landscape.

In Figure 5 (right) we study the trivialization threshold as a function of the noise level $\Delta$. One needs to take care of the symmetries on $\hat{\mathbf{W}}$, so we first define $\hat{\mathbf{S}}$:

$$\hat{\mathbf{S}}(\mathbf{W}^{(0)}, \mathcal{D}) \coloneqq \frac{1}{m} \left( \hat{\mathbf{W}}(\mathbf{W}^{(0)}, \mathcal{D}) \right)^{\mathsf{T}} \hat{\mathbf{W}}(\mathbf{W}^{(0)}, \mathcal{D}),$$

where we mean that for a fixed dataset we run GD, then take a matrix product to obtain $\mathbf{S}$. This procedure allows us to define the **dispersion**

$$\delta_{GD} \coloneqq \mathbb{E}_{\mathcal{D}} \left[ \text{tr} \left( \mathbb{E}_{\mathbf{W}^{(0)}} \left[ \hat{\mathbf{S}}(\mathbf{W}^{(0)}, \mathcal{D}) \right] - \hat{\mathbf{S}}(\mathbf{W}^{(0)}, \mathcal{D}) \right)^2 \right].$$

If the dispersion becomes zero it means that all the runs will converge to the same value. As we increase the sample complexity $\alpha$ the dispersion decreases, until it becomes zero. For each value of the noise level $\Delta$ we indicate the minimum sample complexity for which the dispersion is either less than $10^{-2}$, or less than $10^{-3}$ of the maximum dispersion at fixed $\Delta$.

In Figure 5 (left), where we studied the effect of $\ell_2$ regularization on the weights, we can also see how even a relatively small $\lambda > 0$ regularization leads to a trivialization of the landscape again in the sense that different initializations of GD provide the same prediction and averaging does not lead to a better error.

