# OpenReview forum: "Bayes-optimal learning of an extensive-width neural network from quadratically many samples"
_NeurIPS.cc/2024/Conference — NeurIPS 2024 poster_

### Official Review · Reviewer_XJ6b · 2024-07-01

**Soundness:** 4
**Presentation:** 4
**Contribution:** 4
**Rating:** 8
**Confidence:** 2

**Summary:**

Studies a teacher-student setting with very specific configurations of input data, teach weight distribution, noise distribution and teacher/student architecture. The minimum MSE achievable, Eqn (3), is subjected to a particular limit, Eqn (4), and shown in closed-form to be Eqn (14). From Eqn 14 and assuming the noiseless case., we can derive the "perfect-recovery" threshold, which is stated as one of the main contributions (Eqn (1)).

**Strengths:**

* I found the paper very accessible despite having little pre-existing knowledge of this line of research.
* The context of the current work is illustrated clearly. I did not have to do too much background reading to appreciate what has been done, and how the current submission contributes to advancing this line of research
* Claim 1, which is central to the main result, is shown using the replica method as well as a more rigorous method.
* For the particular setting described in Section 2, this work derives the MMSE under the regime described in Equation (4). This is novel as far as I am aware.
* The empirical comparison between Bayes-optimal and (non-stochastic) gradient descent reveals an intriguing hypothesis that averaged over random initialisations, GD is close to being Bayes-optimal

**Weaknesses:**

* In terms of exposition, I found it a bit awkward that the closed-form asymptotic limit of MMSE_d is given in Eqn 14, which is outside of Section 3 “Main theoretical result”. I think the part of Section 5 titled “Results for the Bayer optimal estimator” is ready to be presented at the beginning of Section 3.
* The notation and setup for Claim 1, to my eyes, belongs to Section 4 “Derivation of the main results”
* It’s hard for me to imagine how this type of analysis can be generalised to a more complex architecture than the one-hidden layer network presented in Eqn (2). And I’m not talking about very exotic architectures. I mean even a small generalisation to multiple hidden layers. Is this really tenable using the theoretical tools employed here?

**Questions:**

* Line 151 defines $MMSE=\lim_{d \to \infty} MMSE_d$. But from Eqn (4), it seems that the asymptotic regime is more than just taking $d \to \infty$. I found this confusing.
* Is Eqn (3) derived somewhere? If not, is it because it’s very elementary and can be found in a standard reference? Please give citation then.
* Is it obvious that the GD estimator is the same as the ERM estimator in the discussion starting on Line 160?

**Limitations:**

Some of the limitations that come to mind have already been acknowledged in the submission. Specifically, there is a need to go beyond Gaussian input data, etc, see Pg. 9 Line 350

---

> ### Author Rebuttal · Authors · 2024-08-06
>
> We thank the reviewer for their time and dedication. Please refer to the common answer for the recurring questions.
>
> - The asymptotic limit (which we generically denote by $\lim_{d \to \infty}$) is achieved by defining $m(d) = \kappa d$, $n(d) = \alpha d^2$, and then taking the limit of $d\to\infty$. This corresponds to take $m$ and $n$ to infinity while keeping the ratios $m/d$ and $n/d^2$ finite. We have added a clarification of this point in our revision.
>
> - Regarding the GD and ERM estimators, we want to clarify that they are in general not the same. The GD estimator is reached from a random initialization, and in non-convex scenarios (such as here when $\kappa < 1$) is generally not a global minimizer of the empirical risk (which the ERM estimator is, by definition). The ERM estimator is generically not reachable by polynomial-time algorithms in non-convex problems. We study numerically the minimizer reached by GD on the stated loss from random initialization.

---

> > ### Comment · Reviewer_XJ6b · 2024-08-12
> > **Thank you!**
> >
> > Thanks to authors for their detailed response. I maintain my original rating.

---

### Official Review · Reviewer_NWbd · 2024-07-03

**Soundness:** 4
**Presentation:** 4
**Contribution:** 3
**Rating:** 8
**Confidence:** 3

**Summary:**

This paper explores the performance of an optimal (in the Bayes-Optimal sense) estimator in the teacher-student setting with aligned architectures being one-hidden layer neural networks with quadratic activations. The number of neurons is extensive in the dimension. Given previous works where the "proportional limit" regime was studied (i.e. asymptotically square dataset), finding that a vanilla estimator could attain optimal performance, they focus on a slightly different regime. Precisely, the author(s) build upon numerical observations of other works and let the number of samples scale quadratically in the dimension.
Having chosen the playground, the analysis proceeds with recasting the problem to a form that exploits the symmetries of the objective function, and a replica computation of the free energy. Such computations allow the author(s) to derive an asymptotic formula of the Minimum Mean Square Error.
To back the Physics-based derivation, they discuss a break-down of the steps required to formalize it completely. Of the three main points, two are conjectures. The final one instead has a mathematical proof.
Given that Gradient Descent is the go-to method, they explore its performance in relation to the theoretically optimal MMSE. Interestingly, they find that in the noiseless case it appears to sample weights from the posterior. The same picture breaks when the labels have noise. There, they discuss the appearance of a phase transition that trivializes the randomness over initialization of GD.

**Strengths:**

- Equation (1) and its natural link to the degrees of freedom of the target function is an appealing result.
- The connections with ellipsoid fitting, computations in random matrix theory, formalized results on neural networks, and extensive-rank matrix denoising validate it. In simple words: it is not a made-up paper, but it is well positioned in between earlier works.
- Understanding industry standard is crucial. Gradient Descent is the idealized version of what is actually used in practice. It is therefore valuable to understand it, as the author(s) did.
- The result about GD opens up to continuations of the work into such direction of understanding formally what is happening there. This paper is not a dead-end.
- Section 4 is an interesting take on trying to vindicate the replica method. I believe there is value in trying to break down what would be needed to make the Physics derivation a mathematically rigorous one. At the same time, it is of independent interest to use the replica method, as it tells us what we have to prove. Anyways, nice idea.
- I was kind of surprised to see that as $\kappa$ increases the MMSE worsens.
- The techniques used are, in my honest opinion, beautiful and effective.
- Even if the MMSE cannot be simulated (unless an algorithm achieving it was known), the author(s) bring to the table experiments for GD that are to the point and motivate further work.
- The references are, to my limited single human reading stack, to the point.
- The weaknesses I will discuss below are to be understood in the sense that I understand that this is a paper with lots of Physics-flavour. Personally, I agree with the modelling idea of finding a solvable model, hence the choice of a "simplified setting". Also, this simplified setting already requires non-trivial computations, and is therefore worth exploring. Nevertheless, I will outline what worries me to stimulate the discussion.

Overall, I believe this was a good read. Thanks to the author(s)!

**Weaknesses:**

As mentioned in lines 350-353, the setting is very restricted. While universality might come in help, at least for conjecturing that some aspects extend nicely in the high dimensional limit, there are at least three features that in my opinion are more complicated. Below, I will elaborate on them.
1. Perhaps the solvable one. You consider matching architectures, with $m = m^{\star}$, I wonder, what is the technical challenge in analyzing the case in which the activations are the same but $m\neq m^{\star}$?
2. Your target function is "neuron agnostic" in the sense that it takes a simple mean of the teacher neurons. How would your technique behave when we take a one hidden-neuron that also has random Gaussian weights $\{a_k\}_{k = 1}^m$ in the second layer? I have not thought this through extensively.
3. The most challenging: a quadratic activation. While the Statistical Physics literature focuses a lot on quadratic activations/phase retrieval, which is thought to be an interesting problem, I am concerned about the extension in this setting to other activations. It appears to me that when recasting the task to a matrix estimation problem (lines 175-180), one heavily relies on "expanding the square" (lines 175). In other words, the symmetry of the activation function naturally builds the matrix $\mathbf{S}^{\star}$  (lines 176-177). Considering other (e.g. non-polynomial, or even non-square) activations, it looks to me as a very hard task. I do appreciate that the author(s) recognized this, but would be even happier if, in the spirit of section 4, the bonus page for publication were spent to discuss how a quadratic activation is not a dead-end.
   More in depth, can you bring over an argument that can convince the reader that the method is worth pushing over the boundary of something that allows you to over-use Gaussian integrals (in the sense that square activations are perhaps the easiest non-linearity)?

The second concern is mostly structural and philosophical. I am convinced that all the results of the paper are of interest to the community. However, from my first reads, I had the impression that the "Replica argument", its "formalization" and the "analysis of GD" had little meaning if placed all inside a conference paper.
For example, while Section 4 is very useful for grounding the replica method, I only appreciate its value in the sense that it connects the dots with other works. In some sense, I find this counter-intuition that there are very few resources on Gaussian Equivalence and modern Statistical Physics of neural networks that discuss this, and then find a nice Section clearing out some aspects in a paper that should be 9 pages. I do wish at some point that these scattered nice sections will be condensated in some resource that will be accessible (e.g. a book).
Similarly, the analysis of GD could be justified for its industrial importance, and even simply because it is the quickest algorithm to compare with the MMSE prediction, but at the same time it poses the question: "What is the soul of this work?".
Please note that this weakness is more provocative than actually requiring action.

**Questions:**

Can you please make the legend in Figure 2 (right) more explicative? I obviously get it, but do not understand why you chose to "condensate it". As a reader, the others are easier to parse, as they label everything.
- The regularization of Figure 3 is never discussed in text, why do you add it? Just to show that the distance from the MMSE becomes larger? What exactly counts to you as "trivialize the gradient descent landscape almost completely" in Figure 3 (Left), as you claim in lines 802-803? Please note that I understand the value of Figure 3 (right).
- Related to the point above, why do you seek to show regularized objectives in the first place? I mean, what motivates your experiment?
- Please, see the first weakness, where I planted some questions.
### Typos
I will write down in this subsection what I noticed by reading. Take it as a friendly contribution. All of them are largely not relevant to the evaluation of the paper.
- The instruction block for the checklist was not deleted.
- (line 161) you write "student", but I noticed that in other instances you used a different way of placing quotation marks, that is the TeX standard.
- Equation (5) lacks punctuation.
- (line 222) the end of the sentence "using probabilistic technical amenable to rigorous treatment" probably changed wording, maybe "technical"was meant to be "techniques"?
- (line 779) the "root" quotation marks.
- (line 792) is $30,\,000$ really the right number? Why three zeros? Maybe I am just not used to this standard.

**Limitations:**

None.

---

> ### Author Rebuttal · Authors · 2024-08-06
>
> We thank the reviewer for their time and dedication. Please refer to the common answer for the recurring questions.
>
> - (W1) -- It is indeed possible to consider that while the teacher has $m^\star$ hidden units, the student has $m$ with $m > m^\star$ ("overparametrized" regime) or $m < m^\star$ (``underparametrized'' regime). While we have extended part of our analysis -- namely the reduction to matrix estimation and the universality conjecture -- to this case (provided both $m^\star, m = \Theta(d)$), new difficulties arise in the analysis of the resulting matrix generalized linear model, as a consequence of the lack of the Bayes-optimality. Nevertheless, based on so-called "dilute" limits of HCIZ integrals [E1], we were able to show that in the noiseless setting, in the overparametrized case $m > m^\star$, the posterior mean estimator reaches perfect recovery for $\alpha_{\rm PR} = \kappa - \kappa^2/2$ (where $\kappa = m/d$). This means that here the posterior mean estimator is far from being Bayes-optimal (as the Bayes-optimal estimator reaches perfect recovery at $\kappa^\star - (\kappa^\star)^2/2$ with $\kappa^\star = m^\star/d$ as we saw).
>
>  - (W2) -- It is indeed possible to consider learning the second layer weights. We verified that this effectively leads to consider the same matrix estimation problem, with ${\bf S}^\star$ now generated as
>
> $$
>     {\bf S}^\star = \frac{1}{m} \sum_{k=1}^m a_k^\star {\bf w}^\star_k ({\bf w}^\star_k)^\top,
> $$
>
> and $(a^\star_k)_{k=1}^m \sim P_a$ are drawn i.i.d., for an arbitrary distribution $P_a$ (not necessarily Gaussian). The case studied in our work then corresponds to $P_a = \delta_1$. We have checked that our analysis generalizes to this setting, and in particular we have derived the equivalent of Claim 1 for this setting. Remarkably, the conclusion is that the MMSE formulas (eqs.(14) and (15)) generalize directly, by replacing the Marchenko-Pastur law with a *generalized Marchenko-Pastur* (or free compound Poisson distribution), which is the free multiplicative convolution of the Marchenko-Pastur law and $P_a$. This distribution can be analytically characterized via its $R$-transform. We have added a detailed statement of these results in a new appendix of the revised version of the paper, but have not explored the phenomenology of this setting further.
>
> - Q1: we will make the legend of Fig.2 explicit.
>
> - (Q2 and Q3: On regularization and trivialization of the landscape) By "trivialization" of the landscape we mean that gradient descent from random initial conditions goes to a minimizer corresponding to the same function, independently of the initialization. As we discussed, this is not expected for $\kappa < 1$, as the problem is then not convex. We study the effect of adding $\ell_2$ regularisation on the weights, as we can see in Fig.2 (right) that the averaged GD algorithm seems to have an "interpolation peak" in noisy settings: it performs worse and worse as $\alpha$ increases, until it becomes identical to non-averaged GD at $\alpha = \alpha_{\rm triv}$. At this point $\alpha_{\rm triv}$ we therefore say that the landscape is trivialized. In simple networks with only one hidden unit, this interpolation peak is mitigated by adding such a a regularization [E2]. We can see in Fig.3 (left) that this does not happen for our network with an extensive number of hidden units: instead, the regularization decreases the trivialization threshold $\alpha_{\rm triv}$. Fig.3 (right) studies the effect of the noise level on this phenomenon.
>
> - We thank the reviewer for the list of typos that we will fix.
>
> - We thank the reviewer for their suggestion to write a book about the general approach. While we agree this is needed, and appreciate the comment, this is clearly beyond the scope of the paper. As for the "soul of the work", we are unsure what to answer.
>
>
> -- [E1] "Instanton approach to large-$n$ Harish-Chandra-Itzykson-Zuber integrals", Bun&al, PRL (2014).
>
> -- [E2]  "Optimal Regularization can Mitigate Double Descent", Nakkiran&al, ICLR (2020).

---

> > ### Comment · Reviewer_NWbd · 2024-08-08
> > **Comment to rebuttal**
> >
> > Dear author(s),
> > thank you for your explanatory comments. In this comment, I also acknowledge the general one you wrote for all reviewers.
> >
> > ## General comments
> > Everything understood: as long as you state in your limitations the points you mention in the way you mention them here, we are done with these questions on my side.
> >
> > ## Specific comments
> > - W1 & W2. Great to hear this. I am curious to see what the final version will be.
> > - Q1-Q2-Q3. Thank you for this. For the trivialization, ok.
> > - thank you for the typos.
> > - as for the last comment, yes, it was more of a rant. Obviously out of scope.
> > - as per the "soul of this work", I do stand with the idea that this is far more than a 9 page conference paper, but this is not a tragedy.
> >
> >
> > Considering the points made above, I have raised my score according to the NEURIPS24 guidelines. I will keep an eye on the discussion with other reviewers and potentially update my score if other thresholds are surpassed.

---

### Official Review · Reviewer_xzmx · 2024-07-11

**Soundness:** 3
**Presentation:** 3
**Contribution:** 2
**Rating:** 6
**Confidence:** 2

**Summary:**

The paper presents some learning theory for learning the large-dimension, large width perceptrons with a quadratic activation functions. The results seem to be twofold: Claim 1, and its specialisation in eq 14, which gives us the MSE test error. In addition they note that empirically the  that the SGD solution averaged over random initialisations leads to a near-bayes-optimal learning error.

**Strengths:**

This seems to be a simple and elegant example of learning for a well-controlled class of high-dimensional problems.

**Weaknesses:**

This is a rather technical result far removed from obvious useful applications; this is not a criticism in itself — we are all aware that Neurips is moving into the territory of COLT etc. However, in a generalist conference it is hard to communicate the significance of a technical result like this. This is not my area, so I am largely taking the author's assertions about the significance and novelty of this result at face value.

**Questions:**

As a non-specialist in this class of problems, my questions are simply why we are concerned with this class of problems? The authors mention connections to phase retrieval and matrix denoising problems; can we give a concrete example, however contrived, that motivates this work by its potential to eventually contribute to a concrete application in those domains,

**Limitations:**

The author's explanations of the limitations are absolutely clear as far as I can see.

---

> ### Author Rebuttal · Authors · 2024-08-06
>
> We thank the reviewer for their time and dedication. Please refer to the common answer for the questions.

---

> > ### Comment · Reviewer_xzmx · 2024-08-11
> >
> > I am embarrassed at the quality of my review. I had intended to return and expand upon it before the deadline, but did not. I apologies to the authors and the AC for this low-quality feedback.
> >
> > The main difficulty is that this paper is relatively far from my area of expertise and I do not feel qualified to assess its relationship to the broader field, which seems for this paper to be the crucial matter. I will keep the confidence of my review low in the hope that more-informed readers will provide more meaningful feedback.

---

### Official Review · Reviewer_bzi8 · 2024-07-12

**Soundness:** 2
**Presentation:** 2
**Contribution:** 2
**Rating:** 4
**Confidence:** 2

**Summary:**

The paper explores Bayes-optimal learning of a neural network with extensive input dimensions and a single hidden layer using quadratic activations. It presents a closed-form expression for the optimal test error when the sample complexity is quadratic. This work connects to matrix denoising and ellipsoid fitting, providing both theoretical insights and empirical validations. Key findings include showing that randomly initialized gradient descent can approach the Bayes-optimal error.

**Strengths:**

The paper is robust in its theoretical analysis, providing clear mathematical derivations and a solid theoretical framework. The breadth of related works cited enriches the context and demonstrates the paper’s relevance to current research frontiers.

**Weaknesses:**

The motivation for focusing on Bayes-optimal accuracy with quadratic activation remains under-explained, which may leave readers uncertain about the broader applicability of the results. Furthermore, the implications of these findings for common neural network training techniques using ReLU or sigmoid activations are not discussed, which could limit the paper’s appeal to a broader audience.

**Questions:**

1. Given the prevalence of activation functions like ReLU and sigmoid in practical applications, what prompted the choice of quadratic activation for this study?

2. In Lines 142–144, the term $P_{prior}(W)$ is mentioned as a prior on the teacher weights $W^*$. Shouldn’t this be $P_{prior}(W^*)$ instead? Could you clarify whether $W$ refers to teacher weights or student weights?

3. How does the Bayes-optimal accuracy relate to the practical performance of gradient descent (GD)? Does high Bayes-optimal accuracy imply high accuracy for GD?

4. Could you explain the origin and significance of the term $\Delta(2+\Delta)$ in Eq. (3)?

5. In Line 128, there appears to be an extraneous 'm' following the definition of $D$.

**Limitations:**

Yes.

---

> ### Author Rebuttal · Authors · 2024-08-06
>
> We thank the reviewer for their time and dedication. Please refer to the common answer for most of their questions.
>
> - (Q2) -- $P_{prior}({\bf W})$ is the distribution from which the teacher weights ${\bf W}^\star$ are sampled, which we denote ${\bf W}^\star \sim P_{prior}$. Since this distribution is assumed to be known to the student, it appears naturally (as a consequence of Bayes' law) in the posterior distribution of the weights given the dataset.

---

> > ### Comment · Reviewer_bzi8 · 2024-08-13
> >
> > Thank you for answering my questions. I keep my original score.

---

### Author Rebuttal · Authors · 2024-08-06

We thank all referees for their interest in our work and their comments that will help to clarify our paper. We will answer here to the most common questions of the reviewers.

- **Why are square-activated networks relevant?** (Q1 of bzi8. Question of xzmx. W3 of NWbd. W3 of XJ6b). We want to stress that this is the first first work that studies non-linear two layer neural networks of extensive width, with enough samples to have feature learning. Furthermore, analogous models have been considered in the machine learning literature for analyzing the implicit bias of gradient descent [A1-A3] and as toy models for probing the advantage of depth in neural networks [B1, B2]. These existing works derive bounds, whereas our tight asymptotic result could help to settle the concerned questions about implicit regularization and advantage of depth more tightly. We thus anticipate that already the quadratic loss case will have broader impact along these lines. We will add these motivations in a revised version.

- **Why is it important in the proof to use square activations?** Q1 of bzi8. W3 of NWbd. As we emphasize in the paper, our approach relies on the square activation function. This allows to implement the main idea of our analysis, which is to reduce this problem of Bayes-optimal learning of a square-activated neural network of extensive width to the Bayes-optimal denoising of a matrix of extensive rank. Going beyond quadratic activations, e.g. to ReLU or sign, is a serious challenge: even if one were to consider a monomial activation of degree $k$ (i.e. $\phi(x) = x^k$), our analysis would map the learning problem to the Bayes-optimal denoising of a $k$-tensor of extensive rank. While denoising low-rank tensors has attracted a lot of attention (see for instance [C1-C3]), the extensive-rank case is still a widely open problem. We will add a discussion of this point in a revised conclusion.

- **Does knowing the Bayes-Optimal performance inform us on Gradient Descent?** Q3 of bzi8. The general answer is no. The gradient descent (GD) algorithm has no information on how the data is generated, so it is expected to be sub-optimal in general. Furthermore, for $\kappa < 1$ the problem is non-convex, so it is *a priori* reasonable to suspect that GD would require $\alpha>\alpha_{\rm PR}$ to achieve zero error (in the noiseless setting). Interestingly, we observe that, in the noiseless setting, $(i)$ a number $\alpha=\alpha_{\rm PR}$ of samples is sufficient for GD to reach perfect recovery, *and* $(ii)$ that averaging over different runs of GD gives Bayes-Optimal performance, for any value of $\alpha$. These are two surprising facts that our theory doesn't account for, and whose explanation is left for future work. We also emphasize that these two observations no longer hold in noisy scenarios (see Fig.3 right). However, it is worth pointing out that our Bayes-optimal analysis can inform us on the performance of an Approximate Message Passing algorithm [D1, D2] that will achieve the Bayes-optimal performance also in noisy settings. We will include an explicit description and analysis of this algorithm in the final version of the paper.

- Q4 of bzi8. Q2 of XJ6b. We thank the reviewers for suggesting a more detailed discussion of the origin of eq.(3). We study the Bayes-optimal performance of the network where the input data is corrupted by a Gaussian noise of variance $\Delta$, cf eq.(2). We clarify that the ``most natural'' definition of the MMSE would be (for $(x, y)$ new test samples):

$$
R = \frac{1}{2} E[(y- \hat{y}_{\mathcal{D}}^{\rm BO}(x) )^2].
$$

Eq.(3) is the same quantity as $R$, up to a rescaling and an additive constant. As we show in Section 3, eq.(3) is equivalent to the matrix MMSE $\kappa E{\rm tr}[({\bf S}^\star - {\bf S})^2]$, in a problem with noiseless input data but noisy labels, with a variance $2\Delta(2+\Delta)/\kappa$. More generally, one should consider eq.(3) as the rescaling of $R$ that satisfies three properties: $(i)$ it is finite in the high-dimensional limit, $(ii)$ it is equal to $1$ for $\alpha = 0$ (in the absence of data), and $(iii)$ it goes to $0$ in the case of perfect recovery (if the posterior concentrates around $W^\star$). We will add in the revised version a detailed discussion of this point, and a mathematical proof of the equivalence of the MMSE of eq.(3) to the matrix MMSE $\kappa E {\rm tr}[({\bf S}^\star - {\bf S})^2]$.

- W1 and W2 of XJ6b. We thank the reviewer for suggesting a better organization of the main results of the paper. In the revised version, we will move eqs.(14) and (15) on the asymptotic MMSE to the beginning of the main results section. We will also clarify the relevance of the ``free entropy'' studied in Claim 1, recalling its relation to the mutual information, and thus to optimal estimation.

-- [A1] "Implicit Regularization in Matrix Factorization", Gunasekar&al, NeurIPS 2017.

-- [A2] "Towards Resolving the Implicit Bias of Gradient Descent for Matrix Factorization: Greedy Low-Rank Learning", Li&al, ICLR 2021.

-- [A3] "Small random initialization is akin to spectral learning: Optimization and generalization guarantees for overparametrized low-rank matrix reconstruction", Stöger&al, NeurIPS 2021.

-- [B1] "Provable Guarantees for Nonlinear Feature Learning in Three-Layer Neural Networks", Nichani&al, NeurIPS 2023.

-- [B2] "Learning Hierarchical Polynomials with Three-Layer Neural Networks", Wang&al, ICLR 2024.

-- [C1] "Statistical and computational phase transitions in spiked tensor estimation", Lesieur&al, ISIT 2017.

-- [C2] "The landscape of the spiked tensor model", Ben Arous&al, Comm. Pure Appl. Math. (2019).

-- [C3] "Statistical limits of spiked tensor models", Perry &al, Ann. Inst. H. Poincaré (2020).

-- [D1] "Message-passing algorithms for compressed sensing", Donoho&al, PNAS (2009).

-- [D2] "Generalized approximate message passing for estimation with random linear mixing", Rangan, ISIT 2011.

---

### Decision · Program_Chairs · 2024-09-25

**Decision:**

Accept (poster)

**Comment:**

This paper assumes a neural network teacher model
and studies the performance of Bayes-optimal estimator.
Authors assume a target function which is a one-hidden layer
neural network with large width, quadratic activation, and random weights.
They assume quadratically many samples,
derive a closed-form expression for the Bayes-optimal test error.
They further complement their results with empirical studies and compare
the Bayes-optimal performance to the one obtained by gradient descent,
supporting their story through numerical studies.


This paper was reviewed by four reviewers and received the following Scores/Confidence: 8/3, 4/2, 6/2, 8/2. I think the paper is studying an interesting topic and the results are relevant to NeurIPS community. The following concerns were brought up by the reviewers:

- poor motivation for broader audience. This should be addressed as it was a main concern of one of the reviewers.
- restricted setting. Although it is unfair to ask authors to generalize this setting, the limitations of the studied setting should be explicitly discussed in the paper.
- presentation can be improved in various places; see reviewer XJ6b's comments.


Authors should carefully go over reviewers' suggestions and address any remaining concerns in their final revision. Based on the reviewers' suggestion, as well as my own assessment of the paper, I recommend including this paper to the NeurIPS 2024 program.